# Atlas of proteomic signatures of brain structure and its links to brain disorders

Peng Ren[1,2,3,12], Xiao-He Hou[1,2,3,4,12], Zeyu Li[1,2,3], Jia You [1,2,3], Yuzhu Li[1,2,3], Wei Zhang[1,2,3], Weikang Gong [5], Bei Zhang [1,2,3], Bangsheng Wu[1,2], Linbo Wang [1,2,3], Chun Shen[1,2,3], Yujie Zhao[1,2,3], Qing Ma[1,2,3], Jujiao Kang [1,2,3], Yuchao Jiang [1,2,3], Neil Roberts [6], Fan Xu [7], Yong He [8,9], Jin-Tai Yu[1,2] ✉, Meiyun Wang[10] ✉ & Wei Cheng [1,2,3,11] ✉

Individual variation in brain structure influences deterioration due to disease and comprehensive profiling of the associated proteomic signature advances mechanistic understanding. Here, using data from 4997 UK Biobank participants, we analyzed the associations between 2920 plasma proteins and 272 neuroimaging-derived brain structure measures. We identified 5358 associations between 1143 proteins and 256 brain structure measures, with NCAN and LEP proteins showing the most associations. Functional enrichment implicated these proteins in neurogenesis, immune/apoptotic processes and neurons. Furthermore, bidirectional Mendelian randomization revealed 33 associations between 32 proteins and 23 brain structure measures, and 21 associations between nine brain structure associated proteins and ten brain disorders. Moreover, the significant associations between the identified proteins and mental health were mediated by brain volume and surface area. In summary, this study generates a comprehensive atlas mapping the patterns of association between proteome and brain structure, highlighting their potential value for studying brain disorders.

Magnetic Resonance Imaging (MRI) continues to play a pivotal role in the study of individual variation in the deterioration of brain structure in a wide range of neurodegenerative and neuropsychiatric disorders[1,2]. Alterations in brain structure are among the earliest pathophysiological changes detectable in neurodegenerative disorders and can manifest decades before overt clinical symptoms appear[3,4]. Besides, longitudinal MRI can potentially be used to monitor disease progression across multiple stages and to predict the risk of developing brain disorders in healthy individuals[4,5]. Furthermore, analysis of brain structure heterogeneity may allow the classification of patients into subtypes with different clinical manifestations and temporal trajectories[6,7]. Altogether, MRI-derived measures of brain

[1]Institute of Science and Technology for Brain-Inspired Intelligence, State Key Laboratory of Medical Neurobiology and MOE Frontiers Center for Brain Science, Fudan University, Shanghai, China. [2]Department of Neurology, Huashan Hospital, Fudan University, Shanghai, China. [3]Key Laboratory of Computational Neuroscience and Brain-Inspired Intelligence (Fudan University), Ministry of Education, Shanghai, PR China. [4]Department of Neurology, Qingdao Hospital, University of Health and Rehabilitation Sciences (Qingdao Municipal Hospital), Qingdao, China. [5]School of Data Science, Fudan University, Shanghai, China. [6]Centre for Reproductive Health (CRH), Institute for Regeneration and Repair (IRR), University of Edinburgh, 4-5 Little France Drive, Edinburgh EH15 4UU, UK. [7]Institute of Mechanics and Computational Engineering, Department of Aeronautics and Astronautics, Fudan University, Shanghai, China. [8]State Key Laboratory of Cognitive Neuroscience and Learning, Beijing Normal University, Beijing, China. [9]Beijing Key Laboratory of Brain Imaging and Connectomics, Beijing Normal University, Beijing, China. [10]Department of Medical Imaging, Henan Provincial People's Hospital & Zhengzhou University People's Hospital, Zhengzhou, China. [11]Fudan ISTBI—ZJNU Algorithm Centre for Brain-inspired Intelligence, Zhejiang Normal University, Zhejiang, China. [12]These authors contributed equally: Peng Ren, Xiao-He Hou. ✉e-mail: jintai_yu@fudan.edu.cn; mywang@zzu.edu.cn; wcheng@fudan.edu.cn

structure may play a crucial role in diagnosing, stratifying, predicting and monitoring brain disorders.

Considering that proteins are the final products of gene expression that contribute directly to variation in brain structure, profiling of the proteomic signature of brain structure has great potential for revealing the microscale mechanisms that underlie brain disorders. It is well established that the presence of proteins in cerebrospinal fluid (CSF) can inform on changes in brain structure that occur in a wide range of brain disorders. In particular, total tau (or isoforms) and neurofilament light chain proteins are related to the degree of neurodegeneration in Alzheimer's disease. Measurement of the reduction of complement proteins was associated with the extent of brain atrophy in mild cognitive impairment patients[8], and α-synuclein is the critical protein that precedes overt neurodegeneration in Parkinson's disease[9,10]. Furthermore, an association between concentration in CSF and plasma has been reported for many proteins[11]. So that analysis of plasma can also be used to determine the level of proteins in the brain. Accordingly, measurement of the concentration of amyloid beta 42 and phosphorylated tau at amino acid 181 in plasma has been validated, by corresponding studies of CSF and brain tissue, to be a biomarker of Alzheimer's disease[12,13]. An association between the expression of plasma proteins and alteration in brain structure in several brain disorders, as well as in older aging, has been indicated by several studies[14–17]. Nevertheless, these initial studies are limited by considering only specific proteins and a small number of brain structures. They do not fully capture the comprehensive and intricate connections between plasma proteins and various aspects of brain structure, leaving significant gaps in the understanding of the underlying biological mechanisms. Whether the associations between proteins and measures of brain structure exhibit specific patterns? We hypothesise that by performing a more comprehensive analysis to uncover the details of the intricate relationship between proteomics and brain structure, it will be possible to unveil the underlying pathogenesis of a wide range of brain disorders and obtain important information regarding potential targets for therapeutic intervention. However, this scientific pursuit has been restricted by the lack of relevant data for a long time. The absence of such data, including genetic information, made it challenging to systematically explore the intricate relationships between plasma proteins and brain structure. The recent availability of large-scale datasets makes it feasible to establish robust links between the genetic regulation of proteins and that of brain structure, opening new avenues for research into the pathogenesis of neurodegenerative and neuropsychiatric disorders.

The database that has been analysed refers to 4997 participants in the UK Biobank (UKB) for whom plasma proteomic and neuroimaging data, and genetic data were available. From neuroimages, a total of 272 different MRI measures of brain structure across five categories were extracted, and the corresponding proteomic signatures were determined. Subsequently, enrichment analysis was performed to elucidate the functional significance of the proteomic signatures. Utilising genetic data, the bidirectional Mendelian randomisation (MR) analysis was performed to enhance the potential associations between proteins and brain structure, as well as between proteins and brain disorders. Finally, the potential mediating effect of brain structure on the association between proteins and brain health was investigated. Overall, we are dedicated to constructing a comprehensive atlas depicting the pattern of association between large-scale proteins and brain structure, and their implications in neurodegenerative and neuropsychiatric disorders.

## Results

A flow diagram of the study design is shown in Fig. 1. The present study included participants from UKB for whom both plasma proteomic and neuroimaging data are available. A total of 4900 healthy participants with a mean age of 63.0 were selected after excluding major medical conditions (Table 1 and Supplementary Data 1). The study used neuroimaging measures derived from the UKB, comprising 222 regional grey matter (GM) metrics assessing volume, thickness, and surface area across brain regions, alongside 54 white matter (WM) microstructure parameters that captured axonal integrity through fractional anisotropy (FA) and mean diffusivity (MD) measurements in major WM tracts (Supplementary Data 2). To enable Mendelian randomisation analysis between protein and brain structure, and between protein and disorders, the imputed genetic dataset released by UKB in July 2017 was used for a genome-wide association study (GWAS) of brain structure. The publicly available GWAS summary statistics of brain disorders were curated (Supplementary Data 3). Furthermore, a total of 3270 participants with a mean age of 54.2 were included in the mediation analysis, which demonstrated brain structure mediates the association between proteins and brain health (Supplementary Data 4).

### Plasma proteins associated with brain structure

The analysis began with an investigation of the association between 2920 plasma proteins and 272 brain structure measures in five categories in 4900 participants (2642 females and 2355 males). Only a few of the included patients were diagnosed with the brain diseases (Supplementary Data 5). Details of the inclusion and exclusion criteria and participant demography can be found in Table 1 and Supplementary Fig 1. Overall, 5358 significant associations were identified between 1143 proteins and 256 brain structure measures across the five categories, with multiple comparisons corrected using the false discovery rate (FDR) method ($P_{FDR} < 0.05$). Significant positive and negative associations were found in each category of brain structure measures. However, the proteins that exhibited the most significant associations differed between categories (Fig. 2A and Supplementary Data 6). In particular, for volume, NCAN showed the strongest association with the left rostral middle frontal cortex (standardised beta = 0.089, $P_{FDR} = 2.40 \times 10^{-11}$); for thickness, OXT showed the strongest association with the superior temporal cortex (standardised beta = −0.085, $P_{FDR} = 3.58 \times 10^{-5}$), and for surface area, MOG showed the strongest association with the rostral middle frontal cortex (standardised beta = 0.097, $P_{FDR} = 3.75 \times 10^{-15}$). For measures of the WM tracts, LEP showed the strongest association with FA in the right corticospinal tract (standardised beta = 0.105, $P_{FDR} = 7.61 \times 10^{-6}$) and MD in the right medial lemniscus tract (standardised beta = −0.142, $P_{FDR} = 9.38 \times 10^{-15}$).

From the view of brain structure measures, significant variations were observed in the number of associations among the different brain structure categories, specifically, the highest number of significant associations were observed for the WM MD measure (2501), followed by GM volume (1370), surface area (719) and thickness (682), and the least were found for WM FA (86) (Supplementary Data 6). In addition, substantial variations were observed for different brain regions and WM tracts within the same category (Fig. 2B). For the five categories, the highest number of associated proteins were with volume of left medial orbitofrontal, surface area of right pars orbitalis, thickness of left temporal pole, FA of right corticospinal tract and MD of right anterior thalamic radiation tract, respectively (see Supplementary Data 7). From the perspective of proteins, NCAN, SLITRK1, MOG, PTPRN2 and SEZ6L were the top five proteins that showed the highest total number of significant positive associations, and LEP, OXT, PAEP, CCN5 and XG were the top five proteins that showed the highest total number of significant negative associations (Fig. 2C). Furthermore, NCAN, LEP, PTPRN2, SLITRK1 and GFAP are the top five proteins with the highest number of significant associations for each structural category (Fig. 2D), and each was represented in at least two categories. The largest intersection between the proteins associated with the five structural categories was that between volume and MD, followed by volume and surface area (Fig. 2E). In sensitivity analyses, compared to

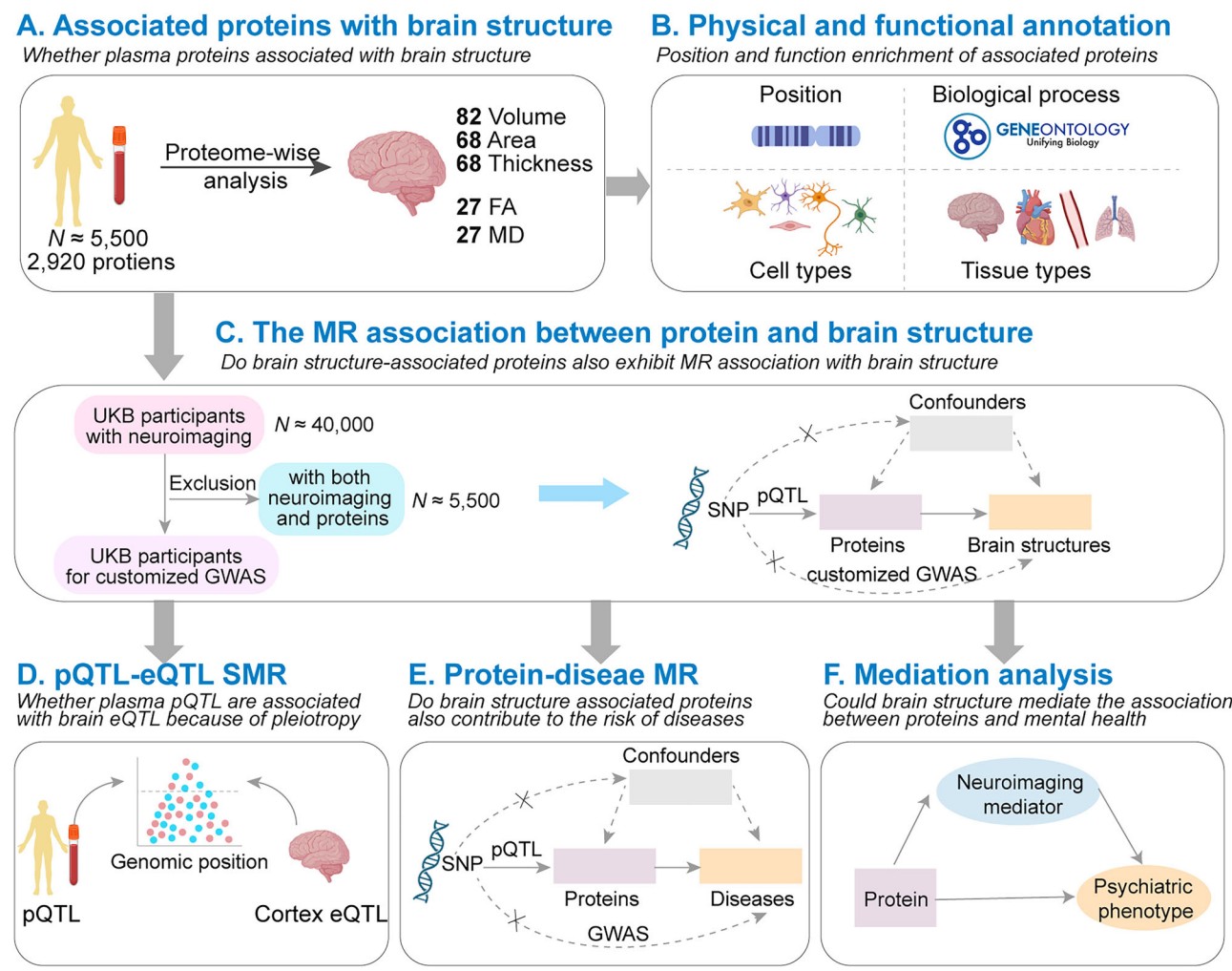

**Fig. 1 | Schematic overview of study design. A** The association between 2920 plasma proteins and measures of brain structure from five categories. **B** The annotation of preferred chromosome positions and functional categories for proteins associated with brain structure. Functional categories were annotated based on Gene Ontology (https://geneontology.org). **C** The protein-brain structure associations satisfied FDR-corrected $P < 0.05$ were further used for MR analysis that enhanced the relationships between plasma proteins and brain structures. Since the MR requires the cohort of exposure and outcome not to overlap, the GWAS of brain structure measures were conducted based on UKB participants for whom neuroimaging data was available but proteomic data was not. The proteins exhibited significant MR associations with brain structure measures (FDR-corrected $P < 0.05$) and were eligible for the following three sub-analyses, i.e., SMR, protein-disease MR and mediation analyses. **D** To explain the possible mechanism of why

plasma proteins and brain structure were connected, SMR analysis was performed to examine whether the pQTL of brain structure-related plasma proteins were associated with eQTL of brain cortex because of pleiotropy. **E** The MR analysis investigating whether proteins exhibiting significant association with brain structure also affect the risk of brain disorders. **F** The mediation analysis investigates whether the brain structure could mediate the association between plasma proteins associated with brain structure and mental health. The mediator and outcome were constructed as latent variables. FA fractional anisotropy, MD mean diffusivity, SNP single- nucleotide polymorphism, pQTL protein quantitative trait loci, GWAS genome-wide association study, eQTL expression quantitative trait loci, MR mendelian randomisation. Created in BioRender. Li, Z. (2025) https://BioRender.com/ghla3ez.

associations with only protein expression, age at imaging visit, sex, total intracranial volume and scanning site as the predictors, the inclusion of additional covariates didn't change the pattern of significant associations (Supplementary Data 8). In addition, the imputation procedure did not affect the pattern of significant associations between protein and brain structure (Supplementary Data 9). Furthermore, we found that the effects of protein expression were highly correlated between females and males, with a correlation coefficient of 0.88 for GM volume, 0.86 for cortical area, 0.89 for cortical thickness, 0.91 for WM FA and 0.85 for WM MD (Supplementary Data 10).

**Physical and functional annotation of brain structure associated proteins**

To investigate whether brain structure-associated proteins favour specific chromosomes and functional categories, the density of

discovered proteins in each chromosome and the functional enrichment of the proteins in each tissue type, brain cellular types and Gene Ontology (GO) biological processes (Fig. 3A) were examined. The brain structure associated proteins tend to be disproportionately located in chromosomes 19, 17 and 22 compared to the short length of these three chromosomes (Fig. 3B). In terms of function, the brain structure associated proteins did not exhibit any preference for specific UKB panels (Fig. 3C). With regard to GO enrichment, the proteins that exhibited significant positive and negative associations with brain structure in the five categories exhibited separate enrichment in immune and catabolic functions (Supplementary Fig. 2). Interestingly, when focused on a single category, obvious differences were observed in GO biological processes. In particular, for volume, the proteins that exhibited positive associations were enriched in GO processes relating to nervous system development and neurogenesis, while the proteins

**Table 1 | Demography of included UKB participants for association analysis**

| | Volume (N = 4908) | Area (N = 4997) | Thickness (N = 4997) | FA (N = 4632) | MD (N = 4632) | P |
|---|---|---|---|---|---|---|
| **Age at imaging visit (year)** | 63.0 (7.96) | 63.0 (7.96) | 63.0 (7.96) | 63.1 (7.92) | 63.1 (7.92) | 0.990 |
| **Age at baseline (year)** | 54.0 (7.79) | 54.1 (7.80) | 54.1 (7.80) | 54.0 (7.75) | 54.0 (7.75) | 0.990 |
| **Interval (year)** | 8.94 (1.76) | 8.93 (1.75) | 8.93 (1.75) | 9.01 (1.76) | 9.01 (1.76) | 0.047 |
| **Gender** | – | – | – | – | – | 0.982 |
| Female | 2603 (53.0%) | 2642 (52.9%) | 2642 (52.9%) | 2470 (53.3%) | 2470 (53.3%) | – |
| Male | 2305 (47.0%) | 2355 (47.1%) | 2355 (47.1%) | 2162 (46.7%) | 2162 (46.7%) | – |
| **eTIV (cm³)** | 1551 (153) | 1550 (152) | 1550 (152) | 1550 (153) | 1550 (153) | 0.997 |
| **SiteID** | – | – | – | – | – | 0.421 |
| Site1 | 1238 (25.2%) | 1264 (25.3%) | 1264 (25.3%) | 1202 (25.9%) | 1202 (25.9%) | – |
| Site2 | 785 (16.0%) | 785 (15.7%) | 785 (15.7%) | 781 (16.9%) | 781 (16.9%) | – |
| Site3 | 2885 (58.8%) | 2948 (59.0%) | 2948 (59.0%) | 2649 (57.2%) | 2649 (57.2%) | – |
| **Education (year)** | 17.4 (3.97) | 17.4 (3.98) | 17.4 (3.98) | 17.4 (3.97) | 17.4 (3.97) | 0.995 |
| **Townsed index** | −1.74 (2.82) | −1.74 (2.82) | −1.74 (2.82) | −1.74 (2.82) | −1.74 (2.82) | 1.000 |
| **Smoking** | – | – | – | – | – | 1.000 |
| Never | 3038 (61.9%) | 3093 (61.9%) | 3093 (61.9%) | 2880 (62.2%) | 2880 (62.2%) | – |
| Previous | 1679 (34.2%) | 1709 (34.2%) | 1709 (34.2%) | 1573 (34.0%) | 1573 (34.0%) | – |
| Current | 191 (3.89%) | 195 (3.90%) | 195 (3.90%) | 179 (3.86%) | 179 (3.86%) | – |
| **Drinking** | – | – | – | – | – | 1.000 |
| Never | 155 (3.16%) | 158 (3.16%) | 158 (3.16%) | 146 (3.15%) | 146 (3.15%) | – |
| Previous | 135 (2.75%) | 137 (2.74%) | 137 (2.74%) | 124 (2.68%) | 124 (2.68%) | – |
| Current | 4618 (94.1%) | 4702 (94.1%) | 4702 (94.1%) | 4362 (94.2%) | 4362 (94.2%) | – |
| **Ethnicity** | – | – | – | – | – | 1.000 |
| White | 4755 (96.9%) | 4842 (96.9%) | 4842 (96.9%) | 4489 (96.9%) | 4489 (96.9%) | – |
| Mixed | 34 (0.69%) | 34 (0.68%) | 34 (0.68%) | 33 (0.71%) | 33 (0.71%) | – |
| Asian or Asian British | 42 (0.86%) | 43 (0.86%) | 43 (0.86%) | 37 (0.80%) | 37 (0.80%) | – |
| Black or Black British | 36 (0.73%) | 37 (0.74%) | 37 (0.74%) | 34 (0.73%) | 34 (0.73%) | – |
| Chinese | 10 (0.20%) | 10 (0.20%) | 10 (0.20%) | 10 (0.22%) | 10 (0.22%) | – |
| Others | 31 (0.63%) | 31 (0.62%) | 31 (0.62%) | 29 (0.63%) | 29 (0.63%) | – |

The unadjusted *P*-values represent the statistical significance of overall differences in demographic characteristics across different brain structural metrics. Categorical variables were compared between groups using Fisher's exact test, and One-way ANOVA was used for continuous variables. All statistical tests were two-sided. *FA* fractional anisotropy, *MD* mean diffusivity.

that exhibited negative associations were enriched in GO processes related to immune function, and for MD, the proteins that exhibited positive associations were enriched in GO processes related to apoptosis, while the proteins that exhibited negative associations were enriched in GO processes related to morphogenesis. No significant enrichments in GO processes were observed for the other three brain structure categories (i.e., thickness, surface area and FA) (Fig. 3D).

In terms of tissue enrichment, the proteins that exhibited association with volume and surface area were enriched in up-regulated differential expression gene (DEG) of brain tissue and the proteins associated with MD were enriched in down-regulated DEG of brain tissues (Fig. 3E). In addition, separate analysis revealed that the tissue enrichment observed for volume and area was attributable to positively associated proteins, while the observed enrichment for MD was attributable to negatively associated proteins (Supplementary Fig 3 and 4). Given the significant enrichment of brain structure-associated proteins in brain tissue, consideration was given to whether the coding gene of these proteins prefers to be expressed in specific brain cell types. The volume-associated proteins were enriched in excitatory and inhibitory neurons, while the thickness-associated proteins were enriched in astrocytes and endothelial cell types (Fig. 3F).

### Associations between proteins and brain structure identified through MR analysis

To further enhance the observed extensive associations between proteins and brain structure measures, bidirectional MR analyses were performed for 5358 protein-brain structure pairs that exhibited significant associations. Utilising inverse-variance weighted (IVW) methods, the forward MR results revealed 33 significant MR relationships between 32 proteins and brain structure across the five categories (15 GM regions and 8 WM tracts) at FDR-corrected $p < 0.05$ ($p < 3.08 \times 10^{-4}$). Notably, the cingulate gyrus part of the cingulum, parahippocampal part of the cingulum and thalamic radiation were the tracts that exhibited the most MR associations with proteins. Of the 33 significant forward MR relationships, MD showed not only the highest number of significant MR relationships, but also the highest effect size of associations. The increased protein expression was found, through MR analysis, to be significantly associated with lower MD of WM tracts, with the most notable finding being the association between increased BTN2A1 protein expression and lower MD of the right parahippocampal part of the cingulum (beta = −0.05, 95% CI −0.08 to −0.03, $P_{FDR} < 0.001$). On the contrary, increased protein expression was associated with higher FA of WM tracts, among which the most significant one is that increased expression of REN protein was associated with higher FA of left anterior thalamic radiation (beta = 0.12, 95% CI 0.06 to 0.19, $P_{FDR} < 0.05$). (Fig. 4). When a less strict significance threshold of nominal $p < 0.001$ was applied, a total of 59 proteins, 25 GM regions and 16 WM tracts were involved in 71 potential MR relationships. Interestingly, more brain-enriched proteins, including NCAM1, OXT, and NPTXR, emerged as having a significant association with the measures of brain structure. Specifically, higher expression of NCAM1 and OXT proteins were associated with lower volume of the right precentral and lateral orbitofrontal cortex, and a

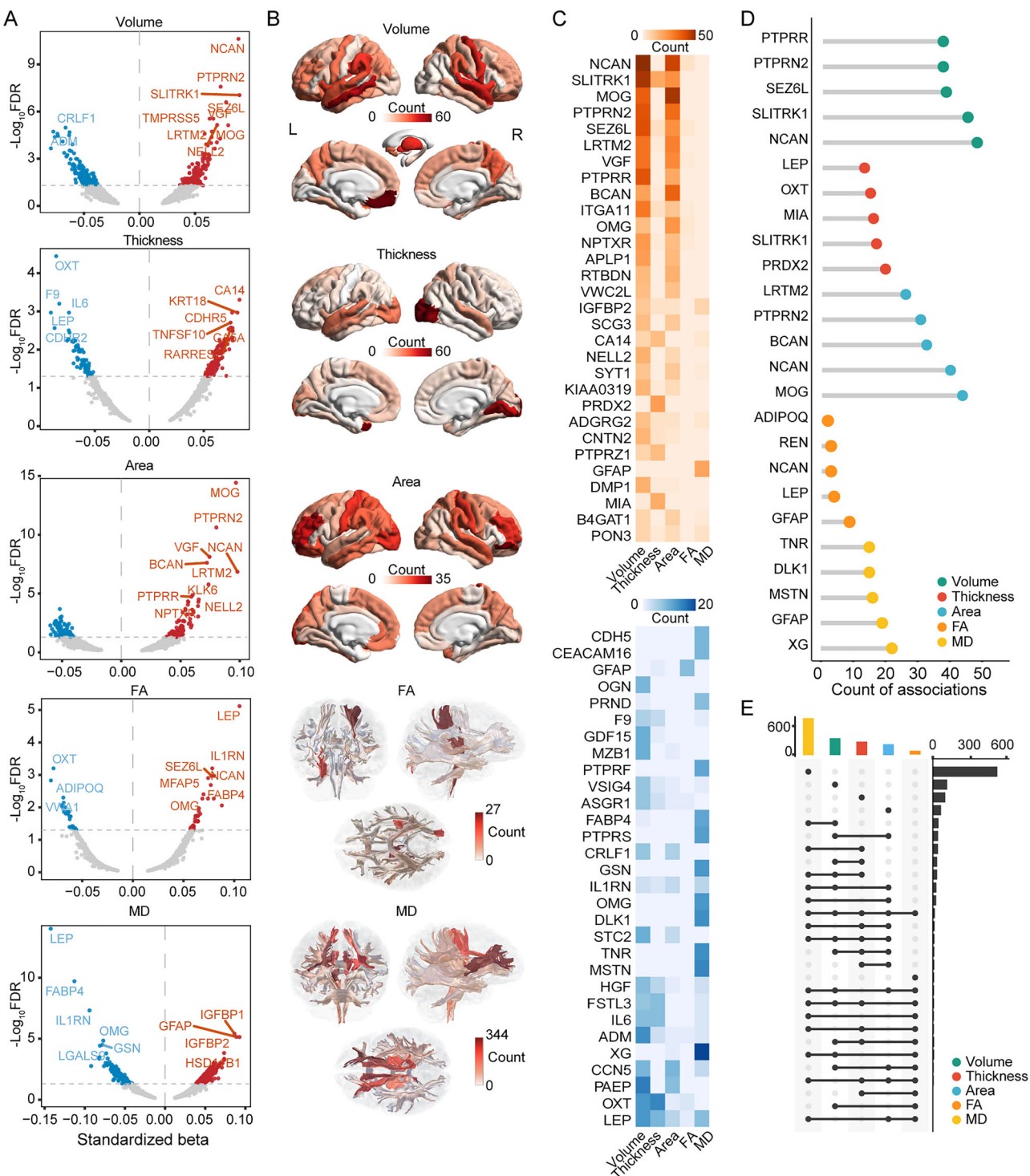

**Fig. 2 | Association between plasma proteins and brain structures. A** For each modality, each dot represents the most significant association between the specific protein and all corresponding metrics (i.e., regional or tract-wise) in Fig. 2B. Positive associations are shown in red and negative associations are shown in blue. Standardised coefficients are shown. **B** The distribution of the count of significantly associated proteins across different brain regions and tracts. Positive and negative associations are shown together. **C** The top 30 proteins with the highest total number of significantly positive (top panel) and negative (bottom panel) associations across modalities. **D** The top five proteins with the highest number of significant associations for each modality. **E** The UpSet plot showing the relationship of significantly associated proteins between different modalities. The counts of associations are shown in rows, and the categories of proteins are shown in columns. Source data are provided as a Source Data file. FA fractional anisotropy, MD mean diffusivity.

higher expression of NPTXR protein was associated with higher FA of left posterior thalamic radiation (Fig. 4). We found that most of the significant protein-brain structure remained significant when analysed using at least one of the other MR methods (Supplementary Data 11).

In sensitivity analysis, most of the MR analyses showed no evidence of directional pleiotropy (MR-Egger intercept) or horizontal pleiotropy (MR-PRESSO). For three out of five MR relationships exhibiting directional pleiotropy, in which the assumption of the instrumental variable was violated, the results from the other three MR

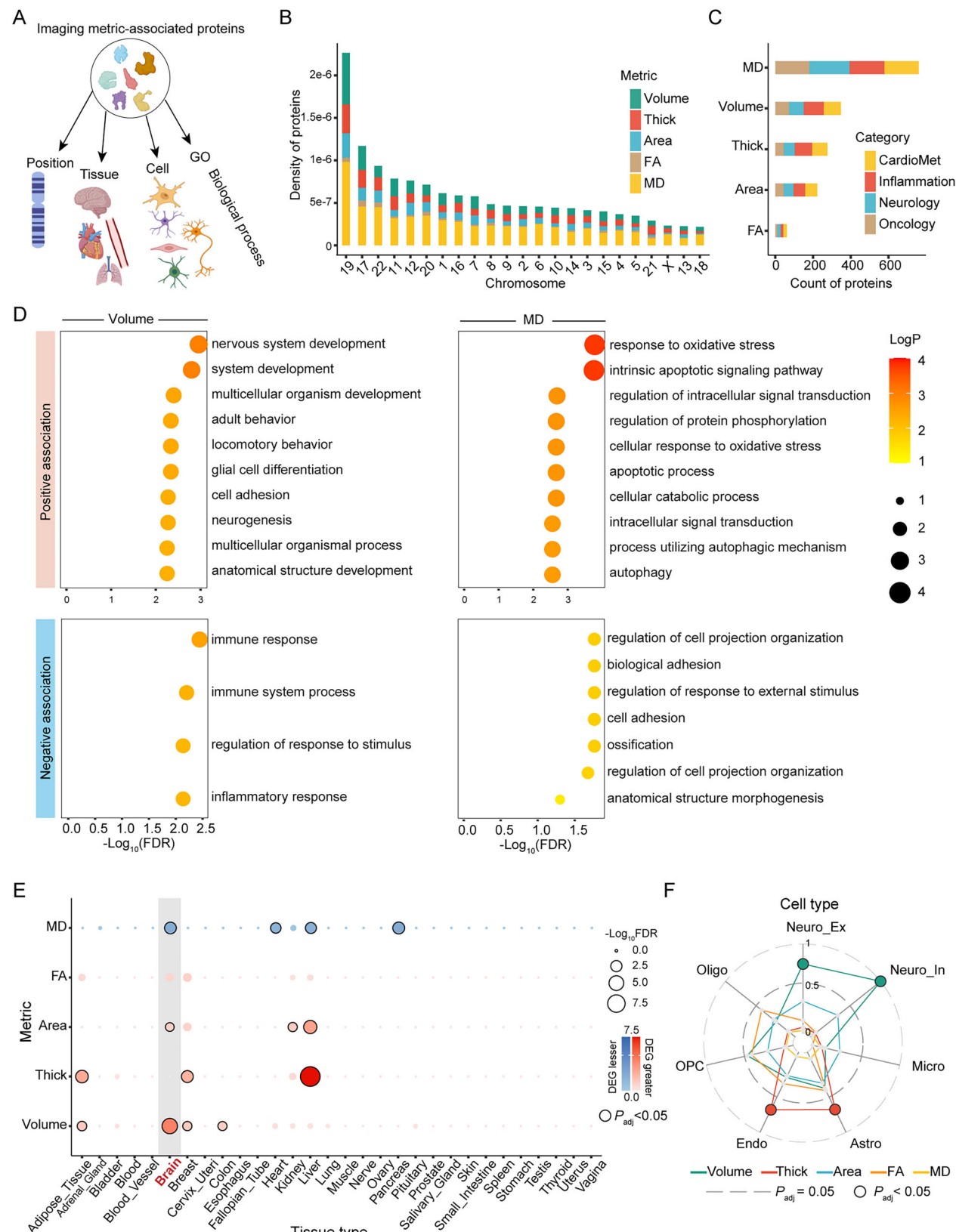

methods remained highly significant (Supplementary Data 11 and 12). To confirm the reliability of the MR relationships, the forward MR analyses were repeated with a stricter clumping at a $p$ threshold of $5 \times 10^{-8}$, the results of which remained significant and showed a consistent direction of effect with those using clumping at the $p$ threshold of $5 \times 10^{-6}$ (Supplementary Fig. 5 and Supplementary Data 13 and 14).

In reverse MR analyses, we identified 42 significant associations between brain structural measures (15 GM regions and 8 WM tracts) and 34 proteins that survived FDR correction $p < 0.05$ ($p < 3.75 \times 10^{-4}$). Of the significant reverse MR relationships, we observed 11 MR associations for GM volume (most significant, Left precentral-CA14, beta = 0.15, $p = 1.48 \times 10^{-5}$), 9 for GM area (most significant, Left

**Fig. 3 | Physical and functional annotation of brain structure associated proteins. A** The overview of the annotation categories, including protein-coding gene position, tissue enrichment, cellular enrichment and GO biological process enrichment. **B** The distribution of coding genes for brain structure association proteins across different chromosomes. The density of the associated proteins is shown, which accounts for the length of the chromosomes. **C** The distribution of brain structure association proteins across different UKB-defined protein categories. **D** The enrichment of coding genes for brain structure-associated proteins across GO biological processes. The positively and negatively associated proteins are shown separately, only volume and MD measures showed significant enrichment. **E** The enrichment of coding genes for brain structure-associated proteins across tissues. The GTEx v8 gene expression data and FUMA GENE2FUN were used.

The enrichment in brain tissue is highlighted in the grey bar. The significant enrichments after the FDR correction are shown with a black border. **F** The enrichment of brain structure-associated proteins across brain cell types. The cell type gene sets are obtained elsewhere, the significant enrichments are shown with coloured circles. For all the enrichment analyses, the statistical significance was determined with hypergeometric tests and the significance threshold was set to FDR-corrected $P < 0.05$. The UKB 2,920 proteins are used as a background. All statistical tests were two-sided. Source data are provided as a Source Data file. GO go ontology, OPC oligodendrocyte precursor cell, FA fractional anisotropy, MD mean diffusivity, DEG differential expression gene. Created in BioRender. Li, Z. (2025) https://BioRender.com/ghla3ez.

fusiform-MOG, beta = 0.21, $p = 3.77 \times 10^{-7}$), 7 for GM thickness (most significant, Right superiortemporal-TGFA, beta = $-0.16$, $p = 7.92 \times 10^{-5}$) and 15 for WM MD (most significant, forceps minor-KLK6, beta = 0.11, $p = 2.30 \times 10^{-7}$). No bidirectional MR relationships were found between proteins and brain structure measures (Supplementary Fig. 7 and Supplementary Data 15). We found that most of the significant brain structure-protein relationships remained significant when analysed using at least one of the other MR methods (Supplementary Data 15). In sensitivity analysis, only one MR relationship showed evidence of directional pleiotropy, for which the results from the other three MR methods remained highly significant (Supplementary Data 16). The reverse MR analyses repeated with a stricter clumping at the $p$ threshold of $5 \times 10^{-8}$ showed a significant and generally consistent direction of effect with those using clumping at the $p$ threshold of $5 \times 10^{-6}$ (Supplementary Data 17 and 18).

To further control the false positive rate in the presence of pleiotropic effects, we performed Causal Analysis Using Summary Effect Estimates (CAUSE) analysis to distinguish causality and genetic correlation. Almost no significant difference was found between the genetic correlation model and causal model at FDR-corrected $P < 0.05$. However, at a clumping $P$ threshold of $5 \times 10^{-6}$, 57 out of 71 protein-brain structure associations tended to be better characterised by causal model than genetic correlation models (i.e., delta_elpd < 0). Similarly, at a clumping $P$ threshold of $5 \times 10^{-8}$, 39 out of 59 protein-brain structure associations tended to be better characterised by a causal model than genetic correlation models (Supplementary Data 11 and 13).

### Pleiotropic association between plasma proteins and brain gene expression

Given the discovered MR relationship between 32 plasma proteins and brain structures, an intuitive question would be how these peripheral plasma proteins connect to the brain structural measures. One possible mechanism is that the peripheral plasma and brain share the same regulatory relationships. This possibility was investigated by first examining the gene expression of 32 proteins that were associated with brain structure in different tissues. While ENPP6, POXDL2, and HPCAL1 were highly enriched in brain tissues, some proteins were expressed in brain tissues without showing enrichment, and others were present only at very low expression (Fig. 5). Next, to test for pleiotropic association between the protein expression and gene expression. The Summary-based Mendelian Randomisation (SMR) analysis found that SULT1A1, REN, AGER, F11R, LEPR, and PAMR1 in the plasma exhibited pleiotropic association with the expression of their protein-coding gene in the brain, with $P_{SMR} < 0.05/32$ and $P$-value of heterogeneity in dependent instruments (HEIDI) > 0.05. In other words, these plasma proteins shared regulation mechanism with the brain.

### Associations between proteins and disorders identified through MR analysis

To investigate whether the proteins associated with brain structure also exhibited associations with the risk of disorders, the bidirectional

MR analyses were performed between the 32 proteins that exhibited significant MR associations with brain structure and ten central nervous system disorders, with a significance threshold at FDR correction $p < 0.05$ ($p < 3.40 \times 10^{-3}$). For the associations between protein and disease, a total of 21 significant protein-disease relationships were observed, corresponding to nine proteins and ten disorders, with the effect size expressed as odds ratio (OR). The strongest MR relationships were discovered between FKBP5 and amyotrophic lateral sclerosis (ALS; OR = 0.48, 95% CI = 0.30 to 0.78, $p = 2.81 \times 10^{-3}$) and between INHBC and ALS (OR = 1.32, 95% CI = 1.10 to 1.59, $p = 2.85 \times 10^{-3}$). Significant associations were found for all ten central nervous system disorders. Except in the case of attention deficit hyperactivity disorder, increased expression of BTN3A2 was associated with higher risk of Alzheimer's disease (AD; OR = 1.09, 95% CI = 1.06 to 1.12, $p = 2.04 \times 10^{-9}$), anxiety (OR = 1.03, 95% CI = 1.01 to 1.04, $p = 1.51 \times 10^{-5}$), bipolar disorder (BP; OR = 1.08, 95% CI = 1.05 to 1.12, $p = 3.27 \times 10^{-6}$), major depressive disorder (MDD; OR = 1.07, 95% CI = 1.05 to 1.09, $p = 2.66 \times 10^{-9}$) and schizophrenia (SCZ; OR = 1.23, 95% CI = 1.21 to 1.25, $p = 1.65 \times 10^{-93}$). On the contrary, increased levels of BTN2A1 proteins exhibited a protective effect on the risk of these disorders. BTN2A1 and BTN3A2 proteins showed the strongest associations with BP and SCZ. Moreover, the increased expression of ENPP6 was associated with a lower risk of Parkinson's disease (PD; OR = 0.86, 95% CI = 0.79 to 0.93, $p = 1.72 \times 10^{-4}$) and multiple sclerosis (MS; OR = 0.68, 95% CI = 0.56 to 0.82, $p = 4.82 \times 10^{-5}$). (Fig. 6 and Supplementary Data 19). We found that most of the significant protein-disease relationships remained significant when analysed using at least one of the other MR methods (Supplementary Data 19).

Sensitivity analysis revealed that most of the MR analyses showed no evidence of directional pleiotropy (MR-Egger intercept) or horizontal pleiotropy (MR-PRESSO). Furthermore, for four MR relationships exhibiting directional pleiotropy, which violated the assumptions of instrument variables for MR, the results from the other three MR methods (weighted mean, weighted and MR-Egger) remained highly significant (Supplementary Data 19 and 20). In order to confirm the reliability of the MR relationships that have been discovered, the MR analysis was repeated using a stricter clumping $p$ threshold of $5 \times 10^{-8}$, in which case 17 out of 21 MR relationships remained significant and showed consistent direction of effect (see Supplementary Fig. 6, Supplementary Data 21 and 22).

In reverse MR analyses, we identified 11 significant associations between seven brain disorders and nine proteins that survived FDR correction $p < 0.05$ ($p < 1.96 \times 10^{-3}$). Of the significant reverse MR relationships (Supplementary Fig. 8), the brain disorders demonstrated associations with the increased expression of six proteins (PAMR1, MAVS, F11R, REN, GER and LEPR) and decreased expression of three proteins (TNFRSF4, BTN2A1, ENPP6). The bidirectional relationships were only found between ENPP6 and MS, and between BTN2A1 and BP. We found that most of the significant disease-protein relationships remained significant when analysed using at least one of the other MR methods (Supplementary Data 23). In sensitivity analysis, No MR relationship showed evidence of directional pleiotropy

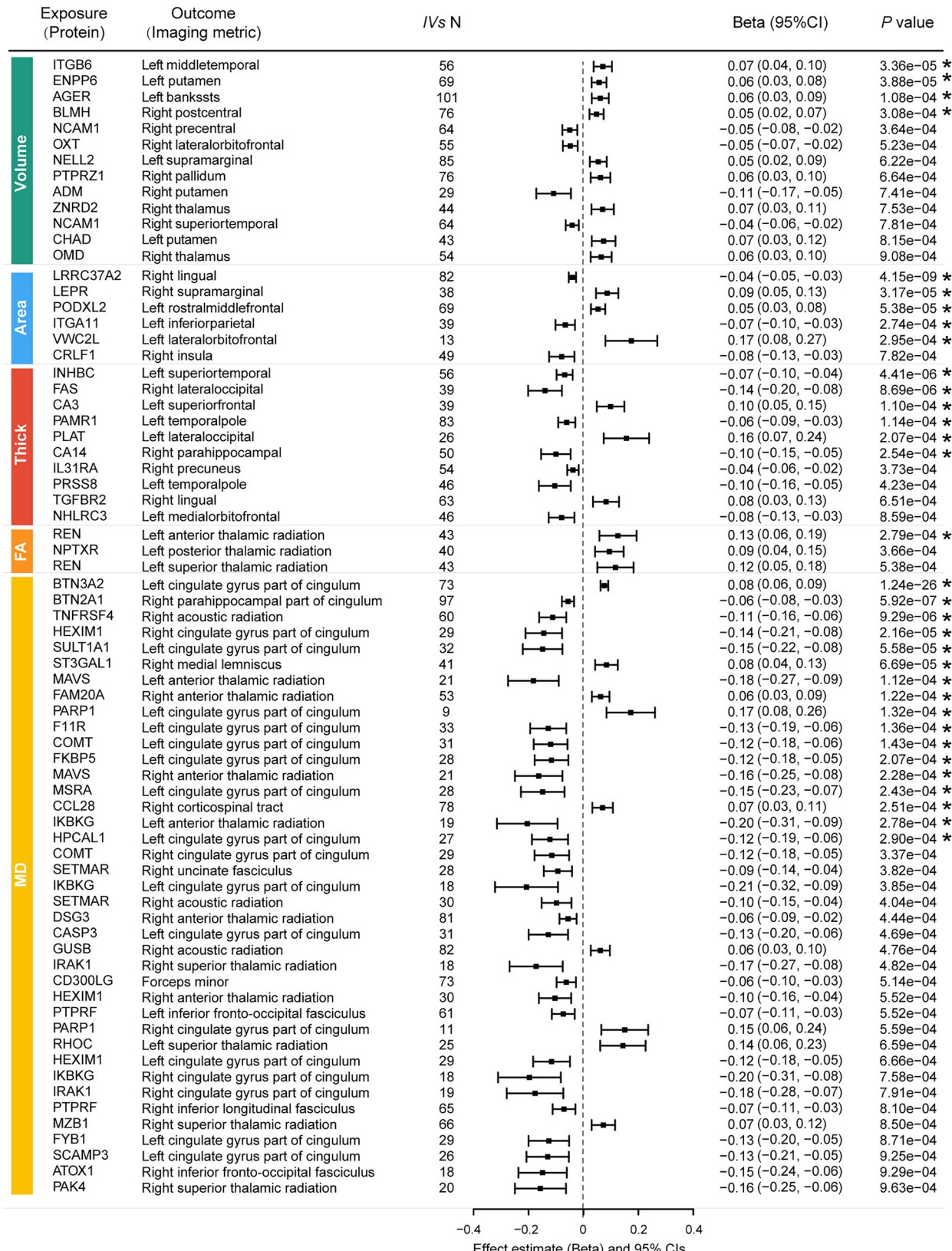

**Fig. 4 | The association between protein and brain structure in the forward MR.** The forest plot shows the significant MR relationships of the IVW method at a clumping *P* threshold of 5×10$^{-6}$. The significance was determined with a Wald test, and all MR results at a nominal *P* < 0.001 are shown. Raw *P*-values are shown in the rightmost column. The MR relationships that meet the significance threshold of FDR-corrected *P* < 0.05 are marked with an asterisk. The centre of the error bar means the estimated effect of the MR relationship using the IVW method. All statistical tests were two-sided. Source data are provided as a Source Data file. FA fractional anisotropy, MD mean diffusivity, IV independent variant.

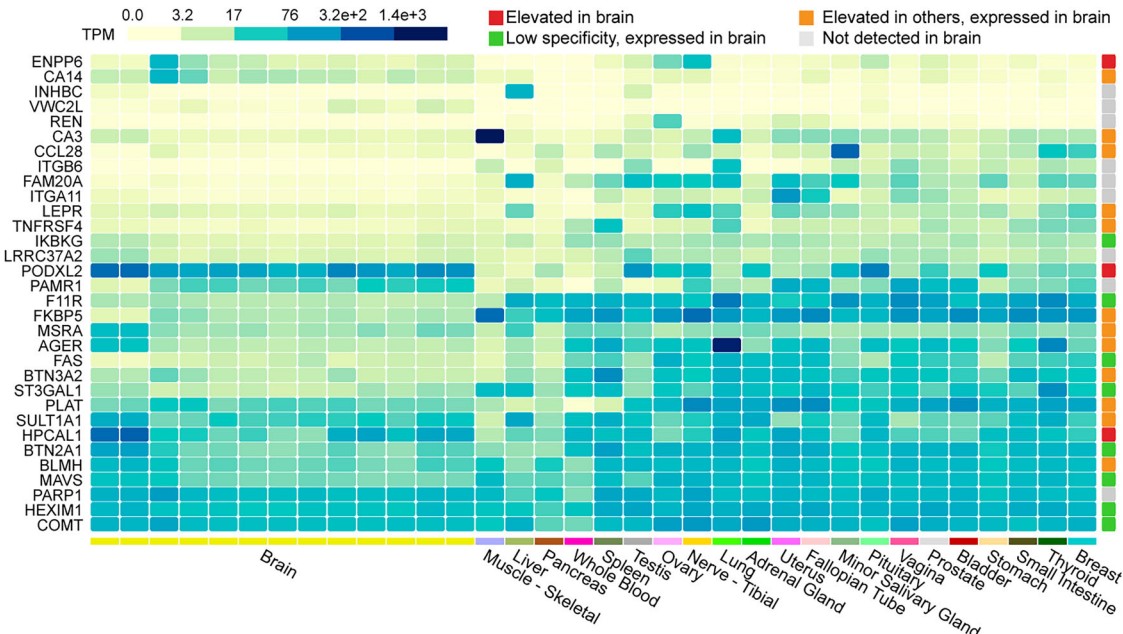

**Fig. 5 | Association between brain structure-associated proteins and gene expression of brain.** The expression of coding genes across tissues for proteins that exhibited significant MR associations with brain structure. Source data are provided as a Source Data file. TPM transcripts per million.

(Supplementary Data 24). The reverse MR analyses repeated with a stricter clumping at the $p$ threshold of $5 \times 10^{-8}$ showed generally consistent results with those using clumping at the $p$ threshold of $5 \times 10^{-6}$ (Supplementary Data 25 and 26).

To further control the false positive rate in the presence of pleiotropic effects, the CAUSE analysis found almost no significant difference between the genetic correlation model and causal model at FDR-corrected $P < 0.05$. However, at a clumping $P$ threshold of $5 \times 10^{-6}$, 16 out of 21 protein-disease associations tended to be better characterised by the causal model than the genetic correlation models (i.e., delta_elpd < 0). At a clumping $P$ threshold of $5 \times 10^{-8}$, 27 out of 34 protein-disease associations tended to be better characterised by causal model than the genetic correlation models (Supplementary Data 19 and 21)

### Brain structure mediated association between protein and mental health

Since the plasma proteins were associated with the brain structures and the brain structure-mental health connections were already well-documented, we further asked whether the brain structure could mediate the association between protein and mental health. A total of 3270 participants with a mean age of 54.2 were eligible for inclusion in mediation analysis (Supplementary Data 4). The proteins that exhibited significant MR associations with brain structure at FDR $p < 0.05$ were considered, and by using a structural equation model (SEM) model, we constructed latent mediators with protein-associated brain structures and latent outcomes with mental health phenotypes[18]. A total of four significant mediation relationships were identified, including one full mediation and three partial mediations (FDR-corrected $p < 0.05$). In particular, the regional volume of right precentral and left putamen could fully mediate 65% of the negative association between ENPP6 protein expression and latent mental health phenotype. Furthermore, the surface area of the right supramarginal mediated 16% of the negative association between ITGA11 protein expression and latent mental health phenotype. For all four mediation relationships, depression and anxiety were the observable variables contributing most to the latent mental health phenotype (Fig. 7 and Supplementary Data 27). For exploratory purposes, a mediation analysis was conducted for those proteins that exhibited significant

protein-neuroimaging MR relationships at a less strict significance threshold of nominal $P < 0.001$. An additional six mediation relationships were revealed, among which half (NCAM1, OMD, and PTPRZ1) were full mediations. In particular, the regional volume of the left supramarginal gyrus and right thalamus could fully mediate 63% of the negative relationship between NCAM1 protein expression and mental health, as well as 53% of the negative relationship between OMD protein expression and mental health. The regional volume of the right precentral gyrus and basal ganglia could fully mediate 26% of the negative relationship between PTPRZ1 protein and mental health. Furthermore, the regional volume of right medial orbitofrontal and left middle temporal mediates the positive association between OXT protein expression and mental health phenotype (Fig. 7 and Supplementary Data 27). Interestingly, ENPP6, NCAM1, NELL2, PTPRZ1, and OXT proteins were found to be highly enriched in brain tissues (Supplementary Fig. 9).

## Discussion

This study of the UK Biobank database has provided comprehensive insights regarding the plasma proteomic signature of brain structure. Altogether, 256 out of 272 brain structure measures from all five categories demonstrated significant associations with plasma proteins. The MD of WM tracts not only exhibited the highest number of associated proteins but also the highest strength of association. The highest number of common proteins was shared by the MD of WM tracts and volume measures. NCAN, SLITRK1, and MOG had the highest number of positive associations, while LEP, OXT, and PAEP had the highest number of negative associations. Functional enrichment analysis revealed that the proteins associated with brain structure were enriched in neurogenesis, brain tissues, and neuronal cell types, as well as immune and apoptotic-related processes. The proteins exhibited significant associations with brain structure measures also demonstrated associations with the risk of neurodegenerative and psychiatric disorders. Moreover, significant associations were discovered between identified proteins and mental health phenotypes, which could be mediated by protein-related brain structural measures. Collectively, the findings form a comprehensive atlas delineating the association pattern between large-scale proteins and brain structure, as well as their implications in neurodegenerative and neuropsychiatric disorders.

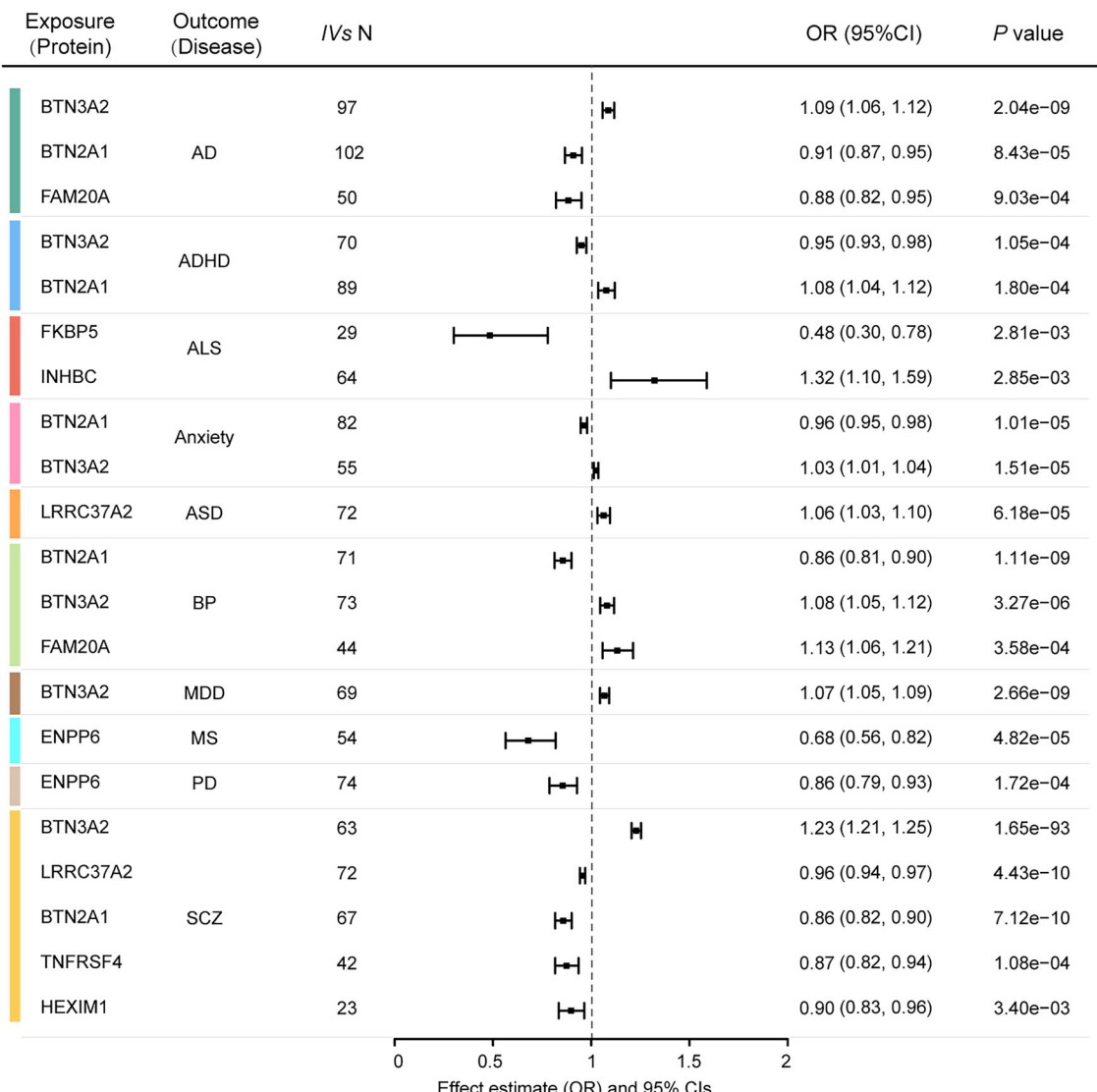

**Fig. 6 | The association between protein and disease in the forward MR.** The forest plot shows the significant MR relationships of the IVW method at a clumping *P* threshold of $5 \times 10^{-6}$. Raw *P*-values are shown in the rightmost column. The significance was determined with a Wald test, and the MR relationships meet the significance threshold of FDR-corrected *P* < 0.05 are shown. The centre of the error bar means the estimated effect of the MR relationship using the IVW method. All statistical tests were two-sided. Source data are provided as a Source Data file. ADHD attention deficit hyperactivity disorder, ASD autism spectrum disorder, BP bipolar disorder, MDD major depressive disorder, SCZ schizophrenia, AD Alzheimer's disease, PD Parkinson's disease, MS multiple sclerosis, ALS amyotrophic lateral sclerosis, IV independent variant, OR odd ratio, CI confidence interval.

Brain structure and its individual variance manifest a pivotal role in brain disorders. Nevertheless, only a few studies have attempted to find a potential link between plasma protein expression and brain structure, among which are reports that neurology-related proteins are associated with brain and total GM volume in normal aging[17] and the MR associations have been reported between 11 blood proteins and volumes of subcortical brain structures[16]. Instead of limiting the association and MR analysis to a narrow scope of proteins or brain structure measures, a more comprehensive atlas of association patterns between proteins and brain structure has been performed in the present study, which has the potential to reveal insights that are both multi-scale and multi-faceted. Accordingly, much more extensive associations and MR relationships between plasma proteins and brain structure were observed in our study. Altogether, 5358 associations were observed between 1143 proteins and 79 GM volume, 65 GM surface area, 65 GM thickness, 20 FA of WM tracts, and 27 MD of WM tracts. MD of WM tracts exhibited the largest number of associations and MR relationships, followed by regional GM volume and surface

area. Though MD and FA are both metrics that reflect the microstructural integrity of white matter, MD demonstrated the most significant associations with proteins, whereas FA showed the least significant associations. The higher number of associations between MD and CSF proteins might stem from the fact that CSF proteins reflect early pathophysiological changes in the brain, to which MD is sensitive, especially in the early stages of neurological diseases such as AD[19]. In patients with MS, MD was more strongly associated with widespread brain degeneration than FA as well[20]. FA is primarily influenced by directional water diffusion and is more specific to structural coherence and fibre integrity[21]. This specificity may make FA less sensitive to diffuse or generalised changes and more reliant on pronounced structural disruptions.

NCAN, SLITRK1, MOG, PTPRN2, and SEZ6L were the top proteins that showed the highest total number of significant positive associations with brain structures. The links between these proteins and brain disorders have been reported in previous studies. NCAN, an extracellular matrix protein, is closely related to the development, neuronal

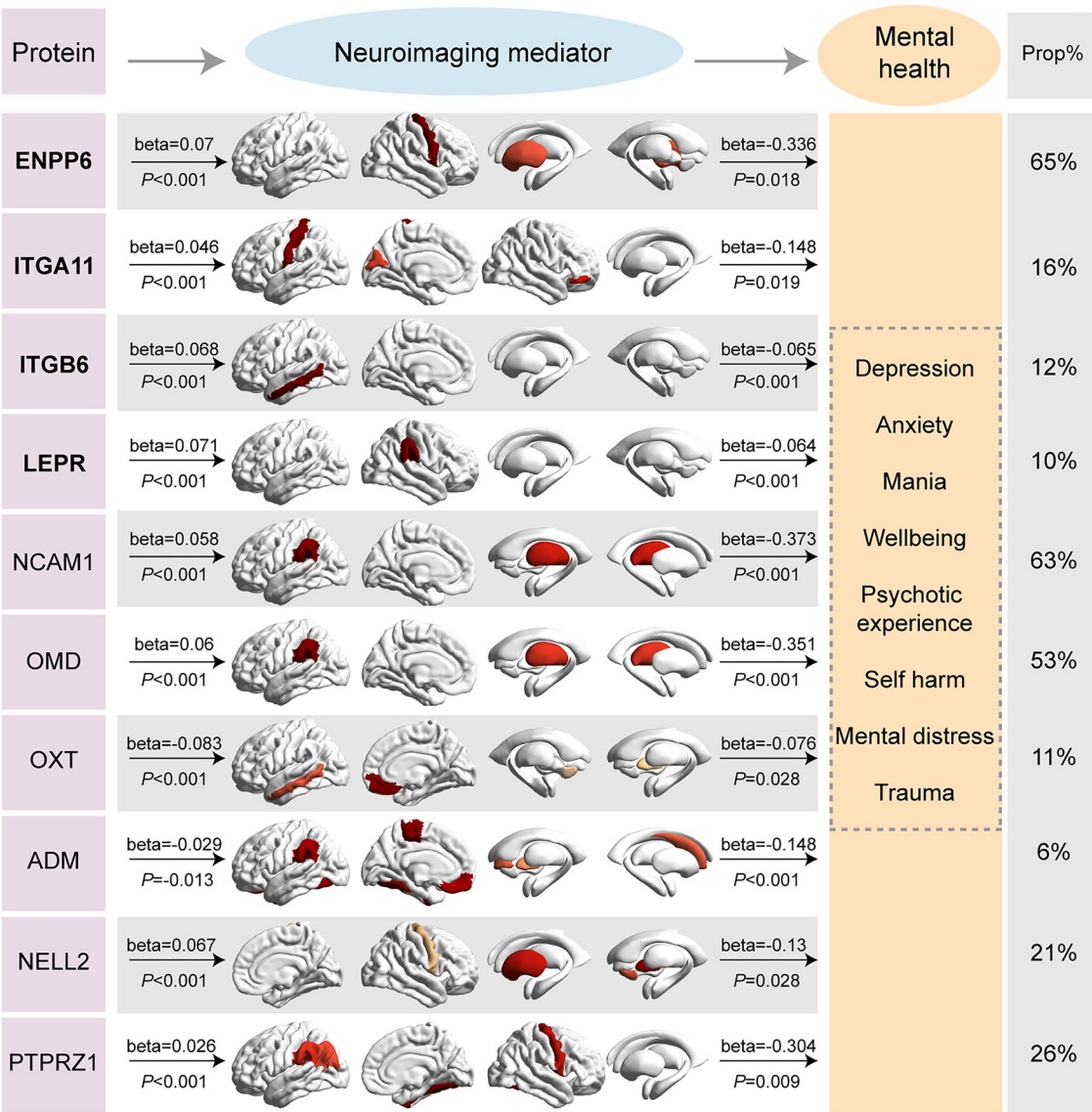

**Fig. 7 | The mediation effect of brain structure on association between protein and mental health.** The SEM model was established for assessing the mediation effect, and the significance of the paths were examined with bootstrapping and $t$ test. The observable variables are shown with rectangles, and the latent variables are shown with ellipses in the top row. Standardised coefficients of path a and path b, as well as the corresponding significance values, are shown. The proportion mediated by brain structure is shown on the rightmost column. The potential high collinearity of observable variables in constructing the latent variables is addressed with an iterative selection strategy. As an exploratory analysis, all proteins that exhibited MR associations with brain structure at nominal $P < 0.001$ were included. The proteins exhibiting significant MR association with brain structure at FDR-corrected $P < 0.05$ were highlighted in bold. All statistical tests were two-sided. Source data are provided as a Source Data file. Prop proportion.

senescence, and apoptosis of neurons. NCAN has been found to be associated with neuropsychiatric and neurodegenerative diseases[22]. In addition, NCAN demonstrated a predominant correlation with AD pathology[23]. Existing evidence has linked the plasma expression of NCAN protein to total brain volume[17]. The findings of the present study have expanded the significant associations of NCAN protein to the volume of 50 regions, surface area of 43 regions, FA of three tracts, and MD of five tracts. Apart from NCAN, the other proteins are also relevant to the central neural system and brain diseases. SLITRK1 function is associated with the control of neurite development and synaptogenesis[24]. SLITRK has been linked to Tourette's syndrome, trichotillomania, and obsessive-compulsive disorder[25]. MOG, expressed in oligodendrocytes in the central nervous system, is an encephalitogenic protein that can trigger a demyelinating immune response[26]. Moreover, PTPRN2 has been implicated in AD and FTD[23], while SEZ6L has been associated with BP[27]. The proteins with

significant negative associations with brain structure include LEP, OXT, PAEP, CCN5, and XG. LEP is well-known for its role in metabolic regulation and has been implicated in neurodegenerative diseases, potentially linking metabolic dysfunction to brain pathology[28]. OXT is associated with social behaviours and stress responses[29]. It has also been implicated in anxiety, depression, and AD[30], although its relationship with brain structure remains unclear. Previous studies have highlighted associations between these proteins and CNS or brain disorders, and our findings further reveal their associations with brain structure. While the roles of PAEP, CCN5, and XG in brain diseases are not yet fully understood, our study identifies significant associations between these proteins and brain structure, offering a valuable basis for future studies.

Thirty-three significant MR relationships between plasma proteins and brain structures were revealed through MR analysis. The strongest associations with proteins were observed in the cingulate gyrus part of

the cingulum, parahippocampal part of the cingulum, and thalamic radiation. The cingulate gyrus, part of the cingulum, is involved in emotional regulation, attention, and memory, and disruptions in this tract have been implicated in conditions such as AD and major depressive disorder[31]. The parahippocampal cingulum is closely associated with episodic memory and spatial navigation, and alterations in this tract are also observed in the early stages of AD[31]. Our results highlight the potential involvement of the cingulum in the pathophysiology of neurodegenerative and psychiatric disorders through associations with specific proteins. The thalamic radiation serves as a critical pathway for relaying sensory and motor signals between the thalamus and cortical regions. Studies have demonstrated that alteration in the anterior thalamic radiation integrity is linked to SCZ and depression[32,33]. Taken together, these findings suggest that the associated proteins may play key roles in maintaining the structural integrity and functional regulation of these tracts, thereby contributing to the pathophysiology of neurodegenerative and psychiatric disorders.

Our analysis revealed that brain structure-associated proteins are disproportionately localised to chromosomes 19, 17, and 22, relative to the relatively short length of these chromosomes. The expression of brain structure-associated proteins on chromosomes 19, 17, and 22 is particularly noteworthy given the established relevance of these chromosomes to brain health and disease. Chromosome 19 has a high gene density and contains key genetic factors related to brain aging and neurodegenerative diseases, including *APOE*, which is strongly linked to AD[34]. Chromosome 17 includes *MAPT*, a gene associated with tau-related pathologies, such as FTD[35]. Chromosome 22 has been implicated in neurodevelopmental and psychiatric disorders, with alterations in this region contributing to schizophrenia and 22q11.2 deletion syndrome[36,37]. Our findings show that many of the proteins identified on these chromosomes, including those encoded on chromosome 17 (OMG[38], BAIAP2[39], GFAP[40]), chromosome 19 (NCAN[23], APLP1[41], GDF15[42]), and chromosome 22 (NPTXR[43], OSM[44], RTN4R[45]) are strongly associated with brain structure and have been previously implicated in various neurological and psychiatric disorders. These results suggest that these chromosomes may serve as hotspots for genes influencing brain morphology and function, reflecting their broader role in neurodevelopmental and disease pathways.

From the point of view of function, the proteins which showed an association with the five measures of brain structure were enriched in relatively broad functions, including immune and catabolic processes. On the other hand, the proteins associated with volume and MD showed a more specific functional relationship. As volume and MD are negatively correlated[46,47], the positively associated proteins should be compared to the negatively associated proteins, and the same is true for the negatively associated proteins. Specifically, the proteins that were positively associated with volume and negatively associated with MD were both enriched in synthesis functions, such as nervous system development, neurogenesis and cell projection organisation, whereas the proteins that were negatively associated with volume and positively associated with MD were enriched in clearance related immune, and apoptotic function, respectively. The immune and apoptotic functions are different forms that are relevant to cell death, among which apoptotic cell death could eliminate cells no longer needed without initiating an immune response[48]. These findings greatly extend knowledge regarding protein-brain structure relationships and highlight the potential functional preference of associated proteins for different brain structures.

With regard to the MR association between brain structure-associated plasma proteins and brain disorders, significant relationships were observed for all ten brain disorders. Among the nine proteins with significant MR associations, BTN3A2, a member of the immunoglobulin superfamily playing a role in T-cell responses within the adaptive immune response, exhibited a broad association with AD, anxiety, attention deficit hyperactivity disorder (ADHD), BP, MDD, and SCZ. This finding is supported by prior evidence that the genetic variants of *BTN3A2* showed pleiotropic effects on SCZ, BP, and MDD[49], as well as that increased expression of BTN3A2 might confer risk for SCZ by altering excitatory synaptic function[50]. The findings of the present study extend knowledge of the role of BTN3A2 to a broader range of disorders. On the contrary, we found that BTN2A1 was associated with decreased risk of AD, anxiety, and SCZ, which is consistent with a report that BTN2A1 is involved in the immunomodulation of the activity of γδ T-cells and has significant anti-neuroinflammatory and neuroprotective effects[51]. Although BTN3A2 and BTN2A1 are known to interact[52], given that BTN3A2 and BTN2A1 were separately identified as risk and protective factors for brain disorders in the present study, further studies are needed to elucidate the underlying mechanisms of how BTN3A2 and BTN2A1 proteins are oppositely involved in brain disorders. Furthermore, the ENPP6 protein exhibited a significant association with the risk of PD. In PD, the putamen, which is part of the basal ganglia network that is crucial for motor control, is heavily impacted[53], and this is potentially related to the observed association between ENPP6 and the volume of left putamen. Altogether, although the exact mechanism underlying the protein-disease relationships remains to be elucidated, the findings of the present study do emphasise the potential of brain-associated proteins as novel risk factors of brain disorders. Given the smaller sample sizes for anxiety and ALS compared to other disorders, the non-significant MR-relationships may be potentially due to the impact of sample size rather than indicating a lack of significance. Therefore, further studies based on a larger case count are warranted to validate our findings.

Now that proteins manifest an effect on both brain structure and brain disorders, elucidating the intricate relationship of proteins, brain structure, and brain disorders is of major clinical importance. However, due to the limited number of patients with both proteomic and neuroimaging data in UKB, direct evaluation of the mediation relationships corresponding to particular brain disorders was not possible. Instead, the information on mental health phenotypes available in UKB was used to investigate whether the protein-associated brain structure measures could serve as potential mediators of the association between proteins and mental health. A total of ten proteins were found to be associated with mental health, that was mediated by brain structure. Three of the ten proteins were highly enriched in brain tissues, namely ENPP6, NCAM1, and OXT, and the mediators with high loadings, including supramarginal cortex, thalamus, putamen, precentral cortex, OFC, were brain regions frequently involved in depression and anxiety[54,55]. In particular, a strong mediating effect of the precentral cortex and putamen was observed on the relationship between ENPP6 and mental health, with a mediation proportion of 65%. The precentral cortex, primarily involved in motor control and coordination, has been increasingly recognised for its role in working memory and emotional regulation[56]. Cortical thinning in the precentral gyrus has been associated with an increased risk of depressive symptomatology[57]. In addition, advanced control relevant activation in the precentral gyrus has been linked with suicide risk in mood disorders[58]. The high mediation proportion suggests that ENPP6 may influence mental health indirectly by modulating the structure or function of these regions. NCAM1 plays a significant role in the development of the nervous system by regulating neurogenesis and neurite outgrowth, while OXT is a precursor protein that produces oxytocin and neurophysin I, which are transported axonally to the nerve endings and secreted into the bloodstream. The association between NCAM1 and mental health was mediated by the supramarginal gyrus and thalamus, whereas the association between OXT and mental health was mediated by the medial orbitofrontal, middle temporal cortex, accumbens, and pallidum. These findings are consistent

with the well-known blunted activation of basal ganglia and medial prefrontal cortex relating to reward in depression[55,59,60], as well as the disturbance of the large-scale cortico-striato-thalamo-cortical circuitry involved in anxiety[54]. Although several proteins, including BTN3A2, BTN2A1 and INHBC, did not exhibit significant mediation relationships for mental health phenotypes in UKB, given that BTN3A2 and BTN2A1 were associated with MD of different parts of the cingulum and INHBC exhibited an association with ALS-related superior temporal cortex, the possibility remains that certain brain structure is potential mediator for associations between BTN3A2, BTN2A1, INHBC and brain disorders in patients with confirmed diagnosis. In summary, our findings do pave the way for understanding the intricate interplay between microscale proteins, brain structure endophenotypes, and behaviour phenotypes.

Several potential limitations should be considered when interpreting the results of the present study. Firstly, although the study is of the largest cohort to have so far been investigated to characterise the association pattern between proteins and brain structures, further replication in independent cohorts and cohorts with different ethnicities is vital. Secondly, due to the acquisition schedule of the UK Biobank, there is a time gap between plasma collection and neuroimaging scanning. Given that the plasma proteome could change with age, this may induce potential bias[61,62]. The interval has been accounted for as a covariate. However, future study designs with more synchronised data collection could address this concern. Thirdly, although this research has utilised comprehensive proteomic databases, it is still far from measuring all circulating proteins. Future advances in proteomic measurement technologies will increase the possibility of discovering novel linkages. Fourthly, the included participants who underwent both the imaging visit and proteomic analysis exhibited slight differences in age and sex compared to the other UKB participants. Although we have partially controlled the effects of age and sex through regression, validation with future releases of proteome-neuroimaging data covering a broader range could strengthen our findings. Fifthly, as the proteins are highly correlated with one another, and the brain measures also exhibit correlations, the significance of protein-MR associations presented little inflation. Although we have performed multiple comparison corrections and validated the results with MR and mediation analyses, future independent validation could enhance the discovered associations between proteins and brain structures. Sixthly, the plasma proteomic profile in younger individuals may differ from that observed in our cohort. The disease-associated proteins identified in this study may reflect later-stage plasma protein markers, rather than early biomarkers, especially considering that disorders such as autism spectrum disorder (ASD), ADHD, schizophrenia, BPD, and anxiety typically have onset in childhood or early adulthood. Consequently, future validation is needed to identify if the disease-associated protein changes are longitudinally predictive of incident diseases.

In conclusion, utilising the unique opportunity provided by large-scale genetic, proteomic, and neuroimaging data, this study represents a comprehensive atlas of the proteomic signature of human brain structure. Our findings underscore the widespread associations between plasma proteins and structural changes in the brain, revealing how plasma proteins may reflect alterations in the central nervous system. The identified proteins exhibited associations with a wide range of neurodegenerative and neuropsychological disorders, highlighting the potential of plasma proteins as critical biomarkers for assessing brain health. Furthermore, brain structure was found to play a mediating role in the relationship between plasma proteins and brain health. By uncovering these associations, this research bridges the gap between plasma protein markers, brain structure, and brain disorders, offering an opportunity to unravel the mechanisms underlying brain disorders and to identify potential therapeutic targets.

## Methods

### Participants

The UK Biobank is a large-scale population-based cohort of approximately 500,000 participants aged 40 to 69 years (https://biobank. ndph.ox.ac.uk/showcase/), who provided fully informed written consent for the collection of genomic, whole-body imaging, electronic health record linkage, body fluid biomarker, and physical and anthropometric measurements[63]. UKB received approval from the National Information Governance Board for Health and Social Care and the National Health Service North West Centre for Research Ethics Committee (Ref: 11/NW/0382). In particular, approximately 40,000 participants received neuroimaging scanning, and a cohort of 54,265 participants for whom plasma samples were collected at the commencement of the study. The present study refers to the participants in UKB for whom both plasma proteomic and neuroimaging data are available, and excludes participants that present obvious brain structural abnormalities, including all cause dementia, stroke, Parkinson's disease, parkinsonism, Huntington's disease, multiple sclerosis, epilepsy, intracranial injury, cerebrovascular diseases, encephalitis, myelitis, encephalomyelitis, bacterial meningitis, intracranial and intraspinal abscess and granuloma, intracranial and intraspinal phlebitis and thrombophlebitis, cerebral infarction, vascular syndromes of brain in cerebrovascular diseases, other demyelinating diseases of central nervous system, congenital malformations of the nervous system, malignant neoplasm of brain, intracranial laceration and haemorrhage due to birth injury or other degenerative disorders of nervous system (Supplementary Data 1). Finally, approximately 4997 participants were used in our primary association analysis (see Table 1 and Supplementary Fig. 1).

### Protein measurements in plasma

The plasma samples were processed using a NovaSeq 6000 Sequencing System (Illumina Inc., San Diego, USA), and the concentrations of 2923 unique proteins were measured using Olink Explore 3072 with eight panels (Olink Proteomics AB, Uppsala, Sweden). Subsequently, quality control, outlier detection, and normalisation are applied to produce Normalised Protein eXpression (NPX) values in relative units on a log2 scale for each protein and each participant[64]. Proteins with a missing rate of 30% or higher were excluded (GLIPR1, NPM1, and PCOLCE), and the remaining missing values were imputed based on median values. All remaining 2920 proteins showed a missing rate of less than 25%, of which 2911 proteins actually showed a missing rate of less than 20%, and 1215 proteins had a missing rate of less than 5%. The value of the expression of each protein was inverse-rank normalised. The effects of sex and age were regressed out from protein expression before the protein-brain structure association analysis because they exhibited high collinearity with the protein. In this way, we ruled out the possibility that the observed protein-brain structure associations were simply the effect of age and sex. Finally, the NPX values of 2920 proteins were available for further analysis.

### Imaging derived phenotypes

Given that grey and white matter are crucial components that serve distinct functions for overall brain health, here we included brain structural imaging derived phenotypes (IDPs) that were previously made available by the UKB neuroimaging group[63]. Similar to previous researches, regional grey matter measures (volume, thickness, and area) derived from T1-weighted images and microstructural measures derived from diffusion weighted imaging, including FA and MD, were chosen to be used in the present study[65]. Specifically, 86 GM volume, 68 GM surface area, and 68 cortical GM thickness measurements were computed for brain regions corresponding to the Desikan-Killiany atlas by using FreeSurfer v6.0.0 (https://surfer.nmr.mgh.harvard.edu), and 27 FA and 27 MD values were available for 27 major WM tracts in

UKB (Supplementary Data 2). Totally, 218 GM measurements of regions and 54 WM measurements of tracts were used. Measurements of the whole brain were only included as covariates. All the IDP data were inverse-rank normalised before statistical analyses.

## Genetic data processing and analysis

With regard to genetics, the imputed genetic dataset released by UKB in July 2017 was used for MR analysis in the present study. The samples and information extracted regarding genotype were subject to strict quality control[66]. Specifically, single-nucleotide polymorphisms (SNPs) with a missing rate > 5%, minimum minor allele frequency < 0.1%, or Hardy–Weinberg equilibrium test $P < 1 \times 10^{-10}$ had been excluded[18,67]. To reduce the difference in population structure, all analyses were based on unrelated individuals. The unrelated individuals were selected with the following criteria: (1) used to compute principal components, (2) not identified as outliers for heterozygosity and missing rates, (3) without putative sex chromosome aneuploidy, (4) with no more than ten putative third-degree relatives. (5) have a genetic background of 'White British'. Since the cohort for the publicly available GWAS summary of IDPs overlaps with the cohort used for proteomic analysis, to satisfy the no-overlapping assumption of the cohort for two-sample MR analysis of the relationship between plasma proteins and IDPs, we performed GWAS analysis based on 25,576 UKB participants for whom neuroimaging data was available but proteomic data was not. A linear association test was then performed for each IDP with PLINK 2[68], with age at imaging visit, sex, education years, scanning site, total intracranial volume, and 40 genetic principal components provided by UKB as covariates. All IDPs were quantile normalised before the GWAS analysis was performed.

## GWAS summary of brain disorders

To conduct Mendelian randomisation analysis between protein and disorders, we curated GWAS summary statistics of brain disorders. Given their significance in public health and the relatively large number of cases, GWAS summary statistics were collected from publicly available databases for six psychiatric disorders (ADHD (cases/total = 19,099/53,293)[69], anxiety (cases/total = 7016/18,186)[70], ASD (cases/total=18,381/46,350)[71], BP (cases/total = 20,352/51,710)[72], MDD (cases/total = 45,396/142,646)[73], SCZ (cases/total=67,390/161,405)[74]) and four neurodegenerative disorders[75] (AD (cases/total=10,520/401,661), PD (cases/total=4,681/407,500), MS (cases/total = 2409/408,561) and ALS (cases/total = 531/184,000)). In particular, the psychiatric disorders GWAS summary was downloaded from the Psychiatric Genomics Consortium (https://www.med.unc.edu/pgc/download-results), and the degenerative disease GWAS summary was downloaded from FinnGen (https://www.finngen.fi/fi). All studies were of European ancestry and had no overlapping individuals with the UKB. A detailed summary can be found in Supplementary Data 3.

## Gene expression and quantitative trait loci

Here, we used the protein quantitative trait loci (pQTL) and gene expression quantitative trait loci (eQTL) summary data to characterise genetic variants of protein and gene expression separately. A "pQTL" stands for "protein quantitative trait locus," which refers to a genetic variant associated with the abundance of a specific protein, while an "eQTL" stands for "expression quantitative trait locus," indicating a genetic variant linked to the expression level of a gene (mRNA transcript) at a specific locus on the chromosome. The protein quantitative trait loci relationships were obtained from the previous UK Biobank Pharma Proteomics Project (http://ukb-ppp.gwas.eu), which provides comprehensive pQTL mapping of 2923 proteins and identifies 14,287 primary genetic associations, in addition to ancestry-specific pQTL mapping in non-Europeans. Consistent with our primary analysis, we restricted the pQTL mapping relationships to participants of European ancestry. Details

of data processing and genetic association analysis have been described elsewhere in previous work[64].

The gene expression data used to characterise the distribution of protein-coding genes were obtained for 34 different tissues from the database of the GTEx project v8 (https://www.gtexportal.org/home/)[76]. All variant-eQTL associations that were tested in brain cortex from GTEx v8, and in particular cis-eQTL, were used for pleiotropy association analysis between pQTL and eQTL.

## Statistical analyses

The present study contains six different research contents, three main analyses, and three sub-analyses. A flow diagram of the analyses that were performed is shown in Fig. 1. In the main analysis, the association analysis between proteins and brain structure measures was first performed, after which, the physical position and functions of the associated proteins were annotated. Furthermore, the association relationships were further enhanced with MR analysis. The proteins exhibited significant MR relationships at FDR-corrected $P < 0.05$ were eligible for the following three sub-analyses, i.e., SMR, protein-disease MR, and mediation analyses. The goal of these analyses was to investigate the potential mechanism by which the plasma proteins were associated with brain structure, whether brain structure-associated proteins would contribute to the risk of brain disorders, and whether brain structure could mediate the association between proteins that are associated with brain structures and mental health.

**Association analysis.** A generalised linear model was used to investigate the association between plasma protein expression and imaging derived measures of brain structure, including regional GM volume, area, and thickness, as well as FA and MD of WM tracts. Protein expression, age at imaging visit, sex, years of education, Townsend index, ethnicity, smoking and drinking status at imaging visit, the interval between baseline and imaging visit, scanning site, and total intracranial volume were used as predictors of the structural measures. The significance was determined by FDR correction across all protein-brain structure pairs. We conducted three sensitivity analyses. First, an additional association analysis with only protein expression, age at imaging visit, sex, total intracranial volume, and scanning site as the predictors was performed to explore the influence of the covariates. Second, we conducted an additional association analysis without performing an imputation of protein expression. Third, to examine potential sex differences, the association analysis was further conducted in female and male separately.

**Physical and functional annotation.** To demonstrate potential differences in associated proteins related to different structural measures, physical and multiple functional annotations were performed for brain structure-associated proteins, including chromosome distribution annotation, UKB panel enrichment, tissue enrichment, cellular enrichment, and biological process enrichment. The physical annotation was performed by first mapping the protein-coding genes to chromosome positions with a previously established *topr* R software[77]. Subsequently, the number of protein-coding genes on each chromosome was divided by the length of that chromosome, which generated the density of associated proteins in each chromosome that accounted for the length of the chromosome. UKB panel and cellular enrichment annotations were performed using over-representation analysis from Python scipy. The four UKB panels, namely, cardiometabolic, inflammation, neurology, and oncology, used to classify proteins were downloaded from the UKB website (https://biobank.ndph.ox.ac.uk/showcase/ukb/auxdata/olink_assay.dat). The marker genes of seven central nervous system cell types, including astrocytes, endothelial cells, microglia, excitatory neurons, inhibitory neurons, oligodendrocytes, and oligodendrocyte progenitor cells, were obtained from a prior study[78]. Tissue enrichment was performed with a

hypergeometric test from the FUMA GENE2FUNC module[79], using GTEx v8 as the source of gene expression. Specifically, DEG sets were pre-calculated by performing a two-sided *t* test for each of the tissue labels in comparison to all the others in turn. Genes with Bonferroni-corrected $p < 0.05$ and absolute log fold change $\geq 0.58$ were defined as DEG in a given tissue compared to others. For each structural measure, tissue enrichment was first conducted for all associated proteins, followed by separate tissue enrichment for positively and negatively associated proteins. Similarly, for each structural measure, GO biological process enrichment was performed separately for positively and negatively associated proteins by using the over-representation analysis function of the WebGestalt toolkit[80]. For all enrichment analyses, the encoding genes of 2920 proteins of UKB were used as a customised background gene set, and the significance threshold was set to FDR corrected $p < 0.05$, unless otherwise specified.

**MR associations between protein expression and brain structure.** Traditional observational epidemiological studies have long been hindered by challenges such as confounding. Instead, the MR method utilises Mendel's laws of segregation and independent assortment, which ensure that genetic variants are distributed independently of environmental influences and other genetic factors. Hence, the genetic associations should therefore be largely free from confounding[81]. This could enhance the reliability of the results by leveraging genetic variants as instrumental variables to assess the MR relationship between protein expression and brain structure. To further enhance the significant protein-brain structure associations in the first analysis, bidirectional MR analyses were performed by using the TwoSampleMR R package (version: 0.5.8)[82]. For forward MR analysis with protein expression as exposure and structural measure as outcome, the IVW method was used as the primary inference. The independent SNPs from the GWAS summary of proteins were selected by using the clumping technique programmed in PLINK 1.9 software[68], with an r2 threshold of 0.01, a window size of 1 Mb, and a $p$ threshold of $5 \times 10^{-6}$. Putative outliers were detected with Cochran's Q test for the IVW method and Rucker's Q' test for the MR-Egger model. The outliers with a nominal $p$-value < 0.05 were excluded. If the final number of suitable genetic instruments was less than three, we excluded the protein-brain structure pairs from MR analyses. The significance of the relationship was determined using two criteria: FDR-corrected $p < 0.05$ and a less strict significance threshold of nominal $p < 0.001$. For reverse MR analyses with structural measures as exposure and protein expression as outcome, genetic instruments were selected from the GWAS summary statistics of brain structure measures at a $P$ threshold of $5 \times 10^{-6}$. The clumping and outlier detection was performed using the same procedure as the forward MR. Moreover, to enhance the reliability of the MR results, the bidirectional MR analyses were repeated with a stricter clumping at a $p$ threshold of $5 \times 10^{-8}$. In addition, we conducted another four MR methods: MR-Egger, weighted median, weighted mode, and Wald ratio, with the former three to complement the potential bias in the IVW results, and the last one when only one genetic instrument was valid.

To consider a potential violation of MR assumptions, three sensitivity analyses were further conducted to verify the significant associations between proteins and brain structures. Firstly, a leave-one-out analysis was utilised to check whether the observed MR relationship was driven by a single SNP[83]. Secondly, the MR-PRESSO technique was used to detect possible horizontal pleiotropy[84], when a genetic variant influences the brain structure (directly or indirectly, through other traits) independently of the hypothesised protein. Thirdly, MR-Egger regression analysis was used to detect possible directional pleiotropy[85], when a genetic variant associated with protein levels influences brain structure through a shared pathway rather than a direct effect of the protein on the brain structure.

To further control the false positive rate in the presence of pleiotropic effects, we performed CAUSE analysis to distinguish causality and genetic correlation using the cause R package. Similar to that of MR analysis, two clumping thresholds (i.e., $5 \times 10^{-6}$ and $5 \times 10^{-8}$), were used, and the significance threshold was set to FDR-corrected $P < 0.05$.

**Pleiotropic association between plasma pQTL and brain eQTL.** If an association is found between the expression of proteins in plasma and brain structure, it is important to try to determine the mechanisms by which the relevant peripheral plasma proteins influence brain structure. In the first sub-analysis, the investigation may be begun by characterising the gene expression of the brain structure-associated proteins across the 34 main tissues represented in the GTEx v8 database. Next, the possibility that the plasma protein-brain structure relationship is attributable to shared genetic regulation between plasma protein and brain gene expression can be examined. Specifically, for each brain structure-associated protein surviving MR analysis, we used the SMR method (version: 1.3.1)[86] to integrate summary-level pQTL data with eQTL data to identify genes whose protein expression in plasma are associated with their expression levels in the brain because of pleiotropy. The HEIDI test was conducted, and the significance threshold was set to $P_{SMR} < 0.05/32$ and $P_{HEIDI} > 0.05$.

**MR analysis between proteins and disorders.** In the second sub-analysis, to investigate whether proteins exhibited potential effects on brain structure that could contribute to the risk of central nervous system disorders, the bidirectional MR analyses were performed between proteins that exhibited significant MR associations with brain structure at FDR-corrected $p < 0.05$ and ten disorders. In other words, we leveraged genetic variants as instrumental variables to assess the MR relationship between protein expression and common diseases. A similar MR procedure to that of protein-brain structure measures was utilised, among which the clumping at a $p$ threshold of $5 \times 10^{-6}$ was performed in the main analysis, and clumping at a $p$ threshold of $5 \times 10^{-8}$ was conducted for validation. The significance threshold for the bidirectional MR analysis was set to FDR-corrected $p < 0.05$, and details of the GWAS summary statistics for six psychiatric disorders and four neurodegenerative disorders are included in the relevant section. For case-control GWAS summary statistics, the OR statistics were converted to log odds before MR analysis. CAUSE analysis was also conducted to distinguish causality and genetic correlation.

**Mediation analysis.** Finally, In the third sub-analysis, it is crucial to examine whether the plasma proteins could contribute to phenotypes relevant to mental health through the brain structure. A total of 3270 participants with a mean age of 54.2 were eligible for inclusion in mediation analysis (Supplementary Data 4). Accordingly, by using the SEM in the lavaan R package (version: 0.6.17)[87], a mediation analysis study was performed for a subset of UKB participants for whom proteomic, neuroimaging, and also mental health data were available. In particular, the protein-associated brain structural measures were used to construct the latent structural measure, the mental health phenotypes derived from UKB, including depression, anxiety, mania, well-being, psychotic experience, self-harm, mental distress, and trauma, were used to construct the latent mental health phenotype[18]. An iterative strategy was used to remove observable variables of high collinearity when constructing latent variables. Specifically, given a specific protein, for a pair of protein-associated structure measures with a correlation greater than the threshold, the structure measures with the less significant protein-structure association were removed. By requiring the weights of all surviving observational variables contributing to latent structural measures to fall within the range $[-1,1]$, while minimising the removal of observational brain structure

measurements, a correlation threshold of 0.55 was chosen. All variables were inverse-rank normalised before SEM analysis, and the SEM models for which there was a significant association between protein and brain structure (i.e., model a), brain structure and mental health (model b), protein and mental health (model c), as well as significant indirect effect (model a x b), were considered as valid mediations. The exploratory mediation analysis was performed for proteins that survived protein-brain structure MR analysis, and investigations were performed for both a strict FDR-corrected threshold of $p < 0.05$ and a less strict threshold of nominal $p < 0.001$.

## Reporting summary

Further information on research design is available in the Nature Portfolio Reporting Summary linked to this article.

## Data availability

All plasma proteomic, neuroimaging, genomic and mental health phenotype data are publicly available at the UK Biobank (http://www.ukbiobank.ac.uk/) and could be accessed with a reasonable request. The data in the present study were used according to the application no. 19542 and 202239. The GWAS summary data for the disease can be found at the Psychiatric Genomics Consortium (https://www.med.unc.edu/pgc/download-results) and the FinnGen website (https://www.finngen.fi/fi). The in vitro tissue gene expression data are available at GTEx (https://www.gtexportal.org/home/). The GTEx v8 cis-summary data of the brain cortex tissue can be downloaded from https://yanglab.westlake.edu.cn/software/smr/#eQTLsummarydata. Source data are provided in this paper.

## Code availability

All software and analytical methods used in this study are publicly available. The code used to run the analyses in this study is publicly available at https://github.com/hitrp/proteomicSignatureOfBrainStructure/. Brain visualisations were generated by the authors using the ENIGMA Toolbox (https://enigma-toolbox.readthedocs.io/), which is distributed under the open-source license permitting academic use. Additional figures were created using the ggplot2 package in R and the BioRender platform. Figures created with BioRender are in compliance with BioRender's Academic License Terms and are intended for publication purposes.

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

## Acknowledgements

This study used the UK Biobank Resource under application numbers 19542 and 202239. We thank all participants and team members of the UK Biobank. We thank the participants and investigators of the FinnGen study. W.C. was supported by grants from the National Key R&D Programme of China (No. 2023YFC3605400) and the National Natural Science Foundation of China (No. 82472055). J.-T.Y. was supported by grants from the Science and Technology Innovation 2030 Major Projects (no. 2022ZD0211600), the National Natural Science Foundation of China (nos 82071201, 81971032 and 92249305), the Shanghai Municipal Science and Technology Major Project (nos 2018SHZDZX01 and 2023SHZDZX02), the Research Start-Up Fund of Huashan Hospital (no. 2022QD002), the Excellence 2025 Talent Cultivation Programme at Fudan University (no. 3030277001), Shanghai Talent Development Funding for the Project (no. 2019074), and the Zhangjiang Lab, Tianqiao and Chrissy Chen Institute, and the State Key Laboratory of Neurobiology and Frontiers Centre for Brain Science of Ministry of Education, Fudan University. J.-F.F. was supported by the National Key R&D Programme of China (nos 2018YFC1312904 and 2019YFA0709502), the Shanghai Municipal Science and Technology Major Project (no. 2018SHZDZX01), the 111 Project (no. B18015), the Shanghai Centre for Brain Science and Brain-Inspired Technology and the Zhangjiang Lab. The funders had no role in study design, data collection and analysis, decision to publish or preparation of the manuscript. The Genotype-Tissue Expression (GTEx) Project was supported by the Common Fund of the Office of the Director of the National Institutes of Health, and by NCI, NHGRI, NHLBI, NIDA, NIMH, and NINDS.

## Author contributions

All authors had full access to the data in the study and accepted responsibility to submit it for publication. Conceptualisation, W.C., J.-T.Y. and M.-Y.W.; Methodology, P.R., X.-H.H., B.-S.W., W.-K.G., C.S. Y.-J.Z., Q.M. and W.Z., Formal Analysis, P.R., X.-H.H., Z.-Y.L., J.Y. and Y.-Z.L.; Data Curation, W.Z., Y.-Z.L., J.Y. and Z.-Y.L.; Writing–Original Draft, P.R. and X.-H.H.; Writing–Review & Editing, P.R., X.-H.H., B.Z., L.-B.W., J.-J.K., Y.-C.J., N.R., Y.H., F.X. and Y.M.; Visualisation, P.R. and X.-H.H.; Supervision, W.C., J.-T.Y. and M.-Y.W.; Project Administration, W.C., J.-T.Y. and M.-Y.W.; Funding Acquisition, W.C., J.-T.Y. and M.-Y.W.

## Competing interests

The authors declare no competing interests.
