## [Transparent Peer Review file · Nature Communications]

Atlas of Proteomic Signatures of Brain Structure and Its Links to Brain Disorders

Corresponding Author: Professor Wei Cheng

Version 0:

Reviewer comments:

Reviewer #1

(Remarks to the Author)

Atlas of proteomic signature of brain structure for study of brain disorders

Running title: Plasma proteomic signature of brain structure

This is a novel and exciting study and is potentially a wonderful resource for the neuroscience and proteomics community.

The key results are overshadowed by the context in which the introduction and parts of the results are phrased and interpreted. While the authors attempt to explain the rationale of using plasma proteins to gain insights into brain structure (and I agree with the concept), careful and substantial revision of the introduction text and description is required. I suggest the authors consider that the brain proteome is not measured in this study. Therefore the plasma proteins measured do not 'contribute directly to the variation in brain structure' but rather act as peripheral markers (or biomarkers) of brain structure measurements, and brain disorders. The authors nicely summarise the previous papers showing correlation between CSF and plasma proteins, and studies which found an association between plasma proteins and brain disorders. However the 'plasma proteomic findings' are presented as having a 'causal' effect on brain structure which is not supported by the data. While I appreciate the MR analysis uses this terminology in the formula and descriptions of the statistical methods, it is not appropriate in the context of interpretation of these results.

The rationale for using the MR analysis should be further explained in methods. I can't comment on its validity as it's not my area of expertise. I realise this is an epidemiological tool however the language used in regards to the plasma protein expression results having a 'causal effect' on disorders and brain structures should be carefully considered and revised.

In regards to the proteomics methodology and measurements, I believe the OLink platform and the QC data meets expected standards in the field. That said, the details required in regards to the proteomics data pre-processing is not sufficient. In the methods section the author's need to state exactly how the quality control was undertaken, and provide exact details on how the protein data was normalised. A major concern is how the missing data was handled. The authors report proteins with missing data of 50% or higher were imputed. The standard in the proteomics field is 70% or higher. 50% is low, can the authors justify. What was the percent of missing data and how much imputation was carried? Was there a cut off applied for the data imputation, how was imputation performed, and how do the authors know excessive imputation did not impact the proteins identified as differentially expressed? It is standard in proteomics to include all this information on data pre-processing, normalisation, imputation etc, in the methods and results.

The raw Olink data was not uploaded/provided. There is a link to github.com but the actual proteomic raw data is not provided. The data file uploaded just contains the -protein and -accession number but the plasma protein expression values/assay results are not provided.

In the 'association analysis' methods section the authors report on using 'protein concentration, age at imaging visit, sex, years of education, Townsend index, 224 ethnicity, smoking and drinking status at imaging visit, interval between baseline and 225 imaging visit, scanning site, total intracranial volume all used as predictors of the structural measures.' It's difficult to interpret how co-varying for all these measures impacted the data. Ideally they should summarise the effect before and after

covariate analysis or present in a table. Elsewhere they report adjusting for 'gender and age'. Can the authors clarify how the data was adjusted and provide a rationale. The sex effects would be very interesting and additional information whether particular brain measurements were associated with gender is important. If an effect of gender or age on brain measurement was previously found, then a sub analysis looking at associations between plasma protein expression and brain measurements in males and females should be considered.

Brain neuroimaging measurements are not clearly defined in the methods, Table 1 or figure legends. While these may be available on the UK biobank links or in previous studies they need to be clearly defined here to aid interpretation. This makes interpretation of the results challenging. In particular results in figure 2 are confusing. It's not clear if the proteins in the volcano plots Fig2A are associated with TOTAL brain volume or the volume of SUB-REGIONS of the brain coloured in Figure 2B. This needs to be clarified and the same applies to proteins associated with thickness (is this cortical thickness?), brain area, FA, MD etc. in figure 2.

The rationale for using the Genetic data and GWAS data is not clear and it complicated the interpretation of the results. To aid interpretation the authors state the research question in the introduction section. The data is not mentioned until it shows up in methods in the 'genetics data' on page 7 of methods, and the 'GWAS summary' on page 8. It's not clear how this data was linked with the proteomics data until the discussion section. This made revising the manuscript particularly difficult given that it's 36 pages, excluding all the supplementary info.

Crucial data on the study population is not provided. From the 4,997 participants – what is the gender distribution? This is not provided in Table 1 or supplementary fig 1. What proportion of these had mental health issues? What proportion of these had brain disorders such as ASD, ADHD, AD etc? Plasma proteomic associations are made with these outcomes yet the number of subjects in each category are not presented in the manuscript. This lack of information prohibits interpretation of the validity of the findings and robustness of the data in regards to mental health and disease outcomes.

The authors do not refer to the supplementary tables available. This should be added throughout the manuscript.

Minor corrections/suggestions to improve manuscript

- No table legends provided / definitions of brain area FA MD etc. are missing.
- Running title is more transparent than the main title of manuscript, suggest this is adopted as main title
- Plasma proteins should be noted in the keywords
- There are too many acronyms which hinder interpretation
- The authors continuously report on 'protein concentration' and various associations but this I believe is incorrect. The authors report that differential expression of plasma proteins across 5000 subjects as opposed to using the absolute 'concentration' of a protein in place. Suggest replacing with 'protein expression' which is the norm in the field of proteomics.

(Remarks on code availability)

the raw Olink data was not uploaded/provided. There is a link to github.com but the actual proteomic raw data is not provided. The data file uploaded just contains the -protein and -accession number but the plasma protein expression values/assay results are not provided.

Reviewer #2

(Remarks to the Author)

This manuscript analyzed the association between blood protein levels and brain morphology and explored their underlying biological and clinical implications. The key finding is the confirmation of a robust association between periphery and central nervous system, and highlighted different role of neural and immunological genes in brain morphology and the key roles of NCAM1 and MOG. I appreciate that this study is rigorous and valuable, but some issues should be clarified before publication.

My major concern is on how the observed association and causality (MR result) is interpreted. In most scenarios, peripheral molecules is regarded as a proxy of central molecules, and the observed association could be simply interpreted as "peripheral changes could reflect CNS alterations". This study, however, added MR analysis, and the result seemed to indicate that peripheral proteins had a direct causal effect on brain structure. This is quite surprising, and more validation is required. In fact, I believe that a more intuitive explanation is that brain morphology is genetically correlated to neuron-related protein expression, and that such genetic correlation drives false positive MR result. A suggestive evidence is that there is quite a lot of reverse causality (although not overlapped with forward causality): how could brain structure impacts blood gene expression? The genetic correlation model is apparently more intuitive here. I thus suggest the author to apply CAUSE to distinguish causality and genetic correlation. I hypothesize that there would be difference between neural and immunological genes: for the former, peripheral expression is simply a proxy of CNS expression; for the latter, peripheral expression represents an average expression across the body, and could have some causal roles. Similar difference may also exist in colocalization test.

Furthermore, I'm very confused on the study design of colocalization test:

- 1) What does CADD have to do here? I've never seen CADD analysis on common SNP.
- 2) Shouldn't colocalization be applied to eQTL and pQTL of the same gene? Does "protein-gene pair" indicate that this

analysis involves nearby eQTL of other genes? If such colocalization happens, it means that peripheral protein expression is actually tagging different proteins in the CNS, and that many biological interpretations based on peripheral signals are flawed.

3) MR is based on genome-wide pQTL, whereas colocalization test is based on cis-pQTL (the method section did not clarify this, but the result indicate so). Why not use methods like SMR that include only cis QTL?

4) It is well-known that QTL of difference tissues could have large proportion of colocalization, so what is the specific conclusion of this paper? Which blood protein shared regulation mechanism with CNS and which is actually tagging another CNS protein? Simply provide the overall result does not step forward from known knowledge.

There are some technical issues to be revised:

1) Is the protein-MRI association p value well-calibrated? Is there any inflation?

2) On figure: What's the meaning of drawing a line from data point to the y-axis? For GO result, consider using bubble plot to visualize the result, where point size or color showed the fold enrichment.

3) Compared with other participants without MRI, proteome, or both, does the included participants showed significant difference in demographic variables?

4) I believe that merely the most straightforward conclusion "blood proteins are associated with CNS alteration" is quite important for the field. This simple conclusion could be more highlighted in the manuscript.

5) On MR analysis: usually the additional MR methods (weighted mode, weighted median) would be applied regardless of sensitivity analysis. All MR methods give consistent significant result is more persuasive.

(Remarks on code availability)

Reviewer #3

(Remarks to the Author)

SYNOPSIS: This manuscript aims to provide a comprehensive atlas mapping the associations between proteomic signatures and brain structure, offering insights into brain disorders. Using data from 4,997 UK Biobank participants, the authors examined relationships between 2,920 plasma proteins and 272 MRI-derived brain structure measures, including regional grey matter volume, thickness, surface area, and white matter integrity (FA, MD). The analysis identified 5,358 significant associations involving 1,143 proteins and 256 brain structure metrics. Functional enrichment analysis revealed links to neurogenesis, brain tissue integrity, neuronal cells, and immune responses. Additionally, bidirectional Mendelian randomization uncovered significant causal relationships between 32 proteins and 23 brain structure measures, as well as 9 proteins linked to 10 neurodegenerative and psychiatric disorders. The study also identified associations between these proteins and mental health, mediated through regional brain volume and surface area.

OVERALL COMMENTS

Overall, while the manuscript presents a valuable/noteworthy set of results, there are several key areas that need further attention. A main concern is the minimal use of references across the Introduction, Methods, and Discussion sections, given that up to 70 refs are allowed.

In the Introduction, the rationale for the study could be expanded to better highlight key gaps in the field. The section would benefit from additional references that provide more robust context and emphasize the significance of addressing these gaps, helping to frame the study's objectives more clearly.

In the Methods section, while the approach taken is logical and sound, several methodological decisions are presented without adequate citation of relevant literature. Including references for key procedures and criteria would enhance the transparency and reproducibility of the study. This is particularly important for justifying the thresholds used in genetic analyses and other statistical methods.

In the Discussion, the results are mostly summarized but lack deeper reflection on their implications. It would be beneficial to explore how these findings address existing gaps and to provide a clearer explanation of the complementary insights offered by the results, demonstrating how they collectively advance our understanding of the topic.

Additionally, there are some grammatical inconsistencies throughout the text that affect its clarity and readability. A thorough review and revision would greatly enhance the manuscript's overall presentation and ensure a smoother reading experience.

----- SPECIFIC COMMENTS -----

METHODS

The rationale for selecting five imaging-derived phenotypes (IDPs)—specifically grey matter volume, surface area, and cortical thickness, along with white matter fractional anisotropy (FA) and mean diffusivity (MD)—from the multiple IDPs released by the UK Biobank is not clearly explained. The authors should provide a clear justification for this selection. While the methods section is generally well-explained, it would benefit from the inclusion of key references.

- The authors should cite major UK Biobank publications that introduced imaging-derived phenotypes (IDPs) and proteomics.

- In the "Genetic Data Processing and Analysis" section, several key concepts and decisions lack citations. Relevant

references should be added to support these points. For example, the exclusion criteria for single nucleotide polymorphisms (SNPs)—those with a missing rate > 5%, minimum minor allele frequency (MAF) < 0.1%, or Hardy–Weinberg equilibrium test $P < 1 \times 10^{-10}$ —should be backed by appropriate references or a more detailed explanation, as these decisions can significantly influence the robustness and generalizability of the findings.

- In the “GWAS summary of CNS disorders,” GWAS references are missing for the neurodegenerative diseases. Please provide. Also, include a rationale for choosing these four neurodegenerative disorders along with the sample sizes for these studies in the main document (in addition to the range of N).

- Please provide citations for key methodological decisions used in the “Causal relationship between protein concentration and brain structure” and ‘eQTL mapping and colocalization’ sections. For example: MR-Egger analysis (line 277), leave-one-out-analysis (line 275) etc.

Provide a definition or brief explanation of “colocalization analysis” in the “Gene Expression and Quantitative Trait Loci” section for clarity (either in the main or supplementary documents).

For improved readability, please present the “Statistical analyses” subsections in the same order as they appear in the Results section.

Clarify whether the physical annotation method, which calculates the density of proteins associated with brain structure on each chromosome, is based on a previously published reference or an in-house method.

Include a concise definition or explanation of horizontal and directional pleiotropy in the “Causal relationship between protein concentration and brain structure” section.

I also suggest reconsidering the subheading to explicitly mention both eQTL and pQTL for clarity. It would be good to have these concepts defined concisely in supplementals.

Please provide N of the UKB subset used in the “Mediation analyses” section of the main document. Additionally, could the authors provide a rationale or reference for the chosen correlation cutoff of 0.55 used to identify and remove high collinearity between protein-associated structure measures in the iterative strategy?

DISCUSSION

The discussion section requires significant improvements. Much of the current text simply restates the results, with minimal interpretation of their significance or connection to existing literature, as reflected by the sparse citations. This approach significantly underutilizes the potential of the discussion, missing a critical opportunity to position this study as “the first comprehensive atlas of the proteomic signature of human brain structure.” While I include a few illustrative examples below, these are only a starting point, and a more thorough, in-depth discussion is needed throughout.

Example 1: The statement regarding the highest number of significant associations observed for WM MD (2,501), followed by GM volume, surface area, thickness, and the fewest for WM FA (86), is clear in terms of the ranking (results section, lines 362-364). However, it would greatly benefit from additional context and interpretation. Could the authors provide an explanation for why WM MD showed the highest number of associations, and why WM FA had the least? A discussion on the potential biological or methodological reasons behind these differences would enhance the reader's understanding and strengthen the results' interpretation.

Example 2: Six proteins (NCAN, SLITRK1, MOG, LEP, OXT, PAEP) were highlighted in lines 549-552 for their numerous associations with brain structure. However, there is no discussion of their established roles in the brain, apart from a brief mention that NCAN is linked to brain volume. Expanding on the relevance of the top five proteins showing positive and negative associations to brain structure (mentioned in results section, lines 369-372) in a cohesive manner would enhance the discussion: NCAN, SLITRK1, MOG, PTPRN2 and SEZ6L; LEP, OXT, PAEP, CCN5 and XG

Example 3: The investigation into the chromosomal locations of brain structure-associated proteins lacks context. The results (Results section, lines 383-385) show a concentration on chromosomes 19, 17, and 22, yet no explanation is provided for why this might be significant. A discussion on the potential links of these chromosomes to brain diseases would add valuable context to these findings.

Example 4: The findings that the cingulate gyrus part of the cingulum, parahippocampal part of the cingulum, and thalamic radiation exhibited the most causal associations with proteins are intriguing (Results section, lines 415-417). However, I would have liked to see a more thorough discussion of these results in the context of existing literature. Specifically, elaborating on the known roles of these tracts in neurodegenerative diseases or cognitive function could provide valuable insights and enhance the overall interpretation of the findings.

Example 5: Please discuss why certain brain regions, such as the precentral cortex, are mediating the association between proteins and mental health (Discussion section, lines 624-630). How do these mediating effects align with what is known about these brain regions in mental health disorders?

(Remarks on code availability)

I do not consider myself an expert in code review.

Reviewer #4

(Remarks to the Author)

(Remarks on code availability)

Version 1:

Reviewer comments:

Reviewer #1

(Remarks to the Author)

The revisions with regards to implying direct causal effects of plasma protein expression on brain structure are satisfactory.

The additional proteomics information regards to the management of QCs, and and swapping 'protein concentration' with 'protein expression' are satisfactory.

Additional information in regards the participant demographics (Table 1), and supplementary table 5 are welcome and adequately addressed.

Outstanding concerns

Although I cant comment on the validity of MR analysis as it's not my area of expertise, I appreciate the rational added (line 281-286) and the more conservative conclusions. Can the authors further clarify what is meant by the following statement 'By using genetic variants that are randomly assigned at conception, the MR analysis could minimize the bias'?

- Is my interpretation correct in that you are using genetic predictors of protein expression to test the association of protein expression on brain structure and on common diseases? This needs to be crystal clear in the paper as its not typical in proteomic analysis.

In regards to imputation, the authors state 'all proteins showed a missing rate of less than 25%' Does this mean 25% of the dataset was missing/imputed? Can the authors insert the % missing data or % imputation in the manuscript for transparency so the quality of the proteomics data generated can be clearly interpreted.

Line 421 of the Results: "We found that the effects of protein expression were highly correlated between females and males." (Supplementary Table 6)."

The authors report protein expression was 'highly correlated' and indeed when a pearsons correlation is performed a perfect correlation $r = 1$ was achieved as the M/F data compared is exactly the same.

Protein expression values reported (FDR value; SE value) is exactly the same for the male only and females only analysis presented in supplementary table 6. This is uncanny as I find it unlikely that the expression data for >2000 proteins is exactly the same for the male only and the female only subjects.

Line 250 of the Methods: "Third, to examine potential sex difference, the association analysis was further conducted in female and male separately".

Please provide results of this association analysis in females and in males in the manuscript. The authors report results in the rebuttal but not the manuscript. Further the authors should comment on these results. In particular, there is weak correlation between cortical thickness (0.55) and WM FA (0.6) between sexes, they authors need to comment address this impact on interpretation of results.

Table 1: The authors report the age at imaging visit (year). Similarly, the authors need to report the age (years)when the blood samples was taken for proteomics in Table 1. It is noted that blood was sampled at the beginning of the study but its now clear if there is a significant gap in years between imaging and blood sampling.

Direction/ referencing corresponding results files in supplementary is unsatisfactory. When referring to results the reader should be pointed to the specific results table in supplementary.

Minor comments

Please provide a reference for this statement, second paragraph in the introduction

'Measurement of the reduction of complement proteins can be used to predict the extent of brain atrophy in mild cognitive impairment patient'

In regards to Figure 6/ Results shown the association between protein and disease outcome. The authors should comment carefully on interpretation of these results, particularly as the age profile of subject included is 40-69 years, and the plasma proteomic profile in younger subjects is likely to differ. The disease associated proteins identified may reflect later plasma protein markers of the disease, particularly as ASD, ADHD, Scz, BPD, Anxiety have a onset in childhood and/or early adulthood. Therefore these bloods plasma changes have limited prognostic value, and future validation is needed to identify if the disease associated protein changes are longitudinal or reflect late stages of the disease.

(Remarks on code availability)

Links to the code is provided via github

I queried a folder labelled 'data' in github as follows;

The raw Olink data was not uploaded/provided. There is a link to github.com but the actual proteomic raw data is not provided. The data file uploaded just contains the -protein and -accession number but the plasma protein expression

values/assay results are not provided.

Response: Thank you for your helpful feedback regarding the data availability. We recognize the importance of providing comprehensive data for reproducibility and transparency. The raw Olink data used in our study was acquired from the UK Biobank, as described in the Methods section. As part of the UK Biobank policies, we are not allowed to provide actual proteomic raw data elsewhere. The Olink data is publicly available on the UK Biobank website upon application for access (<https://www.ukbiobank.ac.uk/>).

Reviewer #2

(Remarks to the Author)

The authors have addressed all my concerns

(Remarks on code availability)

Reviewer #3

(Remarks to the Author)

Thank you for addressing the overall comments which were:

“In the Introduction, the rationale for the study could be expanded to better highlight key gaps in the field. The section would benefit from additional references that provide context and emphasize the significance of addressing these gaps, helping to frame the study's objectives more clearly.

In the Methods section, several methodological decisions are presented without adequate citation of relevant literature. Including references for key procedures and criteria would enhance the transparency and reproducibility of the study. This is particularly important for justifying the thresholds used in genetic analyses and other statistical methods.

In the Discussion, the results are somewhat summarized but lack deeper reflection on their implications. It would be beneficial to explore how these findings address existing gaps and to provide a clearer explanation of the complementary insights offered by the results, demonstrating how they collectively advance our understanding of the topic.

Additionally, there are some grammatical inconsistencies throughout the text that affect its clarity and readability. A thorough review and revision would greatly enhance the manuscript's overall presentation and ensure a smoother reading experience.”

Here are the responses to the specific comments which were shared with the author.

METHODS

Comment 6: The rationale for selecting five imaging-derived phenotypes (IDPs)—specifically grey matter volume, surface area, and cortical thickness, along with white matter fractional anisotropy (FA) and mean diffusivity (MD)—from the multiple IDPs released by the UK Biobank is not clearly explained. The authors should provide a clear justification for this selection.

Response: Thank you for pointing this out. As we are only interested in brain structures that could be stably measured, the functional IDPs in UKB are beyond the scope of this manuscript. To provide a clearer rationale for selecting the five imaging-derived phenotypes (IDPs) in our study, we have revised the manuscript to include a detailed justification for our choices, explaining how each IDP contributes to our understanding of brain structure in the context of our research objectives: Line 160 of the Methods section: “Given that gray and white matter are crucial components that serve distinct functions for overall brain health, here we included brain structural imaging derived phenotypes (IDPs) that were previously made available by the UKB neuroimaging group¹⁹. Similar to previous researches, regional grey matter measures (volume, thickness, and area) derived from T1-weighted images and microstructural measures derived from diffusion weighted imaging, including fractional anisotropy (FA) and mean diffusivity (MD), were chosen to be used in the present study²¹.”

Reviewer response: The comment was acknowledged and the author clarifies that gray and white matter are crucial for brain health and are stable in nature, hence chosen for analysis.

Comment 7: The authors should cite major UK Biobank publications that introduced imaging derived phenotypes (IDPs) and proteomics.

Response: We apologize for missing the major UKB publication that introduced IDPs and proteomics. We have now added the corresponding citations as follows: Line 162 for IDPs: (Thomas J. Littlejohns et al., Nature Communications, 2020) Line 151 for proteomics: (Benjamin B. Sun et al., Nature, 2023)

Reviewer response: The comment was acknowledged and the author has added the necessary references.

Comment 8: In the "Genetic Data Processing and Analysis" section, several key concepts and decisions lack citations. Relevant references should be added to support these points. For example, the exclusion criteria for single nucleotide polymorphisms (SNPs)—those with a missing rate > 5%, minimum minor allele frequency (MAF) < 0.1%, or Hardy–Weinberg equilibrium test $P < 1 \times 10^{-10}$ —should be backed by appropriate references or a more detailed explanation, as these decisions can significantly influence the robustness and generalizability of the findings.

Response: We apologize for missing the key citations supporting the key methodology decisions. According to your suggestions, we have added the appropriate references as follows: Line 178 of the Methods section: (Clare Bycroft et al., Nature, 2018; Yuzhu Li et al., Nature Aging, 2022)

Reviewer response: The comment was acknowledged and the author has added the necessary references (Bycroft et al., 2018; Li et al., 2022), which are strong choices for supporting SNP quality control.

Comment 9: In the "GWAS summary of CNS disorders," GWAS references are missing for the neurodegenerative diseases. Please provide. Also, include a rationale for choosing these four neurodegenerative disorders along with the sample sizes for these studies in the main document (in addition to the range of N).

Response: We apologize for not including the introduction of the four neurodegenerative diseases. We have now added the rationale and relevant GWAS references for the selected neurodegenerative diseases as follows: Line 194 of the Methods section: "Given their significance in public health and the relatively large number of cases, GWAS summary statistics were collected from publicly available databases for six psychiatric disorders (attention deficit hyperactivity disorder (ADHD, cases/total=19,099/53,293), anxiety (cases/total=7,016/18,186), autism spectrum disorder (ASD, cases/total=18,381/46,350), bipolar disorder (BP, cases/total=20,352/51,710), major depressive disorder (MDD, cases/total=45,396/142,646), schizophrenia (SCZ, cases/total=67,390/161,405)), and four neurodegenerative disorders (Mitja I. Kurki et al., Nature, 2023) (Alzheimer's disease (AD, cases/total=10,520/401,661), Parkinson's disease (PD, cases/total=4,681/407,500), multiple sclerosis (MS, cases/total=2,409/408,561) and amyotrophic lateral sclerosis (ALS, cases/total=531/184,000))".

Reviewer response: The psychiatric GWAS sample sizes vary significantly, with anxiety disorder having a much smaller N than schizophrenia or MDD.

Similarly, ALS has a very small case count (531 cases out of 184,000 total) compared to AD or PD. Is this accounted for in the analysis? If so, how?

Comment 10: Please provide citations for key methodological decisions used in the "Causal relationship between protein concentration and brain structure" and 'eQTL mapping and colocalization' sections. For example: MR-Egger analysis (line 277), leave-one-out-analysis (line 275) etc.

Response: Thank you for raising these issues. We recognize that detailed citation for key methodologies could strengthen the validity of our manuscript. According to your suggestions, we have added citations for MR-Egger analysis in line 312 (Jack Bowden et al., International Journal of Epidemiology, 2015), leave-one-out-analysis in line 309 (Gibran Hemani et al., Elife, 2018). According to the 2nd reviewer's suggestion, we have replaced the colocalization analysis with Summary-based Mendelian Randomization (SMR) analysis. Therefore, we did not include citation for colocalization but we added citation for SMR in line 328 (Zhihong Zhu et al., Nature Genetics, 2016).

Reviewer response: The comment was acknowledged and the author has added the necessary references.

Comment 11: Provide a definition or brief explanation of "colocalization analysis" in the "Gene Expression and Quantitative Trait Loci" section for clarity (either in the main or supplementary documents).

Response: Thanks for your suggestion. According to the 2nd reviewer's suggestion, we have replaced the colocalization analysis with Summary-based Mendelian Randomization (SMR) analysis. Therefore, following your suggestion, we have added an explanation of "SMR analysis". Line 327 of the Methods: "Specifically, for each brain structure-associated protein surviving MR analysis, we used the SMR method⁴² to integrate summary-level pQTL data with eQTL data to identify genes whose protein expression in plasma are associated with their expression levels in the brain because of pleiotropy. The heterogeneity in dependent instruments (HEIDI) test was conducted and the significance threshold was set to $PSMR < 0.05/32$ and $PHEIDI > 0.05$."

Reviewer response: The comment was acknowledged and the author has added the needed explanation for SMR.

Comment 12: For improved readability, please present the "Statistical analyses" subsections in the same order as they appear in the Results section.

Response: Thank you for your suggestion. We have reorganized the "Statistical Analyses" subsections to align with the order in which they appear in the Results section. We hope this adjustment could enhance readability and make it easier for readers to follow the analyses.

Reviewer response: Clearly acknowledges the change and explains the reasoning.

Comment 13: Clarify whether the physical annotation method, which calculates the density of proteins associated with brain structure on each chromosome, is based on a previously published reference or an in-house method.

Response: We apologize for missing the methodology details of the physical annotation. The mapping of the protein-coding genes to chromosome positions was based on a previously established *topr* R software (Thorhildur Juliusdottir, BMC Bioinformatics, 2023). The definition of the density used an in-house method that dividing the number of protein-coding genes of each chromosome by the length of each chromosome. We have incorporated detailed descriptions: Line 257 of the Methods section: "The physical annotation was performed by first mapping the protein-coding genes to chromosome positions with a previously established *topr* R software³⁴. Subsequently, the number of protein-coding genes on each chromosome was divided by the length of that chromosome, which generated the density of associated proteins in each chromosome that accounted for the length of the chromosome."

Reviewer response: Clarifies whether a published method or in-house approach was used.

Comment 14: Include a concise definition or explanation of horizontal and directional pleiotropy in the "Causal relationship between protein concentration and brain structure" section.

Response: We apologize for ignoring the definition of horizontal and directional pleiotropy. To give a better understanding of the Mendelian randomization results, we have incorporated the definitions in the Methods section: Line 309 of the Methods section: "Secondly, the MR-PRESSO technique was used to detect possible horizontal pleiotropy⁴⁰, when a genetic variant influences the brain structure (directly or indirectly, through other traits) independently of the hypothesized protein. Thirdly, MR-Egger regression analysis was used to detect possible directional pleiotropy⁴¹, when a genetic variant associated with protein levels influences brain structure through a shared pathway rather than a direct effect of the protein on the brain structure."

Reviewer response: Adds missing definitions in the Methods section, thank you for the addition.

Comment 15: I also suggest reconsidering the subheading to explicitly mention both eQTL and pQTL for clarity. It would be good to have these concepts defined concisely in supplementals.

Response: Thank you for this helpful comment. We have revised the subheading “Colocalization of plasma proteins and brain gene expression” to “Pleiotropic association between plasma pQTL and brain eQTL” in line 320 of the Methods section. Meanwhile, we have added the brief explanation of eQTL and pQTL in the Methods: Line 210 of the Methods: “Here we used pQTL and eQTL summary data to characterize genetic variants of protein and gene expression separately. A “pQTL” stands for “protein quantitative trait locus,” which refers to a genetic variant associated with the abundance of a specific protein, while an “eQTL” stands for “expression quantitative trait locus,” indicating a genetic variant linked to the expression level of a gene (mRNA transcript) at a specific locus on the chromosome.”.

Reviewer response: Clarifies subheading and adds definitions. Thank you.

Comment 16: Please provide N of the UKB subset used in the “Mediation analyses” section of the main document.

Additionally, could the authors provide a rationale or reference for the chosen correlation cutoff of 0.55 used to identify and remove high collinearity between protein associated structure measures in the iterative strategy?

Response: Thanks for your professional suggestions. We agree that the demographical information of UKB subset used in Mediation analysis should not only be provided in Supplementary Table 4 but also be briefly stated in the Methods section. Accordingly, we have added the statement as follows: Line 346: “A total of 3,270 participants with a mean age of 54.2 were eligible for inclusion in mediation analysis (Supplementary Table 4)”. -- The correlation cutoff of 0.55 was selected because it can ensure the weights of all surviving observational variables contributing to latent brain structures fall within the range [-1,1], while minimizing the removal of observational brain structure measurements. We have included the rationale as follows: Line 357: “By requiring the weights of all surviving observational variables contributing to latent structural measures fell within the range [-1,1], while minimizing the removal of observational brain structure measurements, a correlation threshold of 0.55 was chosen”.

Reviewer response: Adds missing N and explains correlation cutoff selection.

DISCUSSION

The discussion section has been substantially improved, addressing key concerns related to interpretation, contextualization, and citation density. The added discussions on brain regions, chromosomal associations, white matter tracts, and protein-mental health mediation effects enhance the manuscript’s contribution by framing it within the context of neurodegenerative and psychiatric research.

Comment 18: Example 1: The statement regarding the highest number of significant associations observed for WM MD (2,501), followed by GM volume, surface area, thickness, and the fewest for WM FA (86), is clear in terms of the ranking (results section, lines 362-364). However, it would greatly benefit from additional context and interpretation. Could the authors provide an explanation for why WM MD showed the highest number of associations, and why WM FA had the least? A discussion on the potential biological or methodological reasons behind these differences would enhance the reader’s understanding and strengthen the results’ interpretation.

Response: Thank you for your helpful comment. We have revised the Discussion sections to provide additional context and interpretation regarding the observed differences in the number of significant associations for WM MD and FA: Lines 637 of the Discussion section: “Though MD and FA are both metrics that reflect the microstructural integrity of white matter, MD demonstrated the most significant associations with proteins, whereas FA showed the least significant associations. The higher number of associations between MD and CSF proteins might stem from the fact that CSF proteins reflect early pathophysiological changes in the brain, to which MD is sensitive, especially in the early stages of neurological diseases such as AD45. In patients with MS, MD was more strongly associated with widespread brain degeneration than FA as well46. FA is primarily influenced by directional water diffusion and is more specific to structural coherence and fiber integrity47. This specificity may make FA less sensitive to diffuse or generalized changes and more reliant on pronounced structural disruptions.”.

Reviewer response: Thank you for the additional context and response. Also, the references used aptly support the statements made in the discussion section.

Comment 19: Example 2: Six proteins (NCAN, SLITRK1, MOG, LEP, OXT, PAEP) were highlighted in lines 549-552 for their numerous associations with brain structure. However, there is no discussion of their established roles in the brain, apart from a brief mention that NCAN is linked to brain volume. Expanding on the relevance of the top five proteins showing positive and negative associations to brain structure (mentioned in results section, lines 369-372) in a cohesive manner would enhance the discussion: NCAN, SLITRK1, MOG, PTPRN2 and SEZ6L; LEP, OXT, PAEP, CEN5 and XG.

Response: Thank you for your valuable feedback. To provide a more comprehensive exploration of the established roles of the top proteins showing positive and negative associations with brain structure, we have added the description of the proteins: Line 647 of the Discussion “NCAN, SLITRK1, MOG, PTPRN2 and SEZ6L were the top proteins that showed the highest total number of significant positive associations with brain structures. The links between these proteins and brain disorders have been reported in previous studies. NCAN, an extracellular matrix protein, is closely related to the development, neuronal senescence and apoptosis of the neurons. NCAN has been found to be associated with neuropsychiatric and neurodegenerative diseases48. In addition, NCAN demonstrated a predominant correlation with AD pathology49. Existing evidence has linked the plasma concentration of NCAN protein to total brain volume18. The findings of the present study have expanded the significant associations of NCAN protein to the volume of 50 regions, surface area of 43 regions, FA of three tracts and MD of five tracts. Apart from NCAN, the other proteins are also relevant to central neural system and brain diseases. SLITRK1 function is associated with the control of neurite development and synaptogenesis50. SLITRK has been linked to Tourette’s syndrome, trichotillomania, and obsessive-compulsive disorder51. MOG, expressed

in oligodendrocytes in the central nervous system, is an encephalitogenic protein that can trigger a demyelinating immune response⁵². Moreover, PTPRN2 has been implicated in AD and FTD⁴⁹, while SEZ6L has been associated with bipolar disorder⁵³. The proteins with significant negative associations with brain structure include LEP, OXT, PAEP, CCN5, and XG. LEP is well-known for its role in metabolic regulation and has been implicated in neurodegenerative diseases, potentially linking metabolic dysfunction to brain pathology⁵⁴. OXT is associated with social behaviors and stress responses⁵⁵. It has also been implicated in anxiety, depression and AD⁵⁶, although its relationship with brain structure remains unclear. Previous studies have highlighted associations between these proteins and CNS or brain disorders, and our findings further reveal their associations with brain structure. While the roles of PAEP, CCN5, and XG in brain diseases are not yet fully understood, our study identifies significant associations between these proteins and brain structure, offering a valuable basis for future studies.”

Reviewer response: Thank you for the additional context and response. It adequately addresses the comment.

Comment 20: Example 3: The investigation into the chromosomal locations of brain structure associated proteins lacks context. The results (Results section, lines 383-385) show a concentration on chromosomes 19, 17, and 22, yet no explanation is provided for why this might be significant. A discussion on the potential links of these chromosomes to brain diseases would add valuable context to these findings.

Response: Thanks for your suggestion. We have included an expanded discussion on the significance of the observed concentration of these proteins on chromosomes 19, 17, and 22, highlighting their potential links to brain diseases: Line 687 of the Discussion: “Our analysis revealed that brain structure-associated proteins are disproportionately localized to chromosomes 19, 17, and 22, relative to the relatively short length of these chromosomes. The concentration of brain structure-associated proteins on chromosomes 19, 17, and 22 is particularly noteworthy given the established relevance of these chromosomes to brain health and disease. Our findings show that many of the proteins identified on these chromosomes, including those encoded on chromosome 17 (OMG60, BAIAP261, GFAP62), chromosome 19 (NCAN49, APLP163, GDF1564), and chromosome 22 (NPTXR65, OSM66, RTN4R67) are strongly associated with brain structure and have been previously implicated in various neurological and psychiatric disorders. These results suggest that these chromosomes may serve as hotspots for genes influencing brain morphology and function, reflecting their broader role in neurodevelopmental and disease pathways.”

Reviewer response: Your response adequately addresses the comment but could be strengthened by adding disease relevance for each chromosome explicitly. Example: The concentration of brain structure-associated proteins on chromosomes 19, 17, and 22 is particularly notable given their established relevance to brain health and disease. Chromosome 19 is highly gene-dense and includes APOE, a key risk factor for Alzheimer’s disease. Chromosome 17 harbors genes implicated in neurodegenerative disorders, including tauopathies such as frontotemporal dementia. Chromosome 22 has been linked to neurodevelopmental and psychiatric conditions, such as schizophrenia and 22q11.2 deletion syndrome. (add necessary refs)

Comment 21: Example 4: The findings that the cingulate gyrus part of the cingulum, parahippocampal part of the cingulum, and thalamic radiation exhibited the most causal associations with proteins are intriguing (Results section, lines 415-417). However, I would have liked to see a more thorough discussion of these results in the context of existing literature. Specifically, elaborating on the known roles of these tracts in neurodegenerative diseases or cognitive function could provide valuable insights and enhance the overall interpretation of the findings.

Response: Thank you for your insightful feedback. To provide a more thorough exploration of the roles of the cingulate gyrus part of the cingulum, parahippocampal part of the cingulum, and thalamic radiation in neurodegenerative diseases and cognitive function, we have revised the Discussion sections as follows: Line 672 of the Discussion: “Thirty-three significant MR relationships between plasma proteins and brain structures were revealed through MR analysis. The strongest associations with proteins were observed in the cingulate gyrus part of the cingulum, parahippocampal part of the cingulum, and thalamic radiation. The cingulate gyrus part of the cingulum is involved in emotional regulation, attention, and memory, and disruptions in this tract have been implicated in conditions such as AD and major depressive disorder⁵⁷. The parahippocampal cingulum is closely associated with episodic memory and spatial navigation, and alterations in this tract are also observed in the early stages of AD⁵⁷. Our results highlight the potential involvement of the cingulum in the pathophysiology of neurodegenerative and psychiatric disorders through associations with specific proteins. The thalamic radiation serves as a critical pathway for relaying sensory and motor signals between the thalamus and cortical regions. Studies have demonstrated that alteration in the anterior thalamic radiation integrity is linked to schizophrenia and depression^{58,59}. Taken together, these findings suggest that the associated proteins may play key roles in maintaining the structural integrity and functional regulation of these tracts, thereby contributing to the pathophysiology of neurodegenerative and psychiatric disorders.”

Reviewer response: Your response effectively addresses the comment. Thank you for the explanation.

Comment 22: Example 5: Please discuss why certain brain regions, such as the precentral cortex, are mediating the association between proteins and mental health (Discussion section, lines 624-630). How do these mediating effects align with what is known about these brain regions in mental health disorders?

Response: Thank you for your thoughtful feedback. To address how the mediating effects of certain brain regions align with existing knowledge in mental health disorders. We have revised the Discussion section our manuscript as follows: Line 748 of the Discussion section: “The precentral cortex, primarily involved in motor control and coordination, has been increasingly recognized for its role in working memory and emotional regulation⁷⁸. Cortical thinning in the precentral gyrus has been associated with an increased risk of depressive symptomatology⁷⁹. In addition, advanced control relevant activation in the precentral gyrus has been linked with suicide risk in mood disorders⁸⁰. The high mediation proportion suggests that ENPP6 may influence mental health indirectly by modulating the structure or function of these regions.”

Reviewer response: Your response effectively addresses the comment. Thank you for the explanation.

(Remarks on code availability)
see response above

Reviewer #4

(Remarks to the Author)

(Remarks on code availability)

Version 2:

Reviewer comments:

Reviewer #1

(Remarks to the Author)

The additional information and improvements is appreciated and is important in interpreting the findings.

It is noteworthy that there is a 9 year gap between blood sampling for proteomics and the brain imaging which impacts interpretation of findings. Although I appreciate the authors note this gap was included as a covariate in the analysis. Further comments should be made in regards to caution when interpreting these proteomic findings given important literature demonstrating that the plasma proteome is known to change with age.

Lehallier, B. et al. Undulating changes in human plasma proteome profiles across the lifespan. *Nat. Med.* 25, 1843–1850 (2019).

Niu, L., Stinson, S.E., Holm, L.A. et al. Plasma proteome variation and its genetic determinants in children and adolescents. *Nat Genet* 57, 635–646 (2025). <https://doi.org/10.1038/s41588-025-02089-2>

(Remarks on code availability)

As per previous remarks

Here are point-to-point responses to Reviewers' comments. The Reviewers' comments are marked in blue. All the revisions made to the manuscript are highlighted with red colour. In case of multiple questions in one comment, the response to each question is preceded by '- -'.

Reviewer #1 (Remarks to the Author):

Comment 1: *This is a novel and exciting study and is potentially a wonderful resource for the neuroscience and proteomics community.*

Response: Thank you for all your insightful and constructive feedback. We appreciate your acknowledgment of the contribution of our work to the relevant fields. Your support encourages us to continue exploring this important area of research.

Comment 2: *The key results are overshadowed by the context in which the introduction and parts of the results are phrased and interpreted. While the authors attempt to explain the rationale of using plasma proteins to gain insights into brain structure (and I agree with the concept), careful and substantial revision of the introduction text and description is required. I suggest the authors consider that the brain proteome is not measured in this study. Therefore the plasma proteins measured do not 'contribute directly to the variation in brain structure' but rather act as peripheral markers (or biomarkers) of brain structure measurements, and brain disorders. The authors nicely summarise the previous papers showing correlation between CSF and plasma proteins, and studies which found an association between plasma proteins and brain disorders. However, the 'plasma proteomic findings' are presented as having a 'causal' effect on brain structure which is not supported by the data. While I appreciate the MR analysis uses this terminology to in the formula and descriptions of the statistical methods, it is not appropriate in the context of interpretation of these results.*

Response: Thank you for raising this important point. We appreciate your insights regarding the interpretation of our results and the context provided in the introduction. Following your suggestions, we have carefully revised the interpretation of MR analysis to avoid implying direct causal effects. We have replaced the description of “causal effect or relationship” with “associations or MR associations” throughout the manuscript. Here are some examples:

Line 61 of the Abstract: “*Utilizing bidirectional Mendelian randomization (MR), 33 significant associations between 32 proteins and 23 measures of brain structure, and 21 associations between nine brain structure associated proteins (e.g. BTN3A2, BTN2A1 and ENPP6) and ten neurodegenerative and psychiatric disorders were discovered.*”

Line 117 of the Introduction: “*Utilizing genetic data, the bidirectional Mendelian randomization analysis was performed to enhance the potential associations between proteins and brain structure...*”

Line 453 of the Results: “*Associations between proteins and brain structure identified through MR analysis*”

Line 460 of the Results: “*Notably, the cingulate gyrus part of cingulum, parahippocampal part of cingulum and thalamic radiation were the tracts that exhibited the most MR associations with proteins*”

Line 617 of the Discussion: “*The proteins exhibited significant associations with brain structure measures also demonstrated associations with the risk of neurodegenerative and psychiatric disorders.*”

Line 1068 of the Figure Legend: “*Figure 4 The association between protein and brain structure in the forward MR.*”

Comment 3: *The rational for using the MR analysis should be further explained in methods. I can't comment on its validity as it's not my area of expertise. I realize this is an epidemiological tool however the language used in regards to the plasma protein expression results having a 'causal effect' on disorders and brain structures should be carefully considered and revised.*

Response: Thank you for bringing this to our attention. Following your suggestion, we have introduced the rational for MR analysis as follows:

Line 281 of the Methods section: “*Given that the widespread associations between proteins and brain structure were obtained from observational data, there is always bias caused by confounding variables. By using genetic variants that are randomly assigned at conception, the MR analysis could minimize the bias and enhance the reliability of the results.*”

- - Though the results of Mendelian randomization are typically interpreted as ‘causal effect’, we agree that a more conservative conclusion would be more appropriate. As stated in Comment 2, we have carefully reviewed and refined the language throughout the manuscript to replace “causal effect or relationship” with “associations or MR associations”.

Comment 4: *In regards to the proteomics methodology and measurements, I believe the OLink platform and the QC data meets expected standards in the field. That said, the details required in regards to the proteomics data pre-processing is not sufficient. In the methods section the author's need to state exactly how the quality control was undertaken, and provide exact details*

on how the protein data was normalised. A major concern is how the missing data was handled. The authors report proteins with missing data of 50% or higher were imputed. The standard in the proteomics field is 70% or higher. 50% is low, can the authors justify. What was the percent of missing data and how much imputation was carried? Was there a cut off applied for the data imputation, how was imputation performed, and how do the authors know excessive imputation did not impact the proteins identified as differentially expressed? It is standard in proteomics to include all this information on data pre-processing, normalisation, imputation etc, in the methods and results.

Response: Thanks for your concern regarding the information on the processing of proteomic data. The information regarding how the quality control was undertaken and how the protein data was normalized have been thoroughly documented in the Supplementary Information in previous publication (Benjamin B. Sun et al., *Nature*, 2023), as well as on UKB official website (<https://biobank.ndph.ox.ac.uk/showcase/refer.cgi?id=4658>). Following previous publications, we have incorporated the information as follows.

Line 23 of the Supplementary materials: *“The counts of known sequences were converted into Normalized Protein eXpression (NPX) values, which were derived through within-batch and across-batch normalization, using Olink’s MyData Cloud Software. Within batch normalization centers data at NPX=0 by subtracting the plate-specific median per assay from all samples and assays in the same plate. Across-batch normalization calculates adjustment factors by determining the difference in assay-specific median NPX values for each batch. This process involves two steps: the first addresses plate-to-plate variation within a batch, while the second accounts for batch-to-batch variation across the study. Both steps involve shifting by an assay-specific fixed factor on the NPX scale: the plate median in the first step and the difference between assay-specific medians across batches in the second step.*

The Olink workflow includes an inbuilt quality control system consisting of three engineered internal controls that are spiked into every sample and each abundance block. Olink’s internal QC assessment is performed at two levels; run QC and sample QC. For run QC, each abundance block per panel and sample plate should fulfil the mean absolute deviation (MAD) in both internal controls (Inc Ctrl and Amp Ctrl) which should not exceed 0.3 NPX, the deviation of sample QC level is allowed for up to 1/6 samples and in each panel the median of 90% assays in plate and negative controls should be in the accepted range from predefined values set during validation. The sample QC evaluates each sample individually using the internal controls (Inc Ctrl and Amp Ctrl), which should fall within ± 0.3 NPX of the plate median across the abundance block. Additionally, the mean assay count for a sample must not be less than 500 counts. Samples that do not meet these criteria will receive a warning for the corresponding abundance block in the dataset.

Outliers were identified using two approaches applied to each protein panel: (1) principal component analysis (PCA), and (2) examining the median and IQR of NPX across proteins by sample. Data points were removed if (1) a standardized PC1 (the component that captures the most variation) or PC2 (second largest component) value more than 5 standard deviations from the mean (which is zero in standardized PCA), or (2) a median NPX greater than 5 standard deviations from the mean median, or an IQR of NPX greater than 5 standard

deviations from the mean IQR. We excluded outliers, data points with a QC or assay warning, and likely sample swaps, removing the sample across all panels if half or more of the panels were affected; the remaining data contained 56,695 samples and 52,790 individuals. Suspected sample swaps were identified by examining discrepancies between the proteomic-predicted sex and outliers from cis pQTLs, where the standardized squared residuals for all proteins were summed for each individual and divided by the sum of squared protein levels. Samples with incorrect genotypes were expected to show larger values than those with correct genotypes.”

- - As for how the missing data was handled, we have checked the missing rate of our proteomic data. As shown in the histogram below, all proteins showed a missing rate of less than 25%, which satisfy the standard cutoff you mentioned in the field, such that the same set of proteins were included under the new cutoff. The remaining missing values were imputed based on median value of each protein. To further examine whether imputation would impact the identified proteins, we tried to conduct a complementary association analysis without performing imputation. For each protein, the missing values were removed and the associations with brain structure measurements were examined. As shown in the table below, the pattern of significant associations without imputation were quite similar to that of using imputation. In summary, both the missing rate cutoff and the imputation methods are in line with the common practice in this field, and the results are robust to the performing of imputation. The details of imputation have been clarified in the Methods section, and the results of the sensitivity analysis without imputation have been listed in Supplementary Table 6.

Line 152 of the Methods: *“Proteins with a missing rate of 30% or higher were excluded (GLIPR1, NPM1, and PCOLCE) and the remaining missing values were imputed based on median values.”*

Line 249 of the Methods: *“we conducted an additional association analysis without performing an imputation of protein expression.”*

Line 419 of the Results: *“The imputation procedure did not affect the pattern of significant associations between protein and brain structure (see Supplementary Table 6).”*

Modality	Imputation	No imputation	R
Volume	1,370	1,275	0.99
Area	719	648	0.99
Thickness	682	586	0.99
FA	86	80	0.99
MD	2,501	2,353	0.99

The table compares the number of significant associations and correlation coefficient of protein-brain structure associations between different imputation strategies.

Comment 5: *The raw Olink data was not uploaded/provided. There is a link to github.com but the actual proteomic raw data is not provided. The data file uploaded just contains the -protein and -accession number but the plasma protein expression values/assay results are not provided.*

Response: Thank you for your helpful feedback regarding the data availability. We recognize the importance of providing comprehensive data for reproducibility and transparency. The raw Olink data used in our study was acquired from the UK Biobank, as described in the Methods section. As part of the UK Biobank policies, we are not allowed to provide actual proteomic raw data elsewhere. The Olink data is publicly available on the UK Biobank website upon application for access (<https://www.ukbiobank.ac.uk/>).

Comment 6: *In the ‘association analysis’ methods section the authors report on using ‘protein concentration, age at imaging visit, sex, years of education, Townsend index, 224 ethnicity, smoking and drinking status at imaging visit, interval between baseline and 225 imaging visit, scanning site, total intracranial volume all used as predictors of the structural measures.’ Its difficult to interpret how co-varying for all these measures impacted the data. Ideally they should summarise the effect before and after covariate analysis or present in a table. Elsewhere they report adjusting for ‘gender and age’. Can the authors clarify how the data was adjusted and provide a rationale. The sex effects would be very interesting and additional information whether particular brain measurements were associated with gender is important. If an effect of gender or age on brain measurement was previously found, then a sub analysis looking at associations between plasma protein expression and brain measurements in males and females should be considered.*

Response: Thanks for your insightful suggestion. We agree that presenting the effect before and after covariate analysis in the same table could help interpret the impact of these variables on the protein-brain structure associations. Following your suggestion, we have performed a supplementary association analysis with only protein expression, age at imaging visit, sex, total intracranial volume and scanning site as the predictor. These limited covariates were the well-known factors that influenced brain structure measurements. When included additional covariates (i.e., years of education, Townsend index, ethnicity, smoking and drinking status at imaging visit), the pattern of significant associations remained largely the same. Moreover, when comparing the effect of proteins before and after including additional covariates, we observed a correlation coefficient of 0.99 for GM volume, 0.99 for cortical area, 0.99 for

cortical thickness, 0.99 for WM FA and 0.99 for WM MD. To provide readers with a better understanding, we have added the comparison of statistics before and after including additional covariate in Supplementary Table 6. We have also included the corresponding description as follows:

Line 247 of the Methods: “...an additional association analysis with only protein expression, age at imaging visit, sex, total intracranial volume and scanning site as the predictors was performed to explore the influence of the covariates...”

Line 416 of the Results: “In sensitivity analyses, compared to associations with only protein expression, age at imaging visit, sex, total intracranial volume and scanning site as the predictors, the inclusion of additional covariates didn’t change the pattern of significant associations (Supplementary Table 6).”

- - Gender and age were regressed out from protein expression before the protein-brain structure association analysis because they exhibited high collinearity with the protein (Harris, S. E. et al. *Nature Communications*, 2020). In this way, we ruled out the possibility that the observed protein-brain structure associations were simply the effect of age and sex. We agree that protein-brain structure associations may exhibit sex-specific pattern. To examine this possibility, we conducted a supplementary analysis to perform association analysis in females and males separately. We found that the direction of effects was consistent between different sexes and the effects of protein expression were highly correlated between females and males. In particular, a correlation coefficient of 0.79 for GM volume, 0.71 for cortical area, 0.55 for cortical thickness, 0.60 for WM FA and 0.91 for WM MD. We have added the reason for regressing out gender and age before association and introduced the subgroup analysis:

Line 154 of the Methods section: “The effect of sex and age were regressed out from protein expression before the protein-brain structure association analysis because they exhibited high collinearity with the protein. In this way, we ruled out the possibility that the observed protein-brain structure associations were simply the effect of age and sex”.

Line 250 of the Methods: “Third, to examine potential sex difference, the association analysis was further conducted in female and male separately”.

Line 421 of the Results: “We found that the effects of protein expression were highly correlated between females and males. (Supplementary Table 6).”.

Comment 7: *Brain neuroimaging measurements are not clearly defined in the methods, Table 1 or figure legends. While these may be available on the UK biobank links or in previous studies they need to be clearly defined here to aid interpretation. This makes interpretation of the results challenging. In particular results in figure 2 are confusing. its not clear if the proteins in the volcano plots Fig2A are associated with TOTAL brain volume or the volume of SUB-REGIONS of the brain coloured in Figure 2B. This needs to be clarified and the same applies to proteins associated with thickness (is this cortical thickness?), brain area, FA, MD etc. in figure 2.*

Response: Thank you for this detailed suggestion. The associations referred to were between proteins and regional measurements (specific region or tract) rather than overall measurements

(total volume or mean cortical thickness). Following your suggestion, we have ensured that all neuroimaging measurements are clearly defined:

Line 240 of the Methods section: *“A generalized linear model was used to investigate the association between plasma protein expression and imaging derived measures of brain structure, including regional GM volume, area, and thickness, as well as FA and MD of WM tracts”*.

- - As for volume metric in Figure 2A, each dot represents the most significant association between the specific protein and all volume metrics in Figure 2B. The same is true for the other four structural metrics (i.e., area, thickness, FA, and MD). For example, the NCAN dot in volume panel of Fig 2A represents the most significant association between NCAN and all possible GM volume measurements in Fig 2B. Following your suggestion, we have revised the legend of Figure 2 to make it clearer.

Line 1039 of Figure 2 legend: *“(A) For each modality, each dot represents the most significant association between the specific protein and all corresponding metrics (i.e., regional or tract-wise) in Figure 2B.”*.

Comment 8: *The rationale for using the Genetic data and GWAS data is not clear and it complicated the interpretation of the results. To aid interpretation the authors state the research question in the introduction section. The data is not mentioned until it shows up in methods in the ‘genetics data’ on page 7 of methods, and the ‘GWAS summary’ on page 8. Its not clear how this data was linked with the proteomics data until the discussion section. This made revision the manuscript particularly difficult given that its 36 pages, excluding all the supplementary info.*

Response: Thank you for this valuable suggestion. Following your suggestion, we have revised the manuscript to clarify the rationale for using genetic data and GWAS data. Specifically, we have explicitly explained how the genetic and GWAS data contribute to the study and have added the related statement in the Introduction and Methods section:

Line 107 of the Introduction: *“The absence of such data, including genetic information, made it challenging to systematically explore the intricate relationships between plasma proteins and brain structure. The recent availability of large-scale datasets makes it feasible to establish robust links between genetic regulation of proteins and that of brain structure, opening new avenues for research into the pathogenesis of neurodegenerative and neuropsychiatric disorders.*

The database that has been analyzed refers to 4,997 participants in the UK Biobank (UKB) for whom plasma proteomic and neuroimaging, and genetic data were available...Utilizing genetic data, the bidirectional Mendelian randomization analysis was performed to enhance the potential associations between proteins and brain structure, as well as between proteins and brain disorders...”

Line 174 of the Methods: *“With regard to genetics, the imputed genetic dataset released by UKB in July 2017 was used for MR analysis in the present study.”*

Line 193 of the Methods: *“To conduct Mendelian randomization analysis between protein and disorders, we curated GWAS summary statistics of brain disorders.”*

Comment 9: *Crucial data on the study population is not provided. From the 4,997 participants – what is the gender distribution? This is not provided in Table 1 or supplementary fig 1. What proportion of these had mental health issues? What proportion of these had brain disorders such as ASD, ADHD, AD etc? Plasma proteomic associations are made with these outcomes ye the number of subjects in each category are not presented in the manuscript. This lack of information prohibits interpretation of the validity of the findings and robustness of the data in regards to mental health and disease outcomes.*

Response: We apologize for the confusion we have caused. As shown in Table 1, there is slight difference in the number of participants for different structural metric. The number 4,997 referred to the maximum number of participants used for brain structural metric. i.e., the number of participants included for area and thickness metric. There were 2,642 females (52.9%) and 2,355 males (47.1%).

Line 384 of the Results: “*...in five categories in 4,900 participants (2,642 females and 2,355 males).*”

- - As for your concern about the proportion of participants that had brain health issues, we would like to clarify this in each analysis. For the Mendelian randomization analysis characterizing the associations between proteins and brain disorders, the outcomes were not derived from the UKB participants but from publicly available independent cohorts (i.e., PGC and FinnGen). The reason is that the Mendelian randomization analysis requires no sample overlap between exposure (i.e., proteins) and outcomes (i.e., brain disorders). For mediation analysis, the continuous mental health phenotypes rather than binary diagnoses were used as outcomes, because there are limited number of participants with clinical diagnoses.

To address your concerns, we have also provided the number of patients in each structural category in Supplementary Table 5. Specifically, we only observed few patients diagnosed with the diseases. In summary, the number of patients in UKB would not influence the robustness of the results.

Line 385 of the Results: “*Only few of the included patients were diagnosed with the brain diseases (Supplementary Table 5).*”.

Disease	Volume	Area	Thickness	FA	MD
Anxiety	49	51	51	47	47
BP	8	8	8	7	7
MDD	87	90	90	85	85
SCZ	2	2	2	1	1
AD	0	0	0	0	0
PD	0	0	0	0	0
MS	0	0	0	0	0

Comment 10: *The authors do not refer to the supplementary tables available. This should be added throughout the manuscript.*

Response: We apologize for not sufficiently referencing the supplementary tables in the manuscript. We have ensured to incorporate appropriate citations to all the supplementary

materials throughout the text. This will help enhance the understanding of our findings.

Minor corrections/suggestions to improve manuscript

Comment 11: *No table legends provided / definitions of brain area FA MD et. are missing.*

Response: We apologize for the oversight regarding the absence of table legends, such as the definitions for brain metrics FA and MD. We have revised all the supplementary tables to replace FA and MD with their full name and included detailed legend for Table 1 to ensure easier interpretation of acronyms in the corresponding tables.

Comment 12: *Running title is more transparent than the main title of manuscript, suggest this is adopted as main title.*

Response: Thanks for bringing this to our attention. We appreciate your feedback regarding the running title being more transparent than the main title. We have adopted the running title “*Plasma proteomic signature of brain structure*” as the main title of the manuscript to enhance clarity.

Comment 13: *Plasma proteins should be noted in the keywords*

Response: Thank you for your suggestion to include "plasma proteins" in the keywords. We agree that this term is crucial for accurately representing the focus of our study. We have included "plasma proteins" as the first keyword in line 47, enhancing the visibility and relevance of our research.

Comment 14: *There are too many acronyms which hinder interpretation.*

Response: We understand that excessive acronyms can hinder clarity and interpretation. We have reviewed the manuscript to remove some redundant acronyms, such as minimum minor allele frequency (MAF), T1 weighted (T1w), and diffusion weighted imaging (DWI), and ensured that the other acronyms are clearly defined upon first use, and added detailed legends for both tables and figures. We hope this solution could improve the clarity of our manuscript.

Comment 15: *The authors continuously report on ‘protein concentration’ and various associations but this I believe is incorrect. The authors report that differential expression of plasma proteins across 5000 subjects as opposed to using the absolute ‘concentration’ of a protein in place. Suggest replacing with ‘protein expression’ which is the norm in the field of proteomics.*

Response: Thank you for pointing this out. We appreciate your suggestion to use the term “protein expression” instead of “protein concentration”. We have revised the manuscript to replace "protein concentration" with "protein expression" throughout (e.g., line 155, 241, 287), ensuring that our language aligns with established conventions in the literature.

Reviewer #2 (Remarks to the Author):

Comment 1: *This manuscript analyzed the association between blood protein levels and brain morphology and explored their underlying biological and clinical implications. The key finding*

is the confirmation of a robust association between periphery and central nervous system, and highlighted different role of neural and immunological genes in brain morphology and the key roles of NCAM1 and MOG. I appreciate that this study is rigorous and valuable, but some issues should be clarified before publication.

Response: Thank you for your positive feedback and for recognizing the value of our study. Thank you for providing constructive suggestion to improve the overall quality of our work. We have revised the manuscript to address the issues you mentioned.

Comment 2: *My major concern is on how the observed association and causality (MR result) is interpreted. In most scenario, peripheral molecules is regarded as a proxy of central molecules, and the observed association could be simply interpreted as “peripheral changes could reflect CNS alterations”. This study, however, added MR analysis, and the result seemed to indicate that peripheral proteins had a direct causal effect on brain structure. This is quite surprising, and more validation is required. In fact, I believe that a more intuitive explanation is that brain morphology is genetically correlated to neuron-related protein expression, and that such genetic correlation drives false positive MR result. A suggestive evidence is that there is quite a lot of reverse causality (although not overlapped with forward causality): how could brain structure impacts blood gene expression? The genetic correlation model is apparently more intuitive here. I thus suggest the author to apply CAUSE to distinguish causality and genetic correlation. I hypothesis that there would be difference between neural and immunological genes: for the former, peripheral expression is simply a proxy of CNS expression; for the latter, peripheral expression represents an average expression across the body, and could have some causal roles. Similar difference may also exist in colocalization test.*

Response: Thank you for this professional suggestion. We agree that the interpretation of the causal effects identified by Mendelian randomization analysis needs careful consideration. Following your suggestion, we have performed CAUSE analysis to distinguish causality and genetic correlation. We found that almost no significant difference was found between genetic correlation model and causal model at FDR-corrected $P < 0.05$. However, at a clumping P threshold of 5×10^{-6} , 57 out of 71 protein-brain structure associations and 16 out of 21 protein-disease associations tend to be better characterized by causal model than genetic correlation models (i.e., $\text{delta_elpd} < 0$). Similarly, at a clumping P threshold of 5×10^{-8} , 39 out of 59 protein-brain structure associations and 27 out of 34 protein-disease tend to be better characterized by causal model than genetic correlation models. The results of CUASE analysis have been listed in Supplementary Table 8,10,16 and 18. The Methods and Results were added accordingly:

Line 315 of the Methods: *“To further control the false positive rate in the presence of pleiotropic effects, we performed Causal Analysis Using Summary Effect Estimates (CAUSE) analysis to distinguish causality and genetic correlation using cause R package. Similar to that of MR analysis, two clumping thresholds (i.e., 5×10^{-6} and 5×10^{-8}), were used, and the significance threshold was set to FDR-corrected $P < 0.05$.”*

Line 343 of the Methods: “*CAUSE analysis was also conducted to distinguish causality and genetic correlation.*”

Line 504 of the Results: “*To further control the false positive rate in the presence of pleiotropic effects, we performed CAUSE analysis to distinguish causality and genetic correlation. Almost no significant difference was found between genetic correlation model and causal model at FDR-corrected $P < 0.05$. However, at a clumping P threshold of 5×10^{-6} , 57 out of 71 protein-brain structure associations tended to be better characterized by causal model than genetic correlation models (i.e., $\text{delta_elpd} < 0$). Similarly, at a clumping P threshold of 5×10^{-8} , 39 out of 59 protein-brain structure associations tended to be better characterized by causal model than genetic correlation models (Supplementary Table 8 and 10).*”

Line 570 of the Results: “*To further control the false positive rate in the presence of pleiotropic effects, the CAUSE analysis found almost no significant difference between genetic correlation model and causal model at FDR-corrected $P < 0.05$. However, at a clumping P threshold of 5×10^{-6} , 16 out of 21 protein-disease associations tended to be better characterized by causal model than genetic correlation models (i.e., $\text{delta_elpd} < 0$). At a clumping P threshold of 5×10^{-8} , 27 out of 34 protein-disease associations tended to be better characterized by causal model than genetic correlation models (Supplementary Table 16 and 18).*”

In summary, since we didn't find significant difference in CAUSE analysis at FDR-corrected $P < 0.05$, we have replaced the use of “causal” with “associations” throughout the Introduction, Methods, Results and Discussions sections. Here are some examples:

Line 61 of the Abstract: “*Utilizing bidirectional Mendelian randomization (MR), 33 significant associations between 32 proteins and 23 measures of brain structure, and 21 associations between nine brain structure associated proteins (e.g. *BTN3A2*, *BTN2A1* and *ENPP6*) and ten neurodegenerative and psychiatric disorders were discovered.*”

Line 117 of the Introduction: “*Utilizing genetic data, the bidirectional Mendelian randomization analysis was performed to enhance the potential associations between proteins and brain structure...*”

Line 453 of the Results: “*Associations between proteins and brain structure identified through MR analysis*”

Line 460 of the Results: “*Notably, the cingulate gyrus part of cingulum, parahippocampal part of cingulum and thalamic radiation were the tracts that exhibited the most MR associations with proteins*”

Line 617 of the Discussion: “*The proteins exhibited significant associations with brain structure measures also demonstrated associations with the risk of neurodegenerative and psychiatric disorders.*”

Comment 3: *Furthermore, I'm very confused on the study design of colocalization test:*

- 1) What does CADD have to do here? I've never seen CADD analysis on common SNP.*
- 2) Shouldn't colocalization be applied to eQTL and pQTL of the same gene? Does “protein-gene pair” indicate that this analysis involves nearby eQTL of other genes? If such colocalization happens, it means that peripheral protein expression is actually tagging different proteins in the CNS, and that many biological interpretations based on peripheral*

signals are flawed.

3) MR is based on genome-wide pQTL, whereas colocalization test is based on cis-pQTL (the method section did not clarify this, but the result indicate so). Why not use methods like SMR that include only cis QTL?

4) It is well-known that QTL of difference tissues could have large proportion of colocalization, so what is the specific conclusion of this paper? Which blood protein shared regulation mechanism with CNS and which is actually tagging another CNS protein? Simply provide the overall result does not step forward from known knowledge.

Response: Thanks for raising this important point. We have made revisions accordingly.

1) The CADD score in FUMA is computed for GWAS summary data based on 63 annotations. The higher the score, the more deleterious the SNP is. Following your suggestion, we have removed the use of CADD and FUMA in our analysis.

2) We agree that the colocalization analysis of pQTL and eQTL should be performed on the same gene. The new colocalization analysis on the same gene showed that the pQTL of the proteins with significant MR-identified associations share the same genetic variants with the eQTL of their protein-coding gene in the brain cortex. Specifically, BTN3A2, LRRC37A2, and SULT1A1 showed strong evidence of colocalization ($PP.H4 > 0.8$), BLMH, CCL28, HEXIM1, LEPR, and REN showed medium evidence of colocalization ($0.5 < PP.H4 < 0.8$).

3) Following your suggestion, since the colocalization was performed based on cis-pQTL and cis-eQTL, we have performed SMR analysis to test for pleiotropic association between the protein expression and gene expression. We found that SULT1A1, REN, AGER, F11R, LEPR, and PAMR1 in the plasma exhibited pleiotropic association with the expression of their protein-coding gene in the brain ($P_{SMR} \leq 0.05/32$ and $P_{HEIDI} \geq 0.05$).

4) The pQTL with significant SMR associations with eQTL means that the blood protein shares regulatory mechanism with their protein-coding genes in the brain, otherwise, the blood protein may be tagging another brain proteins. Therefore, we found that SULT1A1, REN, AGER, F11R, LEPR, and PAMR1 had high possibility to share regulatory mechanism with their protein-coding gene in the brain and the other proteins tended to be tagging another CNS protein.

In summary, following your suggestion, we have replaced the colocalization analysis with SMR analysis, and emphasize the specific finding of this study.

Line 320 of the Methods: “*Pleiotropic association between plasma pQTL and brain eQTL*”

Line 327 of the Methods: “*Specifically, for each brain structure-associated protein surviving MR analysis, we used the SMR method⁴² to integrate summary-level pQTL data with eQTL data to identify genes whose protein expression in plasma are associated with their expression levels in the brain because of pleiotropy. The heterogeneity in dependent instruments (HEIDI) test was conducted and the significance threshold was set to $P_{SMR} < 0.05/32$ and $P_{HEIDI} > 0.05$.*”

Line 512 of the Results: “*Pleiotropic association between plasma proteins and brain gene expression*”

Line 521 of the Results: “*To test for pleiotropic association between the protein expression and gene expression. The SMR analysis found that SULT1A1, REN, AGER, F11R, LEPR, and PAMR1 in the plasma exhibited pleiotropic association with the expression of their protein-*

coding gene in the brain ($P_{SMR} \leq 0.05/32$ and $P_{HEIDI} \geq 0.05$). In other words, these plasma proteins shared regulation mechanism with the brain.”

There are some technical issues to be revised:

Comment 4: *Is the protein-MRI association p value well-calibrated? Is there any inflation?*

Response: Thank you for highlighting this issue. In the original results, the reported P values of the protein-MR associations have been corrected with FDR. Following your suggestion, we have checked the distribution of original P values with QQ plot and lambda. We observed that the protein-MR association P values showed little inflation ($\lambda = 1.3$). This may occur because the proteins are highly correlated with one another, and the brain measures also exhibit correlations. To be rigorous, we have added this point as a limitation.

Line 784 of Discussions: *“Fifthly, As the proteins are highly correlated with one another, and the brain measures also exhibit correlations, the significance of protein-MR associations presented little inflation. Although we have performed multiple comparison corrections and validated the results with MR and mediation analyses, future independent validation could enhance the discovered associations between proteins and brain structures.”*

Comment 5: *On figure: What’s the meaning of drawing a line from data point to the y-axis? For GO result, consider using bubble plot to visualize the result, where point size or color showed the fold enrichment.*

Response: We appreciate your suggestion about the visual representation of the data. The line drawn from the data point to the y-axis in the lollipop plot was intended to illustrate the relationship between the measurements; however, we understand that it may not effectively convey the information. We have taken your recommendation to replace Figure 3D with a more informative visualization, i.e., bubble plot, where point size and color can represent fold enrichment for the GO results.

Comment 6: *Compared with other participants without MRI, proteome, or both, does the included participants showed significant difference in demographic variables?*

Response: Thank you for bringing up this important point. To give a clearer context for our results, we have conducted several analyses to assess whether there are significant differences in demographic variables (e.g., age and sex) between these groups. As shown in the table below, by comparing the demographic variables of the included participants with those of other participants at baseline visit, we found that the mean age of the included participants at baseline visit (54.1 ± 7.8) was significantly lower than that of participants without MRI ($P < 0.001$), those without proteome ($P < 0.001$), and those without both ($P < 0.001$) at baseline. The sex distribution of the included participants showed significant differences compared to participants without MRI ($P = 0.004$), those without proteome ($P = 0.01$), and those without both ($P = 0.003$). However, although there were significant differences, the actual magnitude is small. We believe the included participants is representative of the entire UKB cohort. Moreover, validation with future releases of proteome-neuroimaging data could strengthen our findings. To be rigorous, we have incorporated this point as a limitation:

Line 780 of the Discussion: *“Fourth, the included participants who underwent both the imaging visit and proteomic analysis exhibited slight difference in age and sex compared to the other UKB participants. Although we have partially controlled the effects of age and sex through regression, validation with future releases of proteome-neuroimaging data covering a broader range could strengthen our findings”*.

	Included N=4997	Without MRI N=432647	Without proteome N=434940	Without both N=387904
Age	54.1 (7.8)	56.7 (8.1)	56.5 (8.1)	56.6 (8.1)
Sex:				
Female	2642(52.9%)	237367(54.9%)	237351(54.6%)	213064(54.9%)
Male	2355(47.1%)	195280(45.1%)	197589(45.4%)	174840(45.1%)

Comment 7: *I believe that merely the most straightforward conclusion “blood proteins are associated with CNS alteration” is quite important for the field. This simple conclusion could be more highlighted in the manuscript.*

Response: Thank you for giving this constructive suggestion. We agree that this conclusion can enhance the overall impact of this study and we have ensured that it is more prominently highlighted in the manuscript. We have revised the corresponding description as follows:

Line 790 in the Discussion: *“In conclusion, utilizing the unique opportunity provided by large-scale genetic, proteomic and neuroimaging data, this study represents the first comprehensive atlas of the proteomic signature of human brain structure. Our findings underscore the widespread associations between plasma proteins and structural changes in the brain, revealing how plasma protein may reflect alterations in the central nervous system. The identified proteins exhibited associations with a wide range of neurodegenerative and neuropsychological disorders, highlighting the potential of plasma proteins as critical*

biomarkers for assessing brain health. Furthermore, brain structure was found to play a mediating role in the relationship between plasma proteins and brain health. By uncovering these associations, this research bridges the gap between plasma protein markers, brain structure and brain disorders, offering an opportunity to unravel the mechanisms underlying brain disorders and to identify potential therapeutic targets.”.

Comment 8: *On MR analysis: usually the additional MR methods (weighted mode, weighted median) would be applied regardless of sensitivity analysis. All MR methods give consistent significant result is more persuasive.*

Response: Thank you for pointing this out. We agree that applying additional methods, such as weighted mode and weighted median, can strengthen our findings and enhance the robustness of our results. Following your suggestion, we have performed additional MR methods for all examined MR relationships. As shown in the table below, at both clumping thresholds, most of the significant MR relationships remained significant when analyzed using at least one of the other MR methods. We have listed the results of other MR methods in Supplementary Table 8-23 and revised the corresponding description as follows:

Line 302 of the Methods: *“Additionally, we conducted another four MR methods: MR-Egger, weighted median, weighted mode and Wald ratio, with the former three to complement the potential bias in the IVW results, and the last one when only one genetic instrument was valid. All these methods were implemented with the corresponding function in TwoSampleMR R package.”*

Line 477, 496 of the Results: *“We found that most of the significant protein-brain structure remained significant when analyzed using at least one of the other MR methods”.*

MR relationship	5×10^{-6}	5×10^{-8}
Protein-brain structure	31/33	39/42
Brain structure-protein	39/42	85/95
Protein-disease	19/21	30/34
Disease-protein	17/19	19/22

The number of significant MR relationships is shown in the format of other MR methods/IVW method.

Reviewer #3 and #4 (Remarks to the Author):

OVERALL COMMENTS

Comment 1: *Overall, while the manuscript presents a valuable/noteworthy set of results, there are several key areas that need further attention. A main concern is the minimal use of references across the Introduction, Methods, and Discussion sections, given that up to 70 refs are allowed.*

Response: We appreciate your acknowledgment of the value of our results. We recognize the importance of incorporating more relevant references to support our claims and provide a more robust context for our findings. We have conducted a thorough review of the Introduction, Methods, and Discussion sections to include additional references, ensuring that our manuscript is well-supported by existing literature.

Comment 2: *In the Introduction, the rationale for the study could be expanded to better highlight key gaps in the field. The section would benefit from additional references that provide more robust context and emphasize the significance of addressing these gaps, helping to frame the study's objectives more clearly.*

Response: Thank you for this valuable suggestion. We have revised the introduction to better highlight the key gaps in the field and expanded the rationale for the study. Additional references have been incorporated to provide a more robust context, emphasizing the importance of addressing these gaps and framing the study's objectives more clearly. These revisions can be found as follows:

Line 99 of the Introduction section: “*They do not fully capture the comprehensive and intricate connections between plasma proteins and various aspects of brain structure, leaving significant gaps in the understanding of the underlying biological mechanisms.*”.

Comment 3: *In the Methods section, while the approach taken is logical and sound, several methodological decisions are presented without adequate citation of relevant literature. Including references for key procedures and criteria would enhance the transparency and reproducibility of the study. This is particularly important for justifying the thresholds used in genetic analyses and other statistical methods.*

Response: Thanks for your suggestion on the transparency and reproducibility of the Methods section. To enhance it, we have incorporated relative citations to support the methodological decisions, and provided specific references that justify the thresholds used in our genetic analyses and statistical methods. Detailed revisions could be found in the responses to SPECIFIC COMMENTS below.

Comment 4: *In the Discussion, the results are mostly summarized but lack deeper reflection on their implications. It would be beneficial to explore how these findings address existing gaps and to provide a clearer explanation of the complementary insights offered by the results, demonstrating how they collectively advance our understanding of the topic.*

Response: Thank you for this important insight. We appreciate your suggestion to deepen the interpretation of the results and their implications. Following your suggestion, we have provided a clearer explanation of how our findings complement the existing gaps and emphasized the contribution of our research to the field of proteome and neuroimaging. Detailed revisions could be found in the responses to SPECIFIC COMMENTS below.

Comment 5: *Additionally, there are some grammatical inconsistencies throughout the text that affect its clarity and readability. A thorough review and revision would greatly enhance the manuscript's overall presentation and ensure a smoother reading experience.*

Response: We appreciate your attention to details and we have conducted a thorough review to address these issues. By checking the grammar with both native speakers and Grammarly, we are committed to ensuring a smoother reading experience and a better presentation for the audience.

----- SPECIFIC COMMENTS -----

METHODS

Comment 6: *The rationale for selecting five imaging-derived phenotypes (IDPs)—specifically grey matter volume, surface area, and cortical thickness, along with white matter fractional anisotropy (FA) and mean diffusivity (MD)—from the multiple IDPs released by the UK Biobank is not clearly explained. The authors should provide a clear justification for this selection.*

Response: Thank you for pointing this out. As we are only interested in brain structures that could be stably measured, the functional IDPs in UKB are beyond the scope of this manuscript. To provide a clearer rationale for selecting the five imaging-derived phenotypes (IDPs) in our study, we have revised the manuscript to include a detailed justification for our choices, explaining how each IDP contributes to our understanding of brain structure in the context of our research objectives:

Line 160 of the Methods section: *“Given that gray and white matter are crucial components that serve distinct functions for overall brain health, here we included brain structural imaging derived phenotypes (IDPs) that were previously made available by the UKB neuroimaging group¹⁹. Similar to previous researches, regional grey matter measures (volume, thickness, and area) derived from T1-weighted images and microstructural measures derived from diffusion weighted imaging, including fractional anisotropy (FA) and mean diffusivity (MD), were chosen to be used in the present study²¹.”*

Comment 7: *The authors should cite major UK Biobank publications that introduced imaging-derived phenotypes (IDPs) and proteomics.*

Response: We apologize for missing the major UKB publication that introduced IDPs and proteomics. We have now added the corresponding citations as follows:

Line 162 for IDPs: *(Thomas J. Littlejohns et al., Nature Communications, 2020)*

Line 151 for proteomics: *(Benjamin B. Sun et al., Nature, 2023)*

Comment 8: *In the "Genetic Data Processing and Analysis" section, several key concepts and decisions lack citations. Relevant references should be added to support these points. For example, the exclusion criteria for single nucleotide polymorphisms (SNPs)—those with a missing rate > 5%, minimum minor allele frequency (MAF) < 0.1%, or Hardy–Weinberg equilibrium test $P < 1 \times 10^{-10}$ —should be backed by appropriate references or a more detailed explanation, as these decisions can significantly influence the robustness and generalizability of the findings.*

Response: We apologize for missing the key citations supporting the key methodology decisions. According to your suggestions, we have added the appropriate references as follows: Line 178 of the Methods section: *(Clare Bycroft et al., Nature, 2018; Yuzhu Li et al., Nature Aging, 2022).*

Comment 9: *In the “GWAS summary of CNS disorders,” GWAS references are missing for the neurodegenerative diseases. Please provide. Also, include a rationale for choosing these*

four neurodegenerative disorders along with the sample sizes for these studies in the main document (in addition to the range of N).

Response: We apologize for not including the introduction of the four neurodegenerative diseases. We have now added the rationale and relevant GWAS references for the selected neurodegenerative diseases as follows:

Line 194 of the Methods section: *“Given their significance in public health and the relatively large number of cases, GWAS summary statistics were collected from publicly available databases for six psychiatric disorders (attention deficit hyperactivity disorder (ADHD, cases/total=19,099/53,293), anxiety (cases/total=7,016/18,186), autism spectrum disorder (ASD, cases/total=18,381/46,350), bipolar disorder (BP, cases/total=20,352/51,710), major depressive disorder (MDD, cases/total=45,396/142,646), schizophrenia (SCZ, cases/total=67,390/161,405)), and four neurodegenerative disorders (Mitja I. Kurki et al., Nature, 2023) (Alzheimer’s disease (AD, cases/total=10,520/401,661), Parkinson’s disease (PD, cases/total=4,681/407,500), multiple sclerosis (MS, cases/total=2,409/408,561) and amyotrophic lateral sclerosis (ALS, cases/total=531/184,000))”.*

Comment 10: *Please provide citations for key methodological decisions used in the "Causal relationship between protein concentration and brain structure" and 'eQTL mapping and colocalization' sections. For example: MR-Egger analysis (line 277), leave-one-out-analysis (line 275) etc.*

Response: Thank you for raising these issues. We recognize that detailed citation for key methodologies could strengthen the validity of our manuscript. According to your suggestions, we have added citations for MR-Egger analysis in line 312 (Jack Bowden et al., *International Journal of Epidemiology*, 2015), leave-one-out-analysis in line 309 (Gibran Hemani et al., *Elife*, 2018). According to the 2rd reviewer’s suggestion, we have replaced the colocalization analysis with Summary-based Mendelian Randomization (SMR) analysis. Therefore, we did not include citation for colocalization but we added citation for SMR in line 328 (Zhihong Zhu et al., *Nature Genetics*, 2016).

Comment 11: *Provide a definition or brief explanation of "colocalization analysis" in the "Gene Expression and Quantitative Trait Loci" section for clarity (either in the main or supplementary documents).*

Response: Thanks for your suggestion. According to the 2rd reviewer’s suggestion, we have replaced the colocalization analysis with Summary-based Mendelian Randomization (SMR) analysis. Therefore, following your suggestion, we have added explanation of “SMR analysis”. Line 327 of the Methods: *“Specifically, for each brain structure-associated protein surviving MR analysis, we used the SMR method⁴² to integrate summary-level pQTL data with eQTL data to identify genes whose protein expression in plasma are associated with their expression levels in the brain because of pleiotropy. The heterogeneity in dependent instruments (HEIDI) test was conducted and the significance threshold was set to $P_{SMR} < 0.05/32$ and $P_{HEIDI} > 0.05$.”*

Comment 12: *For improved readability, please present the “Statistical analyses” subsections in the same order as they appear in the Results section.*

Response: Thank you for your suggestion. We have reorganized the "Statistical Analyses" subsections to align with the order in which they appear in the Results section. We hope this adjustment could enhance readability and make it easier for readers to follow the analyses.

Comment 13: *Clarify whether the physical annotation method, which calculates the density of proteins associated with brain structure on each chromosome, is based on a previously published reference or an in-house method.*

Response: We apologize for missing the methodology details of the physical annotation. The mapping of the protein-coding genes to chromosome positions was based on a previously established *topr* R software (Thorhildur Juliusdottir, *BMC Bioinformatics*, 2023). The definition of the density used an in-house method that dividing the number of protein-coding genes of each chromosome by the length of each chromosome. We have incorporated detailed descriptions:

Line 257 of the Methods section: *“The physical annotation was performed by first mapping the protein-coding genes to chromosome positions with a previously established topR software³⁴. Subsequently, the number of protein-coding genes on each chromosome was divided by the length of that chromosome, which generated the density of associated proteins in each chromosome that accounted for the length of the chromosome.”*

Comment 14: *Include a concise definition or explanation of horizontal and directional pleiotropy in the "Causal relationship between protein concentration and brain structure" section.*

Response: We apologize for ignoring the definition of horizontal and directional pleiotropy. To give a better understanding of the Mendelian randomization results, we have incorporated the definitions in the Methods section:

Line 309 of the Methods section: *“Secondly, the MR-PRESSO technique was used to detect possible horizontal pleiotropy⁴⁰, when a genetic variant influences the brain structure (directly or indirectly, through other traits) independently of the hypothesized protein. Thirdly, MR-Egger regression analysis was used to detect possible directional pleiotropy⁴¹, when a genetic variant associated with protein levels influences brain structure through a shared pathway rather than a direct effect of the protein on the brain structure.”*

Comment 15: *I also suggest reconsidering the subheading to explicitly mention both eQTL and pQTL for clarity. It would be good to have these concepts defined concisely in supplementals.*

Response: Thank you for this helpful comment. We have revised the subheading “Colocalization of plasma proteins and brain gene expression” to “Pleiotropic association between plasma pQTL and brain eQTL” in line 320 of the Methods section. Meanwhile, we have added the brief explanation of eQTL and pQTL in the Methods:

Line 210 of the Methods: “*Here we used pQTL and eQTL summary data to characterize genetic variants of protein and gene expression separately. A “pQTL” stands for “protein quantitative trait locus,” which refers to a genetic variant associated with the abundance of a specific protein, while an “eQTL” stands for “expression quantitative trait locus,” indicating a genetic variant linked to the expression level of a gene (mRNA transcript) at a specific locus on the chromosome.*”.

Comment 16: *Please provide N of the UKB subset used in the “Mediation analyses” section of the main document. Additionally, could the authors provide a rationale or reference for the chosen correlation cutoff of 0.55 used to identify and remove high collinearity between protein-associated structure measures in the iterative strategy?*

Response: Thanks for your professional suggestions. We agree that the demographical information of UKB subset used in Mediation analysis should not only be provided in Supplementary Table 4 but also be briefly stated in the Methods section. Accordingly, we have added the statement as follows:

Line 346: “*A total of 3,270 participants with a mean age of 54.2 were eligible for inclusion in mediation analysis (Supplementary Table 4)*”.

- - The correlation cutoff of 0.55 was selected because it can ensure the weights of all surviving observational variables contributing to latent brain structures fall within the range [-1,1], while minimizing the removal of observational brain structure measurements. We have included the rational as follows:

Line 357: “*By requiring the weights of all surviving observational variables contributing to latent structural measures fell within the range [-1,1], while minimizing the removal of observational brain structure measurements, a correlation threshold of 0.55 was chosen*”.

DISCUSSION

Comment 17: *The discussion section requires significant improvements. Much of the current text simply restates the results, with minimal interpretation of their significance or connection to existing literature, as reflected by the sparse citations. This approach significantly underutilizes the potential of the discussion, missing a critical opportunity to position this study as “the first comprehensive atlas of the proteomic signature of human brain structure.” While I include a few illustrative examples below, these are only a starting point, and a more thorough, in-depth discussion is needed throughout.*

Response: Thank you for providing these valuable suggestions. We agree that a more thorough and in-depth discussion of our findings could enhance the scientific value of this manuscript. By taking your suggestions below, we have enhanced the discussion by providing a more comprehensive interpretation of our findings, and integrating relevant citations. We hope our revisions could strengthen the contribution of this study to the field of proteome and neuroimaging.

Comment 18: *Example 1: The statement regarding the highest number of significant associations observed for WM MD (2,501), followed by GM volume, surface area, thickness,*

and the fewest for WM FA (86), is clear in terms of the ranking (results section, lines 362-364). However, it would greatly benefit from additional context and interpretation. Could the authors provide an explanation for why WM MD showed the highest number of associations, and why WM FA had the least? A discussion on the potential biological or methodological reasons behind these differences would enhance the reader's understanding and strengthen the results' interpretation.

Response: Thank you for your helpful comment. We have revised the Discussion sections to provide additional context and interpretation regarding the observed differences in the number of significant associations for WM MD and FA:

Lines 637 of the Discussion section: *“Though MD and FA are both metrics that reflect the microstructural integrity of white matter, MD demonstrated the most significant associations with proteins, whereas FA showed the least significant associations. The higher number of associations between MD and CSF proteins might stem from the fact that CSF proteins reflect early pathophysiological changes in the brain, to which MD is sensitive, especially in the early stages of neurological diseases such as AD⁴⁵. In patients with MS, MD was more strongly associated with widespread brain degeneration than FA as well⁴⁶. FA is primarily influenced by directional water diffusion and is more specific to structural coherence and fiber integrity⁴⁷. This specificity may make FA less sensitive to diffuse or generalized changes and more reliant on pronounced structural disruptions.”.*

Comment 19: *Example 2: Six proteins (NCAN, SLITRK1, MOG, LEP, OXT, PAEP) were highlighted in lines 549-552 for their numerous associations with brain structure. However, there is no discussion of their established roles in the brain, apart from a brief mention that NCAN is linked to brain volume. Expanding on the relevance of the top five proteins showing positive and negative associations to brain structure (mentioned in results section, lines 369-372) in a cohesive manner would enhance the discussion: NCAN, SLITRK1, MOG, PTPRN2 and SEZ6L; LEP, OXT, PAEP, CCN5 and XG.*

Response: Thank you for your valuable feedback. To provide a more comprehensive exploration of the established roles of the top proteins showing positive and negative associations with brain structure, we have added the description of the proteins:

Line 647 of the Discussion *“NCAN, SLITRK1, MOG, PTPRN2 and SEZ6L were the top proteins that showed the highest total number of significant positive associations with brain structures. The links between these proteins and brain disorders have been reported in previous studies. NCAN, an extracellular matrix protein, is closely related to the development, neuronal senescence and apoptosis of the neurons. NCAN has been found to be associated with neuropsychiatric and neurodegenerative diseases⁴⁸. In addition, NCAN demonstrated a predominant correlation with AD pathology⁴⁹. Existing evidence has linked the plasma concentration of NCAN protein to total brain volume¹⁸. The findings of the present study have expanded the significant associations of NCAN protein to the volume of 50 regions, surface area of 43 regions, FA of three tracts and MD of five tracts. Apart from NCAN, the other proteins are also relevant to central neural system and brain diseases. SLITRK1 function is associated with the control of neurite development and synaptogenesis⁵⁰. SLITRK has been*

linked to Tourette's syndrome, trichotillomania, and obsessive-compulsive disorder⁵¹. MOG, expressed in oligodendrocytes in the central nervous system, is an encephalitogenic protein that can trigger a demyelinating immune response⁵². Moreover, PTPRN2 has been implicated in AD and FTD⁴⁹, while SEZ6L has been associated with bipolar disorder⁵³. The proteins with significant negative associations with brain structure include LEP, OXT, PAEP, CCN5, and XG. LEP is well-known for its role in metabolic regulation and has been implicated in neurodegenerative diseases, potentially linking metabolic dysfunction to brain pathology⁵⁴. OXT is associated with social behaviors and stress responses⁵⁵. It has also been implicated in anxiety, depression and AD⁵⁶, although its relationship with brain structure remains unclear. Previous studies have highlighted associations between these proteins and CNS or brain disorders, and our findings further reveal their associations with brain structure. While the roles of PAEP, CCN5, and XG in brain diseases are not yet fully understood, our study identifies significant associations between these proteins and brain structure, offering a valuable basis for future studies.”

Comment 20: *Example 3: The investigation into the chromosomal locations of brain structure-associated proteins lacks context. The results (Results section, lines 383-385) show a concentration on chromosomes 19, 17, and 22, yet no explanation is provided for why this might be significant. A discussion on the potential links of these chromosomes to brain diseases would add valuable context to these findings.*

Response: Thanks for your suggestion. We have included an expanded discussion on the significance of the observed concentration of these proteins on chromosomes 19, 17, and 22, highlighting their potential links to brain diseases:

Line 687 of the Discussion: *“Our analysis revealed that brain structure-associated proteins are disproportionately localized to chromosomes 19, 17, and 22, relative to the relatively short length of these chromosomes. The concentration of brain structure-associated proteins on chromosomes 19, 17, and 22 is particularly noteworthy given the established relevance of these chromosomes to brain health and disease. Our findings show that many of the proteins identified on these chromosomes, including those encoded on chromosome 17 (OMG⁶⁰, BAIAP2⁶¹, GFAP⁶²), chromosome 19 (NCAN⁴⁹, APLP1⁶³, GDF15⁶⁴), and chromosome 22 (NPTXR⁶⁵, OSM⁶⁶, RTN4R⁶⁷) are strongly associated with brain structure and have been previously implicated in various neurological and psychiatric disorders. These results suggest that these chromosomes may serve as hotspots for genes influencing brain morphology and function, reflecting their broader role in neurodevelopmental and disease pathways.”*

Comment 21: *Example 4: The findings that the cingulate gyrus part of the cingulum, parahippocampal part of the cingulum, and thalamic radiation exhibited the most causal associations with proteins are intriguing (Results section, lines 415-417). However, I would have liked to see a more thorough discussion of these results in the context of existing literature. Specifically, elaborating on the known roles of these tracts in neurodegenerative diseases or cognitive function could provide valuable insights and enhance the overall interpretation of the findings.*

Response: Thank you for your insightful feedback. To provide a more thorough exploration of the roles of the cingulate gyrus part of the cingulum, parahippocampal part of the cingulum, and thalamic radiation in neurodegenerative diseases and cognitive function, we have revised the Discussion sections as follows:

Line 672 of the Discussion: *“Thirty-three significant MR relationships between plasma proteins and brain structures were revealed through MR analysis. The strongest associations with proteins were observed in the cingulate gyrus part of the cingulum, parahippocampal part of the cingulum, and thalamic radiation. The cingulate gyrus part of the cingulum is involved in emotional regulation, attention, and memory, and disruptions in this tract have been implicated in conditions such as AD and major depressive disorder⁵⁷. The parahippocampal cingulum is closely associated with episodic memory and spatial navigation, and alterations in this tract are also observed in the early stages of AD⁵⁷. Our results highlight the potential involvement of the cingulum in the pathophysiology of neurodegenerative and psychiatric disorders through associations with specific proteins. The thalamic radiation serves as a critical pathway for relaying sensory and motor signals between the thalamus and cortical regions. Studies have demonstrated that alteration in the anterior thalamic radiation integrity is linked to schizophrenia and depression^{58,59}. Taken together, these findings suggest that the associated proteins may play key roles in maintaining the structural integrity and functional regulation of these tracts, thereby contributing to the pathophysiology of neurodegenerative and psychiatric disorders.”*.

Comment 22: *Example 5: Please discuss why certain brain regions, such as the precentral cortex, are mediating the association between proteins and mental health (Discussion section, lines 624-630). How do these mediating effects align with what is known about these brain regions in mental health disorders?*

Response: Thank you for your thoughtful feedback. To address how the mediating effects of certain brain regions align with existing knowledge in mental health disorders. We have revised the Discussion section our manuscript as follows:

Line 748 of the Discussion section: *“The precentral cortex, primarily involved in motor control and coordination, has been increasingly recognized for its role in working memory and emotional regulation⁷⁸. Cortical thinning in the precentral gyrus has been associated with an increased risk of depressive symptomatology⁷⁹. In addition, advanced control relevant activation in the precentral gyrus has been linked with suicide risk in mood disorders⁸⁰. The high mediation proportion suggests that ENPP6 may influence mental health indirectly by modulating the structure or function of these regions.”*

Here are point-to-point responses to Reviewers' comments. The Reviewers' comments are marked in blue. All the revisions made to the manuscript are highlighted with red color. In case of multiple questions in one comment, the response to each question is preceded by '- -'.

Here are responses to Reviewer #1's comments.

Comment 1: *The revisions with regards to implying direct causal effects of plasma protein expression on brain structure are satisfactory. The additional proteomics information regards to the management of QCs, and and swapping 'protein concentration' with 'protein expression' are satisfactory. Additional information in regards the participant demographics (Table 1), and supplementary table 5 are welcome and adequately addressed.*

Response: We appreciate your recognition of the improvements made to the manuscript. Thank you for all your insightful and constructive feedback. We would also be glad to make any further revisions.

Comment 2: *Although I can't comment on the validity of MR analysis as it's not my area of expertise, I appreciate the rationale added (line 281-286) and the more conservative conclusions. Can the authors further clarify what is meant by the following statement 'By using genetic variants that are randomly assigned at conception, the MR analysis could minimize the bias'?*

- Is my interpretation correct in that you are using genetic predictors of protein expression to test the association of protein expression on brain structure and on common diseases? This needs to be crystal clear in the paper as its not typical in proteomic analysis.

Response: Thank you for raising this point. We appreciate your acknowledgment of the rationales added and the revised conclusions. As for “By using genetic variants that are randomly assigned at conception, the MR analysis could minimize the bias”, we have provided

an extended explanation as follows: The conventional observational epidemiological studies have traditionally been plagued by issues such as confounding. Instead, the MR approach draws on Mendel's laws of segregation and independent assortment, whereby genetic variants are allocated independently of environment and other genetic factors. Hence, the genetic associations should therefore be largely free from confounding, thus any difference in outcomes between genetically defined groups can be directly attributed to the exposure (Richmond RC, Davey Smith G. Mendelian Randomization: Concepts and Scope. Cold Spring Harb Perspect Med. 2022;12(1):a040501).

Line 285 of the Methods: *“Traditional observational epidemiological studies have long been hindered by challenges such as confounding. Instead, the MR method utilizes Mendel's laws of segregation and independent assortment, which ensure that genetic variants are distributed independently of environmental influences and other genetic factors. Hence, the genetic associations should therefore be largely free from confounding³⁸. This could enhance the reliability of the results by leveraging genetic variants as instrumental variable to assess the MR relationship between protein expression and brain structure.”*

- - As for the understanding of MR analysis, we were using genetic variants of protein expression as instrumental variables to assess the MR relationship between protein expression and brain structure or common diseases. According to your suggestion, we have revised the description to enhance clarity:

Line 342 of the Methods: *“In other words, we leveraged genetic variants as instrumental variables to assess the MR relationship between protein expression and common diseases.”*

Comment 3: *In regards to imputation, the authors state 'all proteins showed a missing rate of less than 25%' Does this mean 25% of the dataset was missing/imputed? Can the authors insert the % missing data or % imputation in the manuscript for transparency so the quality of the proteomics data generated can be clearly interpreted.*

Response: Thank you for bringing this to our attention. All 2,920 proteins showed a missing rate of less than 25%, of which 2,911 proteins actually showed a missing rate of less than 20% and 1,215 proteins had a missing rate of less than 5%. Thus, the overall quality of our proteomic data was satisfactory. According to your suggestion, we have added more details of imputation: Line 155 of the Methods: *“All remaining 2,920 proteins showed a missing rate of less than 25%, of which 2,911 proteins actually showed a missing rate of less than 20% and 1,215 proteins had a missing rate of less than 5%”*

Comment 4: *Line 421 of the Results: “We found that the effects of protein expression were highly correlated between females and males.” (Supplementary Table 6).”*

The authors report protein expression was 'highly correlated' and indeed when a Pearson's correlation is performed a perfect correlation $r = 1$ was achieved as the M/F data compared is exactly the same.

Protein expression values reported (FDR value; SE value) is exactly the same for the male only and females only analysis presented in supplementary table 6. This is uncanny as I find it

unlikely that the expression data for >2000 proteins is exactly the same for the male only and the female only subjects.

Response: We apologize for making a mistake when merging the results into one Excel sheet. We have updated the results of subgroup analysis in a separate sheet as Supplementary Table 10. We observed a correlation of 0.88 for GM volume, 0.86 for cortical area, 0.89 for cortical thickness, 0.91 for WM FA and 0.85 for WM MD. We have also checked the statistics of other analysis in the supplementary tables to ensure that the right version of the results was used.

Comment 5: *Line 250 of the Methods: “Third, to examine potential sex difference, the association analysis was further conducted in female and male separately”.*

Please provide results of this association analysis in females and in males in the manuscript. The authors report results in the rebuttal but not the manuscript. Further the authors should comment on these results. In particular, there is weak correlation between cortical thickness (0.55) and WM FA (0.6) between sexes, they authors need to comment address this impact on interpretation of results.

Response: Thank you for pointing this out. The previous version of correlation (i.e., a correlation of 0.55 for thickness and 0.6 for WM FA) was calculated between sexes across the protein-brain structure pairs that were significant in both females and males. However, due to the limited sample size in subgroup analysis, only few protein-brain structure pairs survived at $P_{FDR} < 0.05$. The correlation between sexes across these limited pairs could not reflect the overall similarity between sexes. Thus, in the revised manuscript, we compared the coefficients (beta) of males and females across 5,358 protein-brain structure pairs that were significant in the whole cohort analysis. This could reflect the overall similarity between sexes more objectively. Accordingly, we have updated the results of the subgroup analysis, which showed a correlation coefficient of 0.88 for GM volume, 0.86 for cortical area, 0.89 for cortical thickness, 0.91 for WM FA and 0.85 for WM MD. In other words, all the five modalities were highly correlated ($R > 0.8$). According to your suggestion, we have incorporated the results of subgroup analysis in the manuscript:

Line 429 of the Results: *“Furthermore, we found that the effects of protein expression were highly correlated between females and males, with a correlation coefficient of 0.88 for GM volume, 0.86 for cortical area, 0.89 for cortical thickness, 0.91 for WM FA and 0.85 for WM MD (Supplementary Table 10).”*

Comment 6: *Table 1: The authors report the age at imaging visit (year). Similarly, the authors need to report the age (years) when the blood samples was taken for proteomics in Table 1. It is noted that blood was sampled at the beginning of the study but its now clear if there is a significant gap in years between imaging and blood sampling.*

Response: Thank you for your insightful comment. According to your suggestion, we have updated Table 1 to include the age at baseline, when the blood sample was taken. The existing time gap between imaging and blood sampling has also been adopted as one of the Limitations. Line 793 of the Limitations: *“Secondly, due to the acquisition schedule of the UK Biobank, there is a time gap between plasma collection and neuroimaging scanning, which could induce*

potential bias. The interval has been accounted for as a covariate. However, future study designs with more synchronized data collection could address this concern.”

Comment 7: *Direction/ referencing corresponding results files in supplementary is unsatisfactory. When referring to results the reader should be pointed to the specific results table in supplementary.*

Response: Thank you for your detailed feedback regarding the reference of the supplementary tables. To link the description of the results directly to the specific supplementary table, we have split the Supplementary Table 6 into four separate tables and updated all the references to supplementary tables in the manuscript:

Line 424 of the Results: *“In sensitivity analyses, compared to associations with only protein expression, age at imaging visit, sex, total intracranial volume and scanning site as the predictors, the inclusion of additional covariates didn’t change the pattern of significant associations (Supplementary Table 8). Additionally, the imputation procedure did not affect the pattern of significant associations between protein and brain structure (Supplementary Table 9). Furthermore, we found that the effects of protein expression were highly correlated between females and males, with a correlation coefficient of 0.88 for GM volume, 0.86 for cortical area, 0.89 for cortical thickness, 0.91 for WM FA and 0.85 for WM MD (Supplementary Table 10).”*

Comment 8: *Please provide a reference for this statement, second paragraph in the introduction*

‘Measurement of the reduction of complement proteins can be used to predict the extent of brain atrophy in mild cognitive impairment patient’

Response: Thank you for this helpful suggestion. According to your suggestion, we have added an appropriate reference to support this statement in the second paragraph of the introduction:

Line 90 of the Introduction: *“Measurement of the reduction of complement proteins was associated with the extent of brain atrophy in mild cognitive impairment patients (Li, M. et al. Alzheimer’s Research & Therapy. 2024)”*

Comment 9: *In regards to Figure 6/ Results shown the association between protein and disease outcome. The authors should comment carefully on interpretation of these results, particularly as the age profile of subject included is 40-69 years, and the plasma proteomic profile in younger subjects is likely to differ. The disease associated proteins identified may reflect later plasma protein markers of the disease, particularly as ASD, ADHD, Scz, BPD, Anxiety have a onset in childhood and/or early adulthood. Therefore these bloods plasma changes have limited prognostic value, and future validation is needed to identify if the disease associated protein changes are longitudinal or reflect late stages of the disease.*

Response: Thank you for your thoughtful comment. We acknowledge that the plasma proteomic profile in younger individuals may differ from that observed in our study cohort (aged 40–69 years). As a result, the disease-associated proteins identified here may primarily

reflect later-stage changes rather than early biomarkers. According to your suggestion, we have revised the discussion to clarify this limitation:

Line 808 of the Discussion section: *“Sixthly, the plasma proteomic profile in younger individuals may differ from that observed in our cohort. The disease-associated proteins identified in this study may reflect later-stage plasma protein markers, rather than early biomarkers, especially considering that disorders such as ASD, ADHD, schizophrenia, BPD, and anxiety typically have onset in childhood or early adulthood. Consequently, future validation is needed to identify if the disease-associated protein changes are longitudinally predictive of incident diseases.”*

Here are responses to Reviewer #2's comments.

Comment 1: *The authors have addressed all my concerns.*

Response: Thank you for recognizing the improvements in our work and providing constructive comments.

Here are responses to Reviewer #3 and #4's comments.

Comment 1: *Thank you for addressing the overall comments which were:*

“In the Introduction, the rationale for the study could be expanded to better highlight key gaps in the field. The section would benefit from additional references that provide context and emphasize the significance of addressing these gaps, helping to frame the study's objectives more clearly.

In the Methods section, several methodological decisions are presented without adequate citation of relevant literature. Including references for key procedures and criteria would enhance the transparency and reproducibility of the study. This is particularly important for justifying the thresholds used in genetic analyses and other statistical methods.

In the Discussion, the results are somewhat summarized but lack deeper reflection on their implications. It would be beneficial to explore how these findings address existing gaps and to provide a clearer explanation of the complementary insights offered by the results, demonstrating how they collectively advance our understanding of the topic.

Additionally, there are some grammatical inconsistencies throughout the text that affect its clarity and readability. A thorough review and revision would greatly enhance the manuscript's overall presentation and ensure a smoother reading experience.”

Response: We appreciate your recognition of the improvements made to the manuscript. Thank you for all your insightful and constructive feedback. We would also be glad to make any further revisions.

Comment 2: *Here are the responses to the specific comments which were shared with the author.*

METHODS

Comment 6: The rationale for selecting five imaging-derived phenotypes (IDPs)—specifically grey matter volume, surface area, and cortical thickness, along with white matter fractional anisotropy (FA) and mean diffusivity (MD)—from the multiple IDPs released by the UK Biobank is not clearly explained. The authors should provide a clear justification for this selection.

Response: Thank you for pointing this out. As we are only interested in brain structures that could be stably measured, the functional IDPs in UKB are beyond the scope of this manuscript. To provide a clearer rationale for selecting the five imaging-derived phenotypes (IDPs) in our study, we have revised the manuscript to include a detailed justification for our choices, explaining how each IDP contributes to our understanding of brain structure in the context of our research objectives: Line 160 of the Methods section: “Given that gray and white matter are crucial components that serve distinct functions for overall brain health, here we included brain structural imaging derived phenotypes (IDPs) that were previously made available by the UKB neuroimaging group¹⁹. Similar to previous researches, regional grey matter measures (volume, thickness, and area) derived from T1-weighted images and microstructural measures derived from diffusion weighted imaging, including fractional anisotropy (FA) and mean diffusivity (MD), were chosen to be used in the present study²¹.”

Reviewer response: The comment was acknowledged and the author clarifies that gray and white matter are crucial for brain health and are stable in nature, hence chosen for analysis.

Response: Thank you for recognizing the improvements in our work and providing constructive comments.

Comment 3: *Comment 7: The authors should cite major UK Biobank publications that introduced imaging derived phenotypes (IDPs) and proteomics.*

Response: We apologize for missing the major UKB publication that introduced IDPs and proteomics. We have now added the corresponding citations as follows: Line 162 for IDPs: (Thomas J. Littlejohns et al., Nature Communications, 2020) Line 151 for proteomics: (Benjamin B. Sun et al., Nature, 2023)

Reviewer response: The comment was acknowledged and the author has added the necessary references.

Response: We appreciate your recognition of the enhancements in our work and your helpful comments.

Comment 4: *Comment 8: In the "Genetic Data Processing and Analysis" section, several key concepts and decisions lack citations. Relevant references should be added to support these points. For example, the exclusion criteria for single nucleotide polymorphisms (SNPs)—those with a missing rate > 5%, minimum minor allele frequency (MAF) < 0.1%, or Hardy - Weinberg equilibrium test $P < 1 \times 10^{-10}$ —should be backed by appropriate references or a more detailed explanation, as these decisions can significantly influence the robustness and generalizability of the findings.*

Response: We apologize for missing the key citations supporting the key methodology decisions. According to your suggestions, we have added the appropriate references as follows: Line 178 of the Methods section: (Clare Bycroft et al., Nature, 2018; Yuzhu Li et al., Nature Aging, 2022) Reviewer response: The comment was acknowledged and the author has added the necessary references (Bycroft et al., 2018; Li et al., 2022), which are strong choices for supporting SNP quality control.

Response: Thank you for helping improve our manuscript.

Comment 5: *Comment 9: In the “GWAS summary of CNS disorders,” GWAS references are missing for the neurodegenerative diseases. Please provide. Also, include a rationale for choosing these four neurodegenerative disorders along with the sample sizes for these studies in the main document (in addition to the range of N).*

Response: We apologize for not including the introduction of the four neurodegenerative diseases. We have now added the rationale and relevant GWAS references for the selected neurodegenerative diseases as follows: Line 194 of the Methods section: “Given their significance in public health and the relatively large number of cases, GWAS summary statistics were collected from publicly available databases for six psychiatric disorders (attention deficit hyperactivity disorder (ADHD, cases/total=19,099/53,293), anxiety (cases/total=7,016/18,186), autism spectrum disorder (ASD, cases/total=18,381/46,350), bipolar disorder (BP, cases/total=20,352/51,710), major depressive disorder (MDD, cases/total=45,396/142,646), schizophrenia (SCZ, cases/total=67,390/161,405)), and four neurodegenerative disorders (Mitja I. Kurki et al., Nature, 2023) (Alzheimer’s disease (AD, cases/total=10,520/401,661), Parkinson’s disease (PD, cases/total=4,681/407,500), multiple sclerosis (MS, cases/total=2,409/408,561) and amyotrophic lateral sclerosis (ALS, cases/total=531/184,000))”.

Reviewer response: The psychiatric GWAS sample sizes vary significantly, with anxiety disorder having a much smaller N than schizophrenia or MDD.

Similarly, ALS has a very small case count (531 cases out of 184,000 total) compared to AD or PD. Is this accounted for in the analysis? If so, how?

Response: Thank you for raising this point. According to previous research (Eun Pyo Hong et al., *Genomics Inform*, 2012), the minimum number of cases required to reach an α of 5% was 1,974 cases for a SNP with a MAF of 5%. Under the same assumption, a smaller number of cases was required for a SNP with a larger MAF. While the anxiety disorder has less case count than other disorders, they met this minimum requirement ($N_{case} > 1,974$) and could achieve similar statistical power in theory. Given the smaller sample sizes for ALS compared to other disorders, we acknowledge that there may be false negative results due to limited power. In response to your suggestion, we have explicitly discussed this point as a potential limitation: Line 750 in the Discussion section: *“Given the smaller sample sizes for anxiety and ALS compared to other disorders, the non-significant MR-relationships may potentially due to the impact of sample size rather than indicating a lack of significance. Therefore, further studies based on larger case count are warranted to validate our findings.”*

Comment 6: *Comment 10: Please provide citations for key methodological decisions used in the "Causal relationship between protein concentration and brain structure" and 'eQTL mapping and colocalization' sections. For example: MR-Egger analysis (line 277), leave-one-out-analysis (line 275) etc.*

Response: Thank you for raising these issues. We recognize that detailed citation for key methodologies could strengthen the validity of our manuscript. According to your suggestions, we have added citations for MR-Egger analysis in line 312 (Jack Bowden et al., International Journal of Epidemiology, 2015), leave-one-out-analysis in line 309 (Gibran Hemani et al., Elife, 2018). According to the 2nd reviewer 's suggestion, we have replaced the colocalization analysis with Summary-based Mendelian Randomization (SMR) analysis. Therefore, we did not include citation for colocalization but we added citation for SMR in line 328 (Zhihong Zhu et al., Nature Genetics, 2016).

Reviewer response: The comment was acknowledged and the author has added the necessary references.

Response: Thank you for your valuable input in enhancing our manuscript.

Comment 7: *Comment 11: Provide a definition or brief explanation of "colocalization analysis" in the "Gene Expression and Quantitative Trait Loci" section for clarity (either in the main or supplementary documents).*

Response: Thanks for your suggestion. According to the 2nd reviewer 's suggestion, we have replaced the colocalization analysis with Summary-based Mendelian Randomization (SMR) analysis. Therefore, following your suggestion, we have added an explanation of "SMR analysis". Line 327 of the Methods: "Specifically, for each brain structure-associated protein surviving MR analysis, we used the SMR method⁴² to integrate summary-level pQTL data with eQTL data to identify genes whose protein expression in plasma are associated with their expression levels in the brain because of pleiotropy. The heterogeneity in dependent instruments (HEIDI) test was conducted and the significance threshold was set to $PSMR < 0.05/32$ and $PHEIDI > 0.05$."

Reviewer response: The comment was acknowledged and the author has added the needed explanation for SMR.

Response: Thank you for your valuable suggestion in improving our work.

Comment 8: *Comment 12: For improved readability, please present the "Statistical analyses" subsections in the same order as they appear in the Results section.*

Response: Thank you for your suggestion. We have reorganized the "Statistical Analyses" subsections to align with the order in which they appear in the Results section. We hope this adjustment could enhance readability and make it easier for readers to follow the analyses.

Reviewer response: Clearly acknowledges the change and explains the reasoning.

Response: Thank you for recognizing the advancements in our work and sharing your constructive feedback.

Comment 9: *Comment 13: Clarify whether the physical annotation method, which calculates the density of proteins associated with brain structure on each chromosome, is based on a previously published reference or an in-house method.*

Response: We apologize for missing the methodology details of the physical annotation. The mapping of the protein-coding genes to chromosome positions was based on a previously established topR software (Thorhildur Juliusdottir, BMC Bioinformatics, 2023). The definition of the density used an in-house method that dividing the number of protein-coding genes of each chromosome by the length of each chromosome. We have incorporated detailed descriptions: Line 257 of the Methods section: “The physical annotation was performed by first mapping the protein-coding genes to chromosome positions with a previously established topR software³⁴. Subsequently, the number of protein-coding genes on each chromosome was divided by the length of that chromosome, which generated the density of associated proteins in each chromosome that accounted for the length of the chromosome.”

Reviewer response: Clarifies whether a published method or in-house approach was used.

Response: We appreciate your deep insights into our manuscript.

Comment 10: *Comment 14: Include a concise definition or explanation of horizontal and directional pleiotropy in the "Causal relationship between protein concentration and brain structure" section.*

Response: We apologize for ignoring the definition of horizontal and directional pleiotropy. To give a better understanding of the Mendelian randomization results, we have incorporated the definitions in the Methods section: Line 309 of the Methods section: “Secondly, the MR-PRESSO technique was used to detect possible horizontal pleiotropy⁴⁰, when a genetic variant influences the brain structure (directly or indirectly, through other traits) independently of the hypothesized protein. Thirdly, MREgger regression analysis was used to detect possible directional pleiotropy⁴¹, when a genetic variant associated with protein levels influences brain structure through a shared pathway rather than a direct effect of the protein on the brain structure.”

Reviewer response: Adds missing definitions in the Methods section, thank you for the addition.

Response: Thanks again for your constructive feedbacks.

Comment 11: *Comment 15: I also suggest reconsidering the subheading to explicitly mention both eQTL and pQTL for clarity. It would be good to have these concepts defined concisely in supplementals.*

Response: Thank you for this helpful comment. We have revised the subheading “Colocalization of plasma proteins and brain gene expression” to “Pleiotropic association between plasma pQTL and brain eQTL” in line 320 of the Methods section. Meanwhile, we have added the brief explanation of eQTL and pQTL in the Methods: Line 210 of the Methods: “Here we used pQTL and eQTL summary data to characterize genetic variants of protein and gene expression separately. A “pQTL” stands for “protein quantitative trait locus,” which refers to a genetic variant associated with the abundance of a specific protein, while an “eQTL”

stands for "expression quantitative trait locus," indicating a genetic variant linked to the expression level of a gene (mRNA transcript) at a specific locus on the chromosome. "

Reviewer response: Clarifies subheading and adds definitions. Thank you.

Response: Thank you again for sharing your insightful feedback on our manuscript.

Comment 12: *Comment 16: Please provide N of the UKB subset used in the "Mediation analyses" section of the main document. Additionally, could the authors provide a rationale or reference for the chosen correlation cutoff of 0.55 used to identify and remove high collinearity between protein associated structure measures in the iterative strategy?*

Response: Thanks for your professional suggestions. We agree that the demographical information of UKB subset used in Mediation analysis should not only be provided in Supplementary Table 4 but also be briefly stated in the Methods section. Accordingly, we have added the statement as follows: Line 346: "A total of 3,270 participants with a mean age of 54.2 were eligible for inclusion in mediation analysis (Supplementary Table 4)". - - The correlation cutoff of 0.55 was selected because it can ensure the weights of all surviving observational variables contributing to latent brain structures fall within the range [-1,1], while minimizing the removal of observational brain structure measurements. We have included the rationale as follows: Line 357: "By requiring the weights of all surviving observational variables contributing to latent structural measures fell within the range [-1,1], while minimizing the removal of observational brain structure measurements, a correlation threshold of 0.55 was chosen".

Reviewer response: Adds missing N and explains correlation cutoff selection.

Response: Thank you for your recognition of the enhancements made to our manuscript and for your constructive insights.

Comment 13: DISCUSSION

The discussion section has been substantially improved, addressing key concerns related to interpretation, contextualization, and citation density. The added discussions on brain regions, chromosomal associations, white matter tracts, and protein-mental health mediation effects enhance the manuscript's contribution by framing it within the context of neurodegenerative and psychiatric research.

Response: Thank you for all your constructive comments and the recognition of the improvement made to the manuscript. We would be glad to respond to any further questions and comments that you may have.

Comment 14: *Comment 18: Example 1: The statement regarding the highest number of significant associations observed for WM MD (2,501), followed by GM volume, surface area, thickness, and the fewest for WM FA (86), is clear in terms of the ranking (results section, lines 362-364). However, it would greatly benefit from additional context and interpretation. Could the authors provide an explanation for why WM MD showed the highest number of associations, and why WM FA had the least? A discussion on the potential biological or methodological*

reasons behind these differences would enhance the reader's understanding and strengthen the results' interpretation.

Response: Thank you for your helpful comment. We have revised the Discussion sections to provide additional context and interpretation regarding the observed differences in the number of significant associations for WM MD and FA: Lines 637 of the Discussion section: “Though MD and FA are both metrics that reflect the microstructural integrity of white matter, MD demonstrated the most significant associations with proteins, whereas FA showed the least significant associations. The higher number of associations between MD and CSF proteins might stem from the fact that CSF proteins reflect early pathophysiological changes in the brain, to which MD is sensitive, especially in the early stages of neurological diseases such as AD45. In patients with MS, MD was more strongly associated with widespread brain degeneration than FA as well46. FA is primarily influenced by directional water diffusion and is more specific to structural coherence and fiber integrity47. This specificity may make FA less sensitive to diffuse or generalized changes and more reliant on pronounced structural disruptions.” .

Reviewer response: Thank you for the additional context and response. Also, the references used aptly support the statements made in the discussion section.

Response: Thank you again for your helpful comment and the recognition of the improvement made to the manuscript.

Comment 15: *Comment 19: Example 2: Six proteins (NCAN, SLITRK1, MOG, LEP, OXT, PAEP) were highlighted in lines 549-552 for their numerous associations with brain structure. However, there is no discussion of their established roles in the brain, apart from a brief mention that NCAN is linked to brain volume. Expanding on the relevance of the top five proteins showing positive and negative associations to brain structure (mentioned in results section, lines 369-372) in a cohesive manner would enhance the discussion: NCAN, SLITRK1, MOG, PTPRN2 and SEZ6L; LEP, OXT, PAEP, CCN5 and XG.*

Response: Thank you for your valuable feedback. To provide a more comprehensive exploration of the established roles of the top proteins showing positive and negative associations with brain structure, we have added the description of the proteins: Line 647 of the Discussion “NCAN, SLITRK1, MOG, PTPRN2 and SEZ6L were the top proteins that showed the highest total number of significant positive associations with brain structures. The links between these proteins and brain disorders have been reported in previous studies. NCAN, an extracellular matrix protein, is closely related to the development, neuronal senescence and apoptosis of the neurons. NCAN has been found to be associated with neuropsychiatric and neurodegenerative diseases48. In addition, NCAN demonstrated a predominant correlation with AD pathology49. Existing evidence has linked the plasma concentration of NCAN protein to total brain volume18. The findings of the present study have expanded the significant associations of NCAN protein to the volume of 50 regions, surface area of 43 regions, FA of three tracts and MD of five tracts. Apart from NCAN, the other proteins are also relevant to central neural system and brain diseases. SLITRK1 function is associated with the control of neurite development and synaptogenesis50. SLITRK has been linked to Tourette ’ s syndrome,

*trichotillomania, and obsessive-compulsive disorder*⁵¹. *MOG, expressed in oligodendrocytes in the central nervous system, is an encephalitogenic protein that can trigger a demyelinating immune response*⁵². *Moreover, PTPRN2 has been implicated in AD and FTD*⁴⁹, while *SEZ6L has been associated with bipolar disorder*⁵³. *The proteins with significant negative associations with brain structure include LEP, OXT, PAEP, CCN5, and XG. LEP is well-known for its role in metabolic regulation and has been implicated in neurodegenerative diseases, potentially linking metabolic dysfunction to brain pathology*⁵⁴. *OXT is associated with social behaviors and stress responses*⁵⁵. *It has also been implicated in anxiety, depression and AD*⁵⁶, although its relationship with brain structure remains unclear. *Previous studies have highlighted associations between these proteins and CNS or brain disorders, and our findings further reveal their associations with brain structure. While the roles of PAEP, CCN5, and XG in brain diseases are not yet fully understood, our study identifies significant associations between these proteins and brain structure, offering a valuable basis for future studies.*”

Reviewer response: Thank you for the additional context and response. It adequately addresses the comment.

Response: Thank you for providing constructive comments to improve our manuscript.

Comment 16: *Comment 20: Example 3: The investigation into the chromosomal locations of brain structure associated proteins lacks context. The results (Results section, lines 383-385) show a concentration on chromosomes 19, 17, and 22, yet no explanation is provided for why this might be significant. A discussion on the potential links of these chromosomes to brain diseases would add valuable context to these findings.*

Response: Thanks for your suggestion. We have included an expanded discussion on the significance of the observed concentration of these proteins on chromosomes 19, 17, and 22, highlighting their potential links to brain diseases: Line 687 of the Discussion: “Our analysis revealed that brain structure-associated proteins are disproportionately localized to chromosomes 19, 17, and 22, relative to the relatively short length of these chromosomes. The concentration of brain structure-associated proteins on chromosomes 19, 17, and 22 is particularly noteworthy given the established relevance of these chromosomes to brain health and disease. Our findings show that many of the proteins identified on these chromosomes, including those encoded on chromosome 17 (OMG60, BAIAP261, GFAP62), chromosome 19 (NCAN49, APLP163, GDF1564), and chromosome 22 (NPTXR65, OSM66, RTN4R67) are strongly associated with brain structure and have been previously implicated in various neurological and psychiatric disorders. These results suggest that these chromosomes may serve as hotspots for genes influencing brain morphology and function, reflecting their broader role in neurodevelopmental and disease pathways.”

Reviewer response: Your response adequately addresses the comment but could be strengthened by adding disease relevance for each chromosome explicitly. Example: The concentration of brain structure-associated proteins on chromosomes 19, 17, and 22 is particularly notable given their established relevance to brain health and disease. Chromosome 19 is highly gene-dense and includes APOE, a key risk factor for Alzheimer’s disease. Chromosome 17 harbors genes implicated in neurodegenerative disorders, including

tauopathies such as frontotemporal dementia. Chromosome 22 has been linked to neurodevelopmental and psychiatric conditions, such as schizophrenia and 22q11.2 deletion syndrome. (add necessary refs)

Response: Thank you for your insightful suggestion. In response to your comment, we have revised the manuscript to explicitly discuss the disease relevance of chromosomes 19, 17, and 22:

Line 702 of the Discussion section: *“Chromosome 19 has a high gene density and contains key genetic factors related to brain aging and neurodegenerative diseases, including APOE, which is strongly linked to AD⁶⁰. Chromosome 17 includes MAPT, a gene associated with tau-related pathologies, such as FTD⁶¹. Chromosome 22 has been implicated in neurodevelopmental and psychiatric disorders, with alterations in this region contributing to schizophrenia and 22q11.2 deletion syndrome^{62,63}.”*

Comment 17: *Comment 21: Example 4: The findings that the cingulate gyrus part of the cingulum, parahippocampal part of the cingulum, and thalamic radiation exhibited the most causal associations with proteins are intriguing (Results section, lines 415-417). However, I would have liked to see a more thorough discussion of these results in the context of existing literature. Specifically, elaborating on the known roles of these tracts in neurodegenerative diseases or cognitive function could provide valuable insights and enhance the overall interpretation of the findings.*

Response: Thank you for your insightful feedback. To provide a more thorough exploration of the roles of the cingulate gyrus part of the cingulum, parahippocampal part of the cingulum, and thalamic radiation in neurodegenerative diseases and cognitive function, we have revised the Discussion sections as follows: Line 672 of the Discussion: “Thirty-three significant MR relationships between plasma proteins and brain structures were revealed through MR analysis. The strongest associations with proteins were observed in the cingulate gyrus part of the cingulum, parahippocampal part of the cingulum, and thalamic radiation. The cingulate gyrus part of the cingulum is involved in emotional regulation, attention, and memory, and disruptions in this tract have been implicated in conditions such as AD and major depressive disorder⁵⁷. The parahippocampal cingulum is closely associated with episodic memory and spatial navigation, and alterations in this tract are also observed in the early stages of AD⁵⁷. Our results highlight the potential involvement of the cingulum in the pathophysiology of neurodegenerative and psychiatric disorders through associations with specific proteins. The thalamic radiation serves as a critical pathway for relaying sensory and motor signals between the thalamus and cortical regions. Studies have demonstrated that alteration in the anterior thalamic radiation integrity is linked to schizophrenia and depression^{58,59}. Taken together, these findings suggest that the associated proteins may play key roles in maintaining the structural integrity and functional regulation of these tracts, thereby contributing to the pathophysiology of neurodegenerative and psychiatric disorders.”

Reviewer response: Your response effectively addresses the comment. Thank you for the explanation.

Response: Thank you again for your helpful comment and the recognition of the improvement made to the manuscript.

Comment 18: *Comment 22: Example 5: Please discuss why certain brain regions, such as the precentral cortex, are mediating the association between proteins and mental health (Discussion section, lines 624-630). How do these mediating effects align with what is known about these brain regions in mental health disorders?*

Response: Thank you for your thoughtful feedback. To address how the mediating effects of certain brain regions align with existing knowledge in mental health disorders. We have revised the Discussion section our manuscript as follows: Line 748 of the Discussion section:

“The precentral cortex, primarily involved in motor control and coordination, has been increasingly recognized for its role in working memory and emotional regulation⁷⁸. Cortical thinning in the precentral gyrus has been associated with an increased risk of depressive symptomatology⁷⁹. In addition, advanced control relevant activation in the precentral gyrus has been linked with suicide risk in mood disorders⁸⁰. The high mediation proportion suggests that ENPP6 may influence mental health indirectly by modulating the structure or function of these regions.”

Reviewer response: Your response effectively addresses the comment. Thank you for the explanation.

Response: Thanks again for all your constructive and valuable comments.

Here are point-to-point responses to Reviewers' comments. The Reviewers' comments are marked in blue. All the revisions made to the manuscript are highlighted with red color.

Here are responses to Reviewer's comments.

Comment 1: *The additional information and improvements is appreciated and is important in interpreting the findings. It is noteworthy that there is a 9 year gap between blood sampling for proteomics and the brain imaging which impacts interpretation of findings. Although I appreciate the authors note this gap was included as a covariate in the analysis. Further comments should be made in regards to caution when interpreting these proteomic findings given important literature demonstrating that the plasma proteome is known to change with age.*

Lehallier, B. et al. Undulating changes in human plasma proteome profiles across the lifespan. Nat. Med. 25, 1843 – 1850 (2019).

Niu, L., Stinson, S.E., Holm, L.A. et al. Plasma proteome variation and its genetic determinants in children and adolescents. Nat Genet 57, 635 – 646 (2025). <https://doi.org/10.1038/s41588-025-02089-2>.

Response: We appreciate your recognition of the improvements made to the manuscript. Thank you for all your insightful and constructive feedback. According to your comment, we have incorporated the related reference to the second limitation:

Line 547: *“Secondly, due to the acquisition schedule of the UK Biobank, there is a time gap between plasma collection and neuroimaging scanning. Given that plasma proteome could change with age, this may induce potential bias^{62,63}. The interval has been accounted for as a covariate. However, future study designs with more synchronized data collection could address this concern.”*